# Targeting PRMT3 impairs methylation and oligomerization of HSP60 to boost anti-tumor immunity by activating cGAS/STING signaling

Yunxing Shi[1,2,3,5], Zongfeng Wu[1,2,5], Shaoru Liu[1,2,5], Dinglan Zuo[1,2,5], Yi Niu[1,2], Yuxiong Qiu[1,2], Liang Qiao [1,2], Wei He[1,2], Jiliang Qiu[1,2], Yunfei Yuan [1,2] ✉, Guocan Wang [4] ✉ & Binkui Li [1,2] ✉

Immune checkpoint blockade (ICB) has emerged as a promising therapeutic option for hepatocellular carcinoma (HCC), but resistance to ICB occurs and patient responses vary. Here, we uncover protein arginine methyltransferase 3 (PRMT3) as a driver for immunotherapy resistance in HCC. We show that PRMT3 expression is induced by ICB-activated T cells via an interferon-gamma (IFNγ)-STAT1 signaling pathway, and higher PRMT3 expression levels correlate with reduced numbers of tumor-infiltrating CD8[+] T cells and poorer response to ICB. Genetic depletion or pharmacological inhibition of PRMT3 elicits an influx of T cells into tumors and reduces tumor size in HCC mouse models. Mechanistically, PRMT3 methylates HSP60 at R446 to induce HSP60 oligomerization and maintain mitochondrial homeostasis. Targeting PRMT3-dependent HSP60 methylation disrupts mitochondrial integrity and increases mitochondrial DNA (mtDNA) leakage, which results in cGAS/STING-mediated anti-tumor immunity. Lastly, blocking PRMT3 functions synergize with PD-1 blockade in HCC mouse models. Our study thus identifies PRMT3 as a potential biomarker and therapeutic target to overcome immunotherapy resistance in HCC.

Hepatocellular carcinoma (HCC), the most common form of adult primary liver cancer, is one of the deadliest cancers and remains the fourth leading cause of cancer-related death worldwide, despite advancements in treatment options for late-stage HCC[1]. New therapeutic agents have demonstrated efficacy in clinical trials, such as targeted therapies (e.g., sorafenib, lenvatinib, and regorafenib) and the anti-angiogenic antibody ramucirumab, but the benefit is often modest and transient[2]. While immune checkpoint blockade (ICB) (e.g., nivolumab and pembrolizumab) offers new promise to patients with HCC with the potential of durable responses[3], only 20–30% of HCC patients benefit from immunotherapy due to primary resistance[4]. Additionally, those patients who initially respond to treatment may develop acquired resistance to ICB[5,6]. Therefore, it is crucial to understand the molecular mechanisms underlying the evasion of

[1]State Key Laboratory of Oncology in South China, Guangdong Provincial Clinical Research for Cancer, Sun Yat-Sen University Cancer Center, Guangzhou, China. [2]Department of Liver Surgery, Sun Yat-Sen University Cancer Center, Guangzhou, China. [3]Department of Colorectal Surgery, Guangdong Institute of Gastroenterology, and Guangdong Provincial Key Laboratory of Colorectal and Pelvic Floor Diseases, The Sixth Affiliated Hospital of Sun Yat-sen University, Guangzhou, China. [4]Department of Genitourinary Medical Oncology, The University of Texas MD Anderson Cancer Center, Houston, TX, USA. [5]These authors contributed equally: Yunxing Shi, Zongfeng Wu, Shaoru Liu, Dinglan Zuo. ✉e-mail: yuanyf@mail.sysu.edu.cn; gwang6@mdanderson.org; libk@sysucc.org.cn

T cell-mediated anti-tumor immunity in HCC, which allows us to identify novel therapeutic strategies for improving the response of HCC patients to immunotherapy.

Multiple mechanisms, including cancer cell-intrinsic and tumor microenvironment (TME)-mediated immunosuppression, have been implicated in the de novo resistance and acquired resistance to immunotherapy[7–9]. Correspondingly, targeting these resistant mechanisms improves the response to immunotherapy in preclinical models. Yet, the mechanisms driving the resistance to immunotherapy are still poorly defined in HCC. Regulators of post-translational modifications (PTM), including histone lysine methyltransferases (KMT), histone lysine demethylase (KDM), and protein arginine methyltransferases (PRMT), participate in key biological processes[10] and have emerged as critical players in tumorigenesis and therapeutic resistance (e.g., chemotherapy, targeted therapy and immunotherapy) in various cancer types[11–14]. However, the roles of KMTs, KDMs, and PRMTs in the therapeutic resistance to ICB in HCC remain poorly defined.

Here, through genome-wide transcriptomic, we identify PRMT3 as a key driver of immunotherapy resistance in HCC. Also, we define interferon-gamma (IFNγ)−STAT1 pathway induced *PRMT3* expression in response to anti-PD1 therapy as an adaptive response; PRMT3 induces the methylation of HSP60, which is required for its oligomerization and is essential for mitochondrial homeostasis, to suppress the activation of cGAS/STING pathway. This study provides a potential option to improve efficacy of immunotherapy in HCC.

## Results

### High PRMT3 expression was strongly associated with poorer response to immunotherapy in HCC patients

T cell-mediated anti-tumor immunity plays a critical role in tumor progression[15]. The abundance of CD8$^+$ cytotoxic T cells has been shown to be a prognostic marker for patient survival in many cancer types, including HCC[16]. Since KMTs, KDMs, and PRMTs are key players involved in tumorigenesis and therapeutic resistance, we decided to examine whether the overexpression of these genes is positively or negatively correlated with CD8$^+$ T cell abundance in HCC. We performed tumor deconvolution of the well-annotated TCGA-LIHC dataset[17] using two different methods, ImmuCellAI and CIBERSORT[18,19]. We then determined the correlation between the expression of the individual KMT/KDM/PRMT gene with cytotoxic signature score and CD8$^+$ T cell infiltration, respectively (Fig. 1A). We found that the expression of 16 genes showed a striking negative correlation with both cytotoxic signature score and CD8$^+$ T cell infiltration, whereas only PRDM16 and FBL were positively correlated with both cytotoxic signature score and CD8$^+$ T cell infiltration (Fig. 1A and Supplementary Fig. 1A–D). Among these 16 genes showing a negative correlation with cytotoxic score and CD8$^+$ T cell infiltration, we found that higher expression levels of 7 of them are strongly associated with shorter overall survival in HCC patients (Fig. 1A and Supplementary Fig. 2A). Thus, these data suggest that these 7 genes may suppress T cell-mediated anti-tumor immunity and promote tumor progression. Because we previously showed that PRMT3 plays a crucial role in the therapeutic resistance to oxaliplatin in HCC[13], we decided to focus our attention on PRMT3 (Fig. 1B, C). Further analysis indicated that PRMT3 was also associated with shorter progression-free survival in HCC patients in TCGA-LIHC dataset (Supplementary Fig. 2B). We also found that high PRMT3 expression levels were negatively correlated with the abundance of T cells infiltration in several cancer types in the Tumor Immune Dysfunction and Exclusion (TIDE) database[20] (Supplementary Fig. 2C). To confirm the findings from the TCGA-LIHC dataset, we also examined the expression of PRMT3 and its correlation with patient survival using an in-house dataset without ICB treatment (the SYSUCC cohort) with detailed patient outcomes. Consistent with the findings from the TCGA-LIHC dataset, higher PRMT3 expression was also strongly associated with shorter overall survival and progression-free

survival of HCC patients in the SYSUCC cohort (Fig. 1D). We then examined the expression of PRMT3 in HCC tumor samples and the corresponding adjacent normal liver tissues. We found that PRMT3 protein and mRNA expression levels were higher in HCC tumor tissues than in the adjacent normal tissues (Supplementary Fig. 3A−C). Also, HCC tumor samples with higher PRMT3 expression had clinicopathologic features associated with more aggressive disease, such as larger tumor sizes, advanced tumor stages, and higher AFP levels (Supplementary Fig. 3D−G). To confirm the negative correlation of PRMT3 expression with the abundance of tumor-infiltrating CD8$^+$ T cells in the RNA-seq analyses, we performed multiplex immunofluorescence (mIF) staining of PRMT3, CD45, CD4, and CD8 on 20 cases of HCC samples. We found that high PRMT3 expression levels were negatively correlated with the abundance of CD8$^+$ T cells in HCC tumor samples (Fig. 1E, F).

Since the abundance of CD8$^+$ T cell infiltration was closely related to the efficacy of immunotherapy[15], we then explored whether higher PRMT3 expression levels in pre-treatment samples were associated with poorer responses in HCC patients who received anti-PD-1/PD-L1 therapy in two independent cohorts of patients at SYSUCC. The responses of the patients were defined by the Response Evaluation Criteria in Solid Tumors (RECIST; ver. 1.1) and modified RECIST (mRECIST) criteria[21]. The levels of PRMT3 expression were defined by IHC staining intensity scores determined by two independent pathologists. We found that patients with low PRMT3 expression had a better response to anti-PD-1/PD-L1 therapy, which was defined by the analysis of MRI imaging, than patients with high PRMT3 expression (Fig. 1G). In cohort 1, PRMT3-low patients had a higher objective response rate (ORR) than PRMT3-high patients after receiving immunotherapy when evaluated with both RECIST and mRECIST criteria (Fig. 1H). Importantly, PRMT3-low patients had better overall survival and progression-free survival than PRMT3-high patients (Fig. 1I). Also, our findings from cohort 1 were confirmed in cohort 2, as PRMT3-high patients had lower ORR rates and shorter overall and progression-free survival (Fig. 1J, K). Similarly, high expression of PRMT3 was strongly associated with shorter overall survival after receiving immunotherapy (Supplementary Fig. 3H) in two separate cohorts from published studies[22,23]. Collectively, our data strongly suggest that high PRMT3 expression was negatively correlated with the abundance of T cell infiltration and strongly associated with poor prognosis and poor response to immunotherapy in HCC patients.

### Effector T cells induced by anti-PD1 therapy up-regulate PRMT3 expression via IFNγ-STAT1-dependent pathways

By analyzing PRMT3 expression in HCC patients who received immunotherapy, we found that PRMT3 was notably higher in tumor samples from patients who received anti-PD-1/PD-L1 therapy at both mRNA and protein levels than in tumor samples from patients who did not (Fig. 2A, B and Supplementary Fig. 4A). Using a well-established HCC syngeneic cell line model, Hepa1-6, which displays partial response to ICB[7], we found that anti-PD1 immunotherapy-treated tumors also had increased PRMT3 mRNA and protein expression (Fig. 2C and Supplementary Fig. 4B), which is consistent with the observed effects in human HCC samples. Since it was previously shown that activated CD8$^+$ T cells induced gene expression of cancer cells in ICB-treated tumors[24], we speculated that the increase in PRMT3 expression could be caused by the anti-PD1/PD-L1 treatment. To test this, we examined the effect of conditioned medium (CM) prepared from cultured mouse tumor-infiltrating CD8$^+$ T cells, which were isolated from subcutaneous Hepa1-6 tumors treated with anti-PD1 therapy, on PRMT3 expression in cultured Hepa1-6 cells in vitro. Indeed, CM treatment significantly increased PRMT3 expression in Hepa1-6 cells (Fig. 2D, E). Also, PRMT3 mRNA and protein expression in human PLC-8024 cells was dramatically induced by CM from cultured activated human CD8$^+$ T cells (Fig. 2D, E). We then examined the effect of anti-PD1 on PRMT3

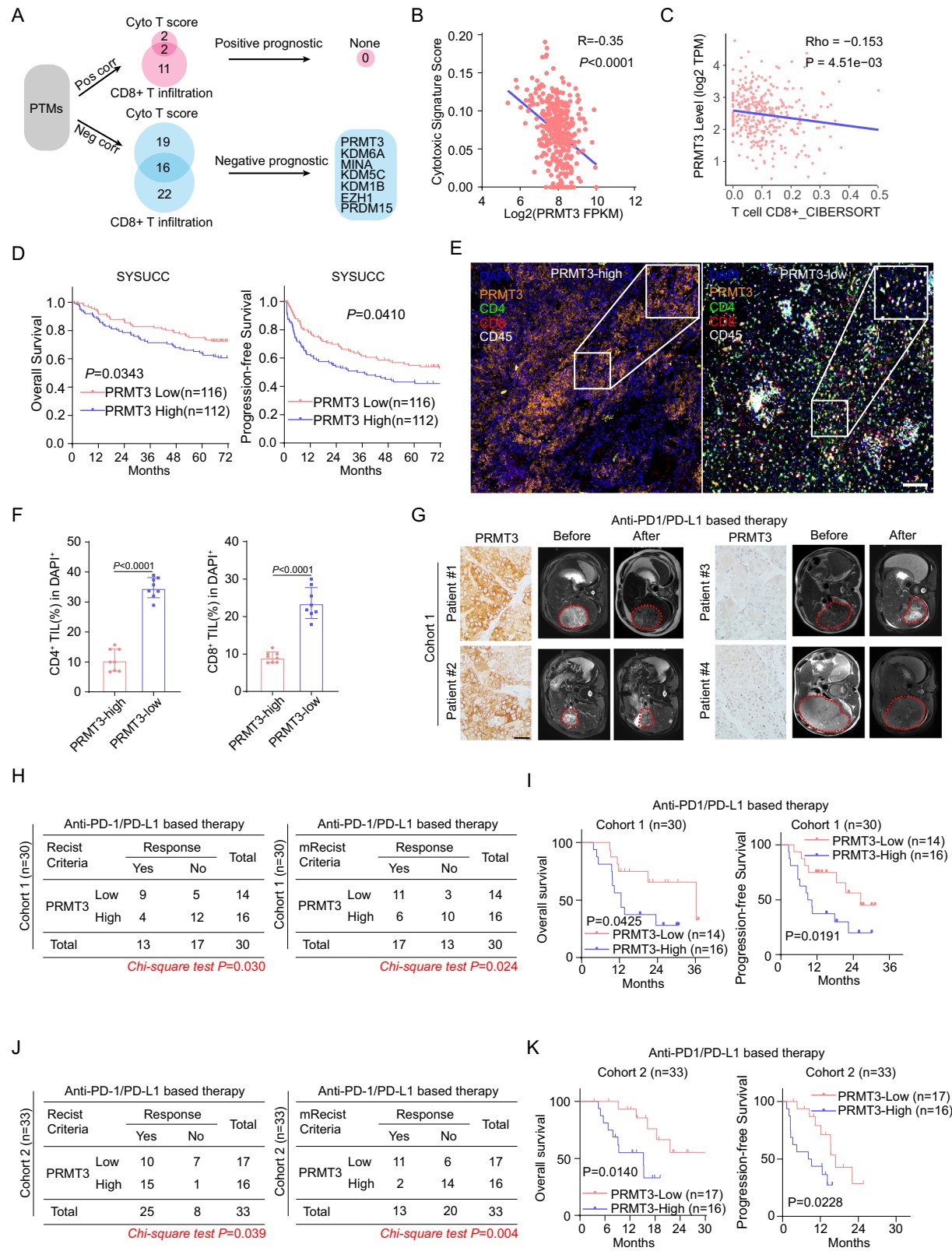

expression in *Myc/Trp53[−/−]* spontaneous tumors, an established model for HCC, which was constructed by hydrodynamic injection via the lateral tail[25]. Interestingly, in sharp contrast to the Hepa1-6 model, anti-PD1 failed to increase PRMT3 expression (Fig. 2F). We then examined whether this was due to the lack of tumor-infiltrating CD8+ T cells in the *Myc/Trp53[−/−]* tumors. Indeed, *Myc/Trp53[−/−]* tumors had very few CD8+ T cell infiltration (Supplementary Fig. 4C, D). Thus, the lack of induction

of PRMT3 expression in anti-PD1-treated *Myc/Trp53[−/−]* spontaneous tumors is likely due to the lack of sufficient activated CD8+ T cells. Collectively, our data suggest that infiltrating activated CD8+ T cells induced PRMT3 expression through paracrine signaling.

Because effector CD8+ T cells secreted IFNγ and TNF to regulate anti-tumor immunity and gene expression in tumor cells[24,26], we speculated that TNF and IFNγ secreted by activated CD8+ T cells

**Fig. 1 | High PRMT3 expression was strongly associated with poorer response to immunotherapy in HCC patients. A** Schematic diagram illustrates the workflow of immunosuppressive PTMs in the TCGA-LIHC dataset. **B** Associations of PRMT3 expression with cytotoxic T cells score evaluated by ImmuneCell AI in TCGA-LIHC dataset. **C** Associations of PRMT3 expression with CD8[+] T cell infiltration evaluated by CIBERSORT in TCGA-LIHC dataset. **D** Kaplan–Meier overall survival curves of individuals with different PRMT3 expression in the SYSUCC cohort. **E** Representative example of PRMT3-high and PRMT3-low HCC. Tumor staining by multiplexed IHC shows the spatial distributions of CD4[+] T cells and CD8[+] T cells (*n* = 8). Scale bar, 100 μm. **F** Percentage of CD4[+] T cells and CD8[+] T cells detected by multiplexed IHC in PRMT3-high and PRMT3-low HCC (*n* = 10). **G** The baseline and post-treatment MRI images of HCC patients, who had low and high PRMT3

expression, respectively, showed the patients' response to the immunotherapy. Scale bar, 50 mm. **H** The maximum response of intrahepatic target lesions in the prospective study using the RECIST and modified RECIST criteria in the PRMT3-low/high groups in immunotherapy cohort 1. **I** Kaplan–Meier overall survival and disease-free survival curves of individuals with different PRMT3 expression in the immunotherapy cohort 1. **J** The maximum response of intrahepatic target lesions in the prospective study using the RECIST and modified RECIST criteria in the PRMT3-low/high groups in immunotherapy cohort 2. **K** Kaplan–Meier overall survival and disease-free survival curves of individuals with different PRMT3 expression in the immunotherapy cohort 2. Data in (**F**) are presented as mean ± SD. Data were analyzed by a two-sided Student's *t* test in (**F**), by Chi-square test in (**H**) and (**J**). Source data are provided as a Source Data file.

may induce PRMT3 expression in HCC. To test this, we treated several HCC cell lines (PLC-8024, Hepa1-6, and H22) with TNF and IFNγ for 24 and 48 h and found that IFNγ, but not TNF, increased PRMT3 expression in a time-dependent manner (Fig. 2G, H and Supplementary Fig. 4E). However, PRMT3 expression is only slightly upregulated at 24 h but dramatically increased at 48 h after IFNγ treatment (Supplementary Fig. 4F, G). As expected, we indeed observed an upregulation of these IFNγ regulated genes at 24 h, but their expression was dampened at 48 h. These findings indicated that the upregulation of PRMT3 in response to IFNγ may suppress the expression of interferon-stimulated genes (ISG), which play a critical role in anti-tumor immunity. Consistent with this finding, we found that the effect of IFNγ on the induction of its downstream target genes was more effective in *Prmt3*-KO Hepa1-6 cells than in the control cells (Supplementary Fig. 4F). To further investigate the effect of IFNγ and TNF on PRMT3 expression, we added inhibitors of IFNγ (IFNγ antagonist 1 acetate)[27] and TNF (R-7050)[28] into CD8[+] T-cell supernatant and examined PRMT3 expression in PLC-8024 cells after incubation. PRMT3 induction was significantly attenuated by IFNγ inhibitor but not affected by TNF inhibitor (Supplementary Fig. 4H). These results suggested that IFNγ, but not TNF, secreted by CD8[+] T-cell induced PRMT3 expression in HCC cells.

Because STAT1 is a key downstream effector of IFNγ signaling that drives the activation of the IFNγ-dependent transcriptional program, we examined the effect of *Stat1* knockdown (KD) on IFNγ-induced PRMT3 expression in Hepa1-6 cells. We found that *Stat1*-KD effectively abrogated the elevation of *Prmt3* mRNA and protein expression induced by IFNγ treatment (Fig. 2I and Supplementary Fig. 4I), which indicates that PRMT3 was regulated via the IFNγ-STAT1 axis. We also examined the relationship between STAT1 expression and PRMT3 expression in clinical samples and found a positive correlation between their mRNA expression levels using the TCGA-LIHC RNA-seq dataset (Supplementary Fig. 4J). Furthermore, we examined the ENCODE chromatin immunoprecipitation sequencing (ChIP-seq) data (HepG2 cell line) and found that STAT1 directly binds to the promotor region of *PRMT3* (Fig. 2J)[29], which contains STAT1 binding motif obtained from JASPAR and FIMO database[30,31] (Supplementary Fig. 4K, L). We then performed ChIP-qPCR to examine the binding of STAT1 to *PRMT3* promoter. We found that STAT1 indeed bound to the putative promoter of *PRMT3* in PLC-8024 and Hepa1-6 HCC cells treated with IFNγ (Fig. 2K and Supplementary Fig. 4L). Since type I interferons (e.g., IFNα) also transduce signals via STAT1, we examined the effect of IFNα on PRMT3 expression in PLC-8024 and Hepa1-6 cells and found that IFNα slightly up-regulated PRMT3 (Supplementary Fig. 4M). These results suggest that PRMT3 is mainly induced by IFNγ secreted by effector CD8[+] T cell in the tumor immune microenvironment. Altogether, these results suggest that *PRMT3* is a previously unknown IFNγ-responsive gene that was up-regulated by anti-PD-1/PD-L1 therapy in HCC and may play a role in the anti-tumor immunity and the adaptive response to ICB.

## *Prmt3*-KO or PRMT3 inhibition increases immune infiltration and activates T cell-mediated anti-tumor immunity

To investigate the role of PRMT3 in regulating T cell-mediated anti-tumor immunity, we employed both genetic and pharmacological approaches to determine the effect of *Prmt3*-KO and PRMT3 inhibition on tumor progression using the Hepa1-6 syngeneic model. We generated *Prmt3*-KO Hepa1-6 cells and confirmed the complete KO of *Prmt3* in two independent clones by Western blot (Supplementary Fig. 5A). We then compared the growth of *Prmt3*-WT and the two *Prmt3*-KO clones in both immune-deficient NSG and immune-competent C57BL/6 mice using subcutaneous injection (subQ). We found that *Prmt3*-KO more profoundly delayed tumor progression in immune-competent mice than in immune-deficient mice for both clones (Fig. 3A–D and Supplementary Fig. 5B, C), suggesting that in addition to the cancer cell-intrinsic functions of PRMT3, T cell-mediated anti-tumor immunity also contributed to the observed delay in tumor growth in *Prmt3*-KO cells. Consistent with this notion, mIF and flow cytometry analysis showed that *Prmt3*-KO dramatically increased the infiltration of CD4[+] T cells and CD8[+] T cells, including the activated IFNγ[+] CD8[+] and GZMB[+] CD8[+] T cells (Fig. 3E–G and Supplementary Fig. 5D). Next, we sought to determine whether PRMT3 inhibition could elicit similar effects on tumor progression and T cell infiltration by treating Hepa1-6 tumors with SGC707, a PRMT3-specific inhibitor that was previously shown to be highly effective in delaying tumor progression in human HCC cell line models[13]. Consistent with our findings from the *Prmt3*-KO cells, SGC707 delayed tumor progression in both immune-deficient and immune-competent mice, and its effects were more profound in immune-competent mice than in immune-deficient mice (Fig. 3H–K and Supplementary Fig. 5E–G). SGC707 treatment also induced an influx of T cells, including CD4[+] T cells and CD8[+] T cells (total CD8[+], IFNγ[+] CD8[+] T cells, and GZMB[+] T cells) into the TME, as shown by mIF and Flow cytometry analyses (Fig. 3L–O). Of note, infiltrating T cells account for about 20% of total CD45[+] immune cells in the Hepa1-6 subQ tumors (Supplementary Fig. 4C, D). Since we showed that *Myc/Trp53*[-/-] spontaneous tumors had a relatively "cold" TME with fewer T cell infiltration than the Hepa1-6 model (Supplementary Fig. 4C, D), we decided to test whether PRMT3 inhibition could suppress the growth of this immunologically "cold" HCC model and reprogram the relative "cold" TME into "hot" TME by inducing T cell infiltration. Strikingly, we found that SGC707 treatment significantly reduced tumor volumes and weights in *Myc/Trp53*[-/-] mice compared to vehicle control (Fig. 3P–R). Also, immunofluorescence (IF) and flow cytometry analyses revealed that PRMT3 inhibition led to a robust increase in the abundance of tumor-infiltrating CD4[+] and CD8[+] T cells, including IFNγ[+] CD8[+] and GZMB[+] CD8[+] T cells (Fig. 3S–U).

With the observed impact of *Prmt3*-KO T cell infiltration, we decided to comprehensively characterize the impact of *Prmt3*-KO on the immune TME using single-cell RNA-seq (scRNA-seq). We sorted CD45[+] cells from *Prmt3*-KO and *Prmt3*-WT Hepa1-6 subQ tumors, which were subjected to scRNA-seq using the 10X Genomics platform. Using Seurat 4.3.0, a total of 8 tumor immune microenvironment cell clusters (B cells, T cells, macrophages, monocytes, proliferating cells, NK cells,

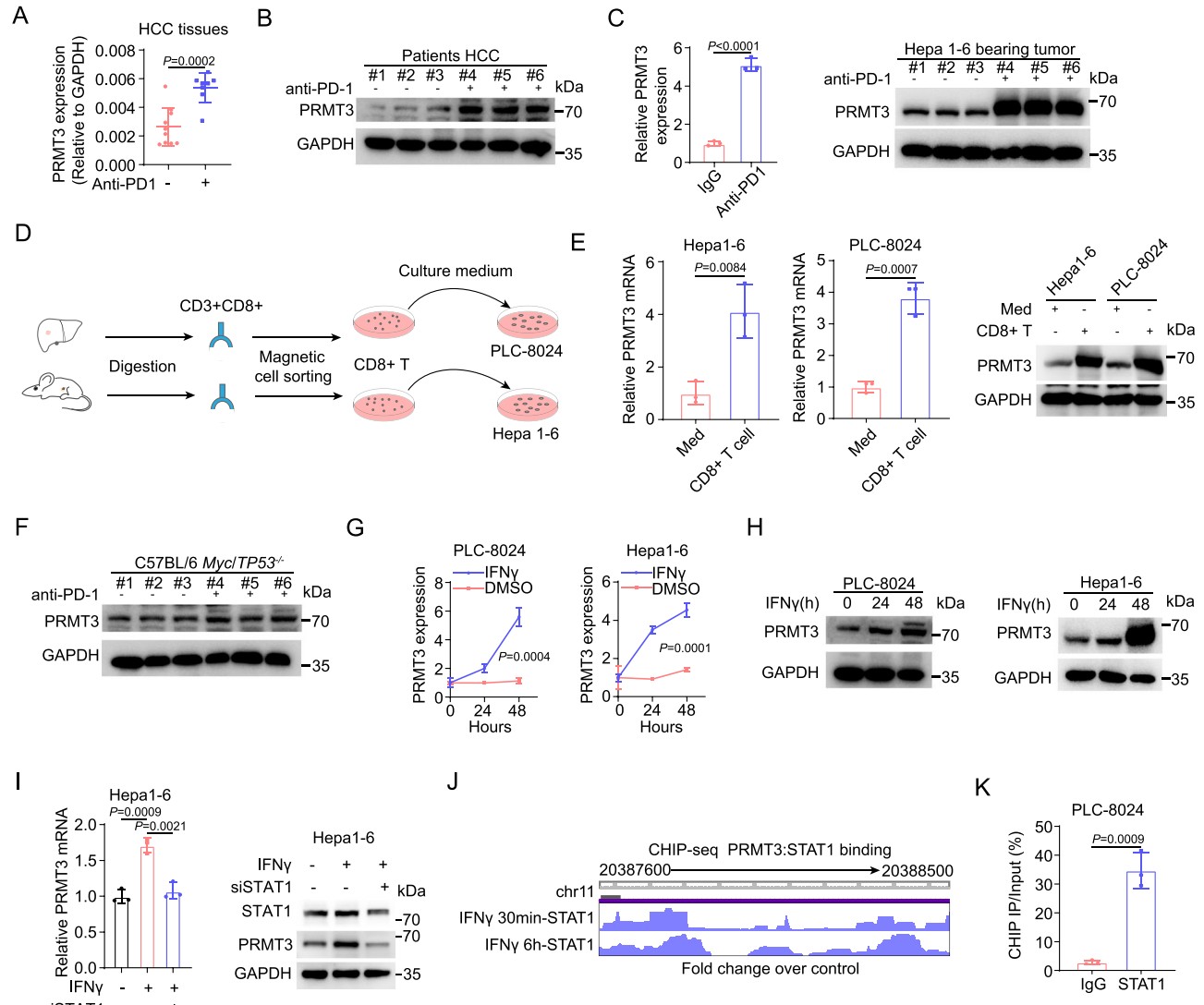

**Fig. 2 | Effector T cells induced by anti-PD1 therapy up-regulate PRMT3 expression via IFNγ-STAT1-dependent pathways. A** PRMT3 expression was detected by qPCR in HCC patients who received anti-PD1 therapy or not ($n = 10$ for none treated group and $n = 8$ for treated group). **B** PRMT3 expression was detected by western blot in HCC patients who received anti-PD1 therapy or not ($n = 3$ biologically independent experiments). **C** PRMT3 expression was detected by western blot and qPCR in Hepa1-6 subcutaneous tumors which received anti-PD1 therapy or not ($n = 3$ biologically independent samples). **D** Schematic diagram illustrates the workflow of CD8⁺ T cells sorting and HCC cells stimulating. **E** Hepa1-6 and PLC-8024 cells were treated with the conditioned medium of CD8⁺ T cells sorted from HCC tumors. PRMT3 expression was analyzed by qPCR and western blot ($n = 3$ biologically independent samples). **F** PRMT3 expression was detected by western blot in

*Myc/Trp53⁻/⁻* spontaneous model who received anti-PD1 therapy or not ($n = 3$ biologically independent samples). **G, H** Hepa1-6 and PLC-8024 cells were treated with IFNγ. PRMT3 expression was analyzed by qPCR (**G**) and western blot (**H**) ($n = 3$ biologically independent samples). **I** PRMT3 expression in *Stat1*-KD and control cells treated with IFNγ (20 ng/ml, 48 h) as shown by qRT-PCR and western blot ($n = 3$ biologically independent samples). **J** STAT1 ChIP-seq data from ENCODE shows the STAT1 binding sites at the *PRMT3* promoter region. **K** ChIP-qPCR was used to determine the binding of STAT1 to *PRMT3* promoter region in PLC-8024 cells treated with IFNγ (20 ng/ml, 48 h) ($n = 3$ biologically independent samples). Data in (**A**, **C**, **E**, **G**, **I**, **K**) are presented as mean ± SD. Data were analyzed by two-sided Student's *t* test in (**A**, **C**, **E**, **G**, **K**), by one-way ANOVA in (**I**). Source data are provided as a Source Data file.

endothelial cells, and dendritic cells) were identified based on the expression of top marker genes among the 76,937 cells (Fig. 3V). We further characterized the T cell population and identified a total of 7T cell clusters (CD4⁺, CD8⁺, NK cell, proliferation T, Treg, MAIT and other T cells) among 35,035 T cells (Supplementary Fig. 5H). Consistent with the observed increased CD4⁺ T cells and CD8⁺ T cells from IHC and flow cytometry analyses, we found that *Prmt3*-KO tumors had a significant increase in the frequency of T cell, including the CD4⁺ subset and CD8⁺ subset (Fig. 3V−X). Given the significant changes in T cell abundance by *Prmt3*-KO and PRMT3 inhibition, we then examined whether the effects of *Prmt3*-KO and PRMT3 inhibition on tumor progression were mediated by CD8⁺ or CD4⁺ T cells. We treated tumor-bearing mice with anti-CD3, anti-CD4, or anti-CD8 antibodies and

found that neutralization of CD3⁺, CD8⁺, or CD4⁺ T cells indeed attenuated the tumor growth inhibition induced by *Prmt3*-KO (Supplementary Fig. 6A−F). Of note, the effect of anti-CD8 antibodies on tumor growth was more profound than anti-CD4 antibody. Collectively, our data suggest that T cells, especially the CD8⁺ T cells, indeed mediated the effects of PRMT3 KO or PRMT3 inhibition on HCC progression.

Collectively, our data suggest that PRMT3 inhibition activated the anti-tumor immunity through increasing CD4⁺ and CD8⁺ T cell infiltration and activating CD8⁺ T cells. These findings suggest that targeting PRMT3 could suppress HCC tumor progression through cancer cell-intrinsic and cancer cell-extrinsic mechanisms (e.g., promoting T cell-mediated anti-tumor immunity).

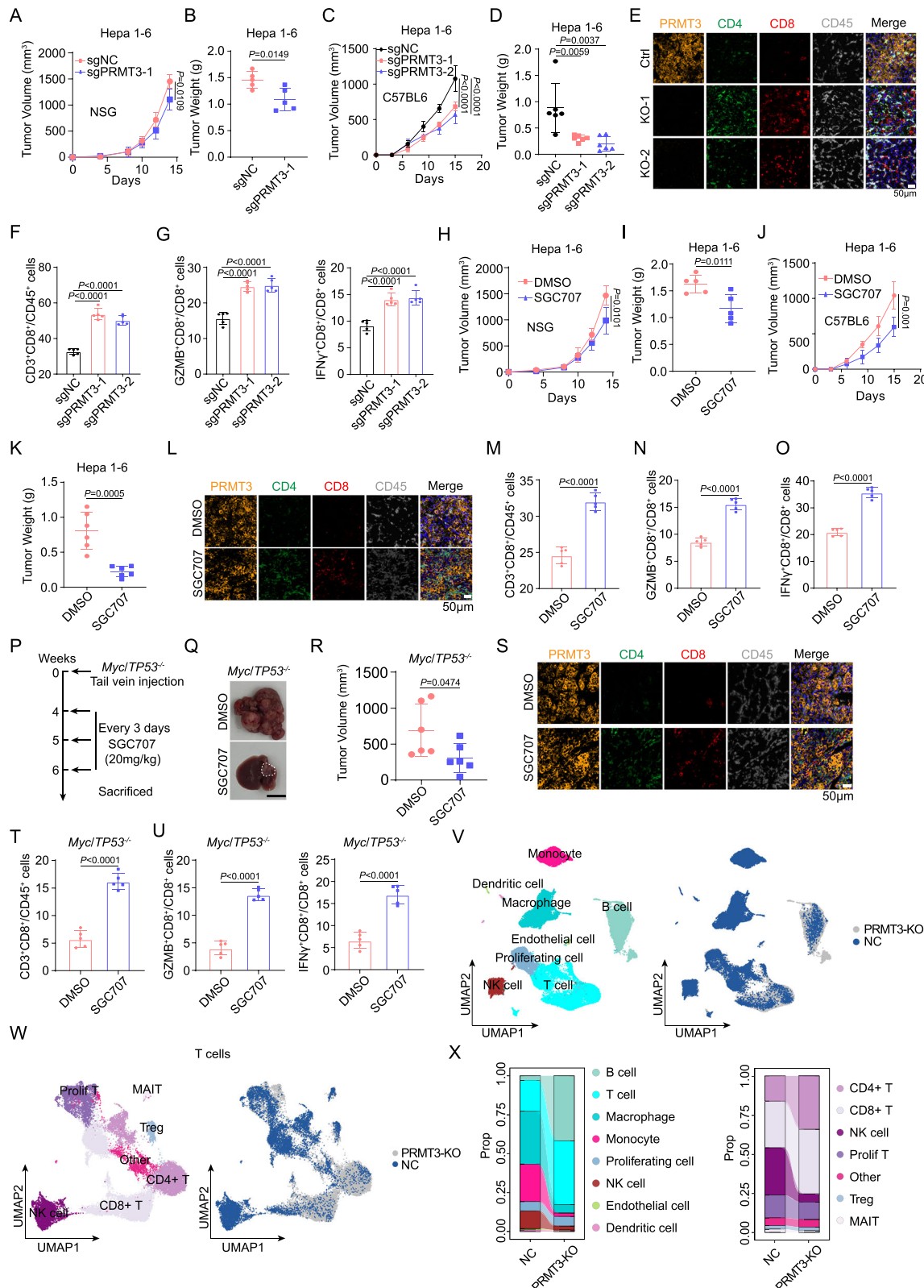

## PRMT3 methylates HSP60 at R446 and promotes its oligomerization

Since PRMT3 promotes tumor progression through the regulation of arginine methylation, we performed immunoprecipitation-mass spectrometry (IP-MS) analysis using the syngenetic Hepa1-6 cells to identify novel PRMT3 substrates that may be the key downstream effectors regulating the T cell-mediated anti-tumor immunity. As

expected, PRMT3 was efficiently pulled down in the IP experiment (Supplementary Data 1). Interestingly, we found that HSP60 (encoded by the *HSPD1* gene), a key mitochondrial chaperone protein that properly folds nascent or denatured polypeptides[32], was the top candidate in the proteins pulled down by PRMT3 (Fig. 4A, Supplementary Fig. 7A and Supplementary Data 1). We then examined publicly available protein-protein interaction data in Biogrid[33] and found that

**Fig. 3 | *Prmt3*-KO or PRMT3 inhibition increases immune infiltration and activates T cell-mediated anti-tumor immunity.** The measurement of tumor volumes (**A**) and tumor weights (**B**) of subcutaneously implanted Hepa1-6 cells (*Prmt3*-KO and *Prmt3*-WT) in NSG mice (*n* = 5). The measurement of tumor volumes (**C**) and tumor weights (**D**) of subcutaneously implanted indicated Hepa1-6 cells (*Prmt3*-KO and *Prmt3*-WT) in C57BL6 mice (*n* = 6). **E** Immunofluorescence identifying CD8⁺ and CD4⁺ T cells in subcutaneous tumors from C57BL/6 injected with the indicated Hepa1-6 cells (*Prmt3*-KO and *Prmt3*-WT) (*n* = 5). Scale bar, 50 mm. **F** Flow cytometry analysis assessing the percentage of CD8⁺ T cells from tumors in indicated groups (*Prmt3*-KO and *Prmt3*-WT) (*n* = 5). Data were analyzed by FlowJo. **G** Flow cytometry analysis assessing the percentage of T cell functional markers IFNγ and GZMB from tumors in indicated groups (*Prmt3*-KO and *Prmt3*-WT) (*n* = 5). Data were analyzed by FlowJo. The measurement of tumor volumes (**H**) and tumor weights (**I**) of subcutaneous tumors treated with SGC707 (20 mg/kg) or DMSO in NSG mice (*n* = 6). The measurement of volumes (**J**) and weights (**K**) of subcutaneous tumors treated with SGC707 (20 mg/kg) or DMSO in C57BL6 mice (*n* = 6). **L** Immunofluorescence identifying CD8⁺ and CD4⁺ T cells in subcutaneous tumors treated with SGC707 (20 mg/kg) or DMSO from C57BL/6 injected with the Hepa1-6 cells (*n* = 6). Scale bar, 50 mm. **M** Flow cytometry analysis assessing the percentage of CD8⁺ T cells from tumors in indicated groups (SGC707, 20 mg/kg or DMSO) (*n* = 5). Data were analyzed by FlowJo. Flow cytometry analysis assessing the percentage of T cell functional markers IFNγ (**N**) and GZMB (**O**) from tumors in indicated groups (SGC707,

20 mg/kg or DMSO) (*n* = 5). Data were analyzed by FlowJo. **P** Schematic diagram illustrates the workflow of *Myc/Trp53⁻/⁻* spontaneous model treated with DMSO or SGC707 (20 mg/kg) (*n* = 6). **Q** The effect of SGC707 (20 mg/kg) or DMSO on the tumor growth of *Myc/Trp53⁻/⁻* spontaneous model in C57BL/6 mice (*n* = 6). Scale bars, 1 cm. **R** The measurement of tumor volumes of *Myc/Trp53⁻/⁻* spontaneous tumor treated with SGC707 (20 mg/kg) or DMSO (*n* = 6). **S** Immunofluorescence identifying CD8⁺ and CD4⁺ T cells in *Myc/Trp53⁻/⁻* spontaneous tumor treated with SGC707 (20 mg/kg) or DMSO (*n* = 6). Scale bar, 50 mm. **T** Flow cytometry analysis assessing the percentage of CD8⁺ T cells from tumors in indicated groups (SGC707, 20 mg/kg or DMSO) (*n* = 5). Data were analyzed by FlowJo. **U** Flow cytometry analysis assessing the percentage of T cell functional markers IFNγ and GZMB from tumors in indicated groups (SGC707, 20 mg/kg or DMSO) (*n* = 5). Data were analyzed by FlowJo. **V** UMAP plot showing the components of immune cells in subcutaneous tumors of indicated groups (*Prmt3*-KO and *Prmt3*-WT). Each dot represents a single cell. The same cell type was color-coded. **W** UMAP plot showing the components of T cells in subcutaneous tumors in indicated groups (*Prmt3*-KO and *Prmt3*-WT). Each dot represents a single cell. The same cell type was color-coded. **X** The proportion of indicated immune cells in subcutaneous tumors in indicated groups (*Prmt3*-KO and *Prmt3*-WT) evaluated by scRNA-seq. Data in (**A–D, F–K, M–O, R, T, U**) are presented as mean ± SD. Data were analyzed by two-sided Student's *t* test in (**A, B, H–K, M–O, R, T, U**), by one-way ANOVA in (**C, D, F, G**). Source data are provided as a Source Data file.

HSP60 was among the PRMT3 interactors identified by multiplex CF/MS in breast cancer cells[34]. Intriguingly, about a third of PRMT3 interacting proteins in our IP-MS data were also identified as HSP60-interacting proteins in the BioGRID database[33] (Supplementary Fig. 7B). Among the top 15 potential PRMT3-interacting proteins, 6 of them were also HSP60-interating proteins, including HSPA5, HSPA8, PDIA3, P4HB, ALDOA and CCT8, suggesting a biological role for PRMT3-HSP60 interaction. Interestingly, HSP60 has been implicated in tumorigenesis as well as modulating immune responses[35]. Thus, we decided to explore whether there is an association between HSP60 expression with HCC tumor progression and immune modulation by analyzing publicly available proteomic[36] and TCGA-LIHC datasets. We found that HSP60 protein was significantly upregulated in HCC tissues compared to adjacent normal tissues (Supplementary Fig. 7C). Also, HSP60 mRNA expression was negatively correlated with CD8⁺ T cell infiltration in various datasets from Timer 2 and TIDE database[20,37] (Supplementary Fig. 7D–F). Moreover, higher HSP60 expression in HCC patients with or without immunotherapy treatment was associated with poorer prognosis (Supplementary Fig. 7G, H). Furthermore, mIF analysis showed that HSP60 expression levels were negatively correlated with the abundance of CD4⁺ and CD8⁺ T cell infiltration in HCC tumors (Supplementary Fig. 7I). Collectively, our findings suggest that HSP60 may be a key substrate of PRMT3 that plays an important role in PRMT3-mediated suppression of anti-tumor immunity.

To confirm the findings from IP-MS analysis, we performed Co-IP experiments using Hepa1-6 and PLC-8024 cells and found that HSP60 indeed interacted with PRMT3 at the endogenous level (Fig. 4B). Furthermore, IF staining showed that PRMT3 and HSP60 colocalized in the cytoplasm in PLC-8024, Hepa1-6, and *Myc/Trp53⁻/⁻* spontaneous tumor (Fig. 4C). We then examined the effect of PRMT3 perturbation on the arginine methylation of HSP60. We found that *PRMT3*-KO or PRMT3 inhibition by SGC707 abolished the asymmetric dimethylarginine (ADMA) of HSP60 in both mouse and human HCC cells (Hepa1-6 and PLC-8024) (Fig. 4D and Supplementary Fig. 7J). Also, SGC707 treatment of tumor-bearing mice significantly decreased HSP60-ADMA in *Myc/Trp53⁻/⁻* spontaneous tumor (Fig. 4E). Moreover, PRMT3 overexpression (OE) increased the ADMA of HSP60 in HepG2 and HEK293T cells (Fig. 4F). Collectively, these data demonstrated that PRMT3 directly regulates arginine methylation of HSP60.

To pinpoint the arginine residues of HSP60 that are methylated by PRMT3, we examined our mass spectrometry data and found that R446 was the only arginine methylation site (Fig. 4G and

Supplementary data 2). The HSP60 amino acid sequence surrounding R446 was evolutionarily conserved across multiple species (Fig. 4H) and matched the consensus arginine methylation motif, indicating that the methylation of this residue may have important biological significance. To determine whether PRMT3 regulates the methylation of R446, we generated R446 to lysine (K) mutant of FLAG-tagged HSP60 (R446K) (Fig. 4I). We overexpressed FLAG-HSP60 or FLAG-HSP60-R446K in HEK293T cells and found that ADMA was abolished in FLAG-HSP60-R446K compared to HSP60-WT (Fig. 4J). Since R446 is located in the equatorial domain 2 (433-573 aa), a domain reported to be essential for the oligomerization of HSP60[38], we speculated that PRMT3-mediated arginine methylation may facilitate HSP60 oligomerization. To test this, we examined the impact of *Prmt3*-KO or PRMT3 inhibition on HSP60 oligomerization using an established method[39]. Indeed, we found that *Prmt3*-KO or PRMT3 inhibition by SGC707 in PLC-8024 and Hepa1-6 cells notably decreased the oligomerization of HSP60 (Fig. 4K and Supplementary Fig. 7K). Moreover, the HSP60-R446K mutant exhibited much less oligomerization formation than HSP60-WT (Fig. 4L). We noticed that there was a lack of noticeable reduction in the HSP60 monomers by *Prmt3*-KO or PRMT3 inhibition in PLC-8024 cells, which is likely due to the very high level of HSP60 expression. However, *Prmt3*-KO indeed increased the amount of HSP60 monomers in Hepa1-6 cells, a cell line with a lower expression level of HSP60 than in PLC-8024 (Supplementary Fig. 7L). Collectively, our results indicate that PRMT3 promotes HSP60 oligomerization through methylation of HSP60 at R446.

## Inhibition of HSP60-R446 methylation increases immune infiltration and activates T cell-mediated anti-tumor immunity

Since we showed that PRMT3-KO or inhibition led to an increase in T cell infiltration, we speculated that HSP60 might be a key downstream effector of PRMT3 in immune modulation. To test this, we first knocked down HSP60 using two independent shRNAs targeting the 3' untranslated region (UTR) of HSP60 in the syngeneic Hepa1-6 cells. The HSP60-KD efficiency was confirmed by Western blot (Supplementary Fig. 8A). HSP60-KD delayed tumor progression in both immune-deficient and immune-competent mice, and its effects were more profound in immune-competent mice than in immune-deficient mice. Also, we found that HSP60-KD elicited similar effects on T cell infiltration as *PRMT3*-KO or PRMT3 inhibition (Supplementary Fig. 8B–G). HSP60-KD reduced tumor volumes and weights (Supplementary Fig. 8B–G) and increased total CD4⁺ T cells and CD8⁺ T cells, including IFNγ⁺ CD8⁺ T cells and GZMB⁺

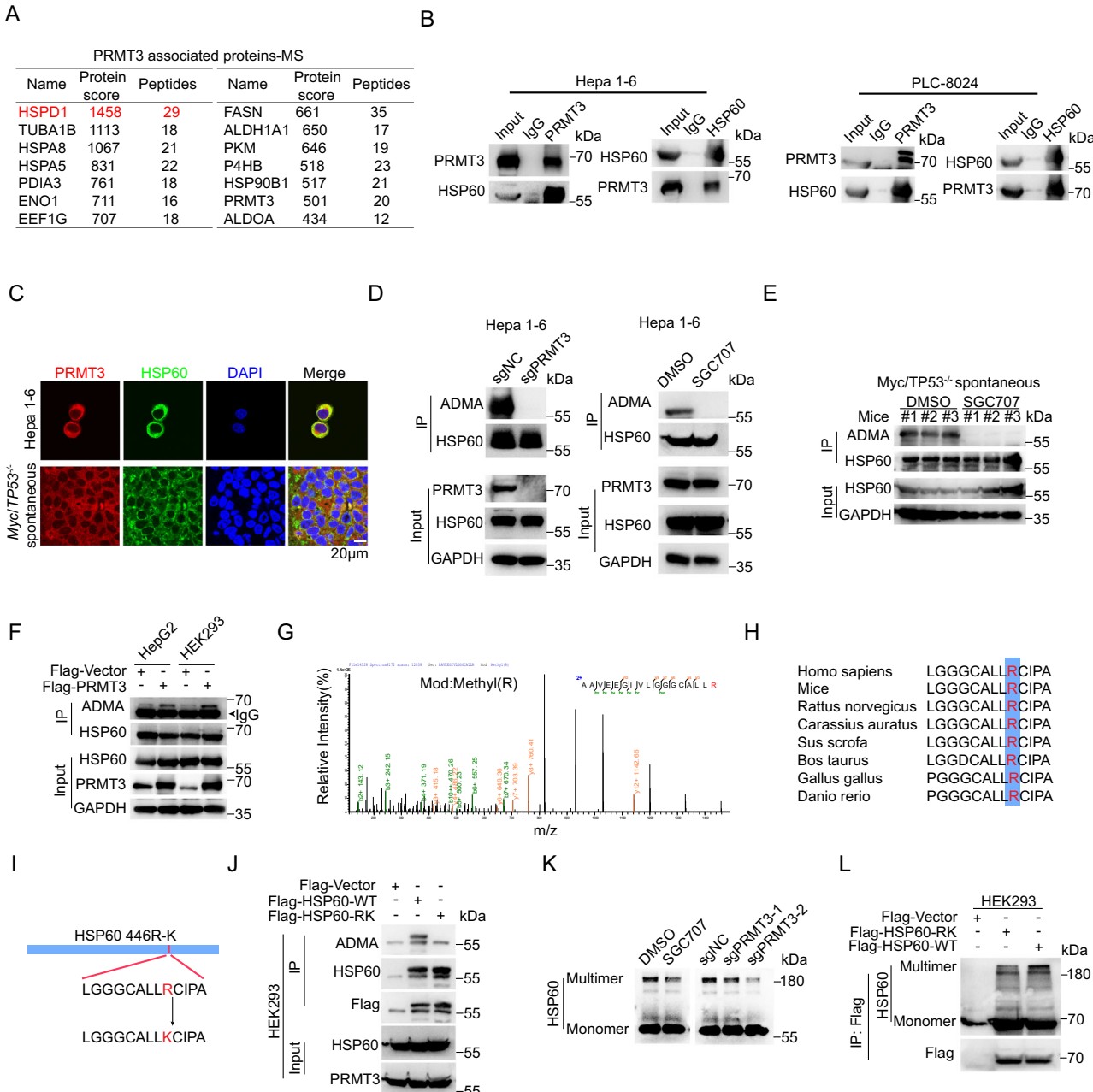

**Fig. 4 | PRMT3 methylates HSP60 at R446 and promotes its oligomerization. A** The score and number of peptide fragments (95% CI) of proteins pulled down by PRMT3 in Hepa1-6 cells were obtained from the IP-MS analysis. **B** WB analysis showed that endogenous PRMT3 and HSP60 interact with each other in PLC-8024 and Hepa1-6 cells using reciprocal co-immunoprecipitation ($n = 3$ biologically independent samples). **C** Immunofluorescence staining showed the co-localization of PRMT3 (red) and HSP60 (green) in HCC cells and $Myc/Trp53^{-/-}$ spontaneous tumor ($n = 3$ biologically independent experiments). Scale bar, 50 mm. **D** WB analysis of immunoprecipitated HSP60 to determine the effect of PRMT3-KO and SGC707 treatment on arginine methylation of HSP60 in Hepa1-6 cells using asymmetric dimethylarginine antibody (ADMA) ($n = 3$ biologically independent samples). **E** WB analysis of immunoprecipitated HSP60 to determine the effect of SGC707 treatment on arginine methylation of HSP60 in $Myc/Trp53^{-/-}$ spontaneous tumors ($n = 3$ biologically independent samples). **F** WB analysis of immunoprecipitated HSP60 to determine the effect of PRMT3-OE on arginine methylation of

HSP60 in HEK293 and HepG2 cells ($n = 3$ biologically independent samples). **G** Fragmentation spectrum of the methylated peptide identified by liquid chromatography/tandem mass spectrometry (LC-MS/MS). **H** The sequences surrounding R446 of HSP60 are evolutionarily conserved across multiple species. **I** A scheme showing the sequences of Flag-tagged HSP60-WT and HSP60-R446K mutant. **J** WB analysis of immunoprecipitated Flag-tagged HSP60-WT and HSP60-R446K mutant showed that R446K mutation dramatically reduced ADMA signal in HEK293 cells overexpressing HSP60-R446K mutant ($n = 3$ biologically independent samples). **K** WB analysis determined the effect of SGC707 treatment and PRMT3-KO on the oligomerization of HSP60 in PLC-8024 cells ($n = 3$ biologically independent samples). **L** WB analysis of immunoprecipitated Flag-tagged HSP60-WT and HSP60-R446K mutant showed that R446K mutation dramatically reduced oligomerization of HSP60 in HEK293 cells overexpressing HSP60-R446K mutant ($n = 3$ biologically independent samples). Source data are provided as a Source Data file.

CD8[+] T cells (Supplementary Fig. 8H−K). We then examined whether R446 methylation is required for HSP60 function in immune modulation by comparing the effect of HSP60-WT overexpression and HSP60-R446K mutant overexpression in HSP60-KD cells on tumor

progression and T cell infiltration. Western blot analysis showed that HSP60-WT and HSP60-R446K mutant expressed at comparable levels (Fig. 5A). We found that the expression of HSP60-WT almost fully reversed the reduction in tumor volumes and weights caused

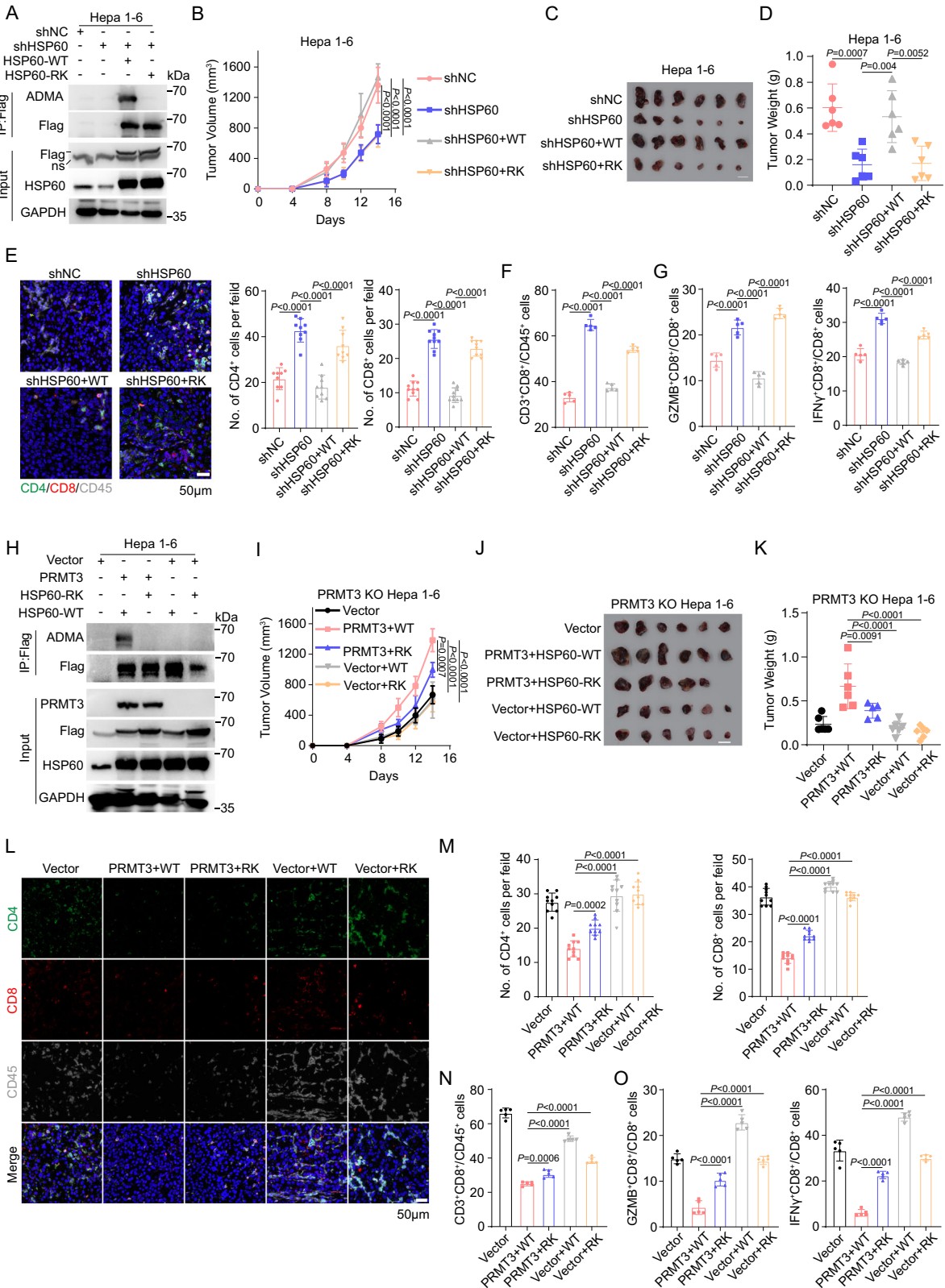

by HSP60-KD compared to the control cells (Fig. 5B–D). Moreover, the overexpression of HSP60-WT in HSP60-KD cells decreased the infiltration of CD4⁺ T cells and CD8⁺ T cells, including IFNγ⁺ CD8⁺ T cells and GZMB⁺ CD8⁺ T cells (Fig. 5E–G). In contrast, the overexpression of HSP60-R446K in HSP60-KD cells did not have any impact on the tumor growth and T cell infiltration in the HSP60-KD cells (Fig. 5B–G). Thus, our data suggest that arginine methylation

of HSP60 at R446 is required for its tumor-promoting and immune modulation functions.

To determine whether HSP60 function depends on PRMT3-mediated arginine methylation, we examined the effect of over-expressing HSP60-R446K mutant and HSP60-WT on tumor progression and T cell infiltration in the presence or absence of PRMT3 overexpression in *Prmt3*-KO cells. WB analysis showed that the ADMA

**Fig. 5 | Inhibition of HSP60-R446 methylation increases immune infiltration and activates T cell-mediated anti-tumor immunity. A** Flag-tagged HSP60-WT and HSP60-R446K were overexpressed in HSP60-KD Hepa1-6 cells ($n = 3$ biologically independent samples). **B** The measurement of tumor volumes to determine the effect of Flag-tagged HSP60-WT and HSP60-R446K mutant OE on the growth of HSP60-KD cells ($n = 6$). **C** The effect of Flag-tagged HSP60-WT and HSP60-R446K mutant OE on the growth of HSP60-KD cells, which were subcutaneously implanted in C57BL/6 mice ($n = 6$). Scale bars, 1 cm. **D** The tumor weights of HSP60-KD cells which co-expressed Flag-tagged HSP60-WT and HSP60-R446K at the endpoint of the experiment (Day 14) ($n = 6$). **E** Immunofluorescence identifying CD4$^+$ and CD8$^+$ T cells in subcutaneous tumors from C57BL6 injected with the indicated Hepa1-6 cells (shNC, shHSP60, shHSP60 + WT and shHSP60 + RK) ($n = 9$ for each group). **F** Flow cytometry analysis assessing the percentage of CD8$^+$ T cells from tumors in indicated groups (shNC, shHSP60, shHSP60 + WT, and shHSP60 + RK) ($n = 5$). Data were analyzed by FlowJo. **G** Flow cytometry analysis assessing the percentage of T cell functional markers IFNγ and GZMB from tumors in indicated groups (shNC, shHSP60, shHSP60 + WT, and shHSP60 + RK) ($n = 5$). Data were analyzed by FlowJo. **H** Flag-tagged HSP60-WT and HSP60-R446K were overexpressed in PRMT3-KO Hepa1-6 cells and PRMT3-KO Hepa1-6 cells with PRMT3-OE ($n = 3$ biologically independent samples). **I** The measurement of tumor volumes to determine the effect of Flag-tagged HSP60-WT and HSP60-R446K mutant OE on the growth of PRMT3-KO Hepa1-6 cells and PRMT3-KO Hepa1-6 cells with PRMT3-OE, which were subcutaneously implanted in C57BL/6 mice ($n = 6$). **J** The Flag-tagged HSP60-WT and HSP60-R446K OE on the growth of PRMT3-KO Hepa1-6 cells and PRMT3-KO Hepa1-6 cells with PRMT3-OE, which were subcutaneously implanted in C57BL/6 mice ($n = 6$). Scale bars, 1 cm. **K** The tumor weights of PRMT3-KO Hepa1-6 cells and PRMT3-KO Hepa1-6 cells with PRMT3 OE which co-expressed HSP60-WT and HSP60-R446K at the endpoint of the experiment (Day 14) ($n = 6$). **L** Immunofluorescence identifying CD4$^+$ and CD8$^+$ T cells in subcutaneous tumors from BALB/c injected with the indicated Hepa1-6 cells (Vector, PRMT3 + WT, PRMT3 + RK, Vector + WT, and Vector + RK). **M** Quantitative analysis of CD4$^+$ and CD8$^+$ T cells per field identified by immunofluorescence in indicated groups (Vector, PRMT3 + WT, PRMT3 + RK, Vector + WT, and Vector + RK) ($n = 10$ for each group). **N** Flow cytometry analysis assessing the percentage of CD8$^+$ T cells from tumors in indicated groups (Vector, PRMT3 + WT, PRMT3 + RK, Vector + WT, and Vector + RK) ($n = 5$). Data were analyzed by FlowJo. **O** Flow cytometry analysis assessing the percentage of T cell functional markers IFNγ and GZMB from tumors in indicated groups (Vector, PRMT3 + WT, PRMT3 + RK, Vector + WT, and Vector + RK) ($n = 5$). Data were analyzed by FlowJo. Data in (**B, D, E–G, I, K, M–O**) are presented as mean ± SD. Data were analyzed by one-way ANOVA in (**B, D, E–G, I, K, M–O**). Source data are provided as a Source Data file.

was abolished in these cells with comparable expression of PRMT3 and HSP60 (Fig. 5H). We found that HSP60-WT OE or R446K OE in the absence of PRMT3 overexpression did not promote tumor progression in *Prmt3*-KO Hepa1-6 cells compared to vector control, as reflected by the similar tumor volumes and tumor weights (Fig. 5I–K) and comparable T cell infiltration (Fig. 5L–O). In contrast, the co-expression of PRMT3 with HSP60-WT in *PRMT3*-KO cells dramatically promoted tumor progression and suppressed T cell infiltration (Fig. 5L–O). Also, co-expression of PRMT3 with HSP60-R446K mutant had a less profound effect on tumor progression and immune infiltration inhibition than co-expression of PRMT3 with HSP60-R446K (Fig. 5L–O). Interestingly, PRMT3-OE led to a small but significant increase in the tumor volumes and weights and a decrease in T cell infiltration (CD4$^+$, CD8$^+$, and IFNγ$^+$ and GZMB$^+$ CD8$^+$ T cells) in HSP60-R446K OE cells compared to vector control (Fig. 5I–O). Collectively, our data suggest that PRMT3-mediated arginine methylation of HSP60 at R446 plays a crucial role in HCC tumor progression and the suppression of T cell infiltration.

**Inhibition of PRMT3-mediated arginine methylation of HSP60 induces mtDNA leakage and cGAS/STING signaling activation**

Since HSP60 methylation by PRMT3 promotes its oligomerization which was essential for proper protein folding in the mitochondria, a key organelle that has been increasingly appreciated as a key regulator of immune response[40], we speculated that PRMT3-mediated arginine methylation of HSP60 is critical for maintaining mitochondria homeostasis. Consequently, blocking PRMT3-mediated HSP60 methylation may lead to defective HSP60 function, resulting in the disruption of mitochondria homeostasis and the subsequent activation of mitochondrial unfolded protein response (UPR). To test this, we first examined the effect of *PRMT3*-KO and PRMT3 inhibition on mitochondrial reactive oxygen species (mtROS), a marker for mitochondrial stress, and mitochondrial potential, an indicator for mitochondria homeostasis. We found that both *PRMT3*-KO and PRMT3 inhibition significantly increased in mtROS, as measured by mitoSOX, and decreased membrane potential (m$\Delta\varphi$), as measured by JC-1 staining (Fig. 6A, B and Supplementary Fig. 9A, B). To determine whether R446 methylation is crucial for maintaining mitochondrial integrity, we examined the effect of HSP60-WT OE and HSP60-R446K OE on mtROS and mitochondrial potential. We found that the HSP60-WT OE, but not HSP60-R446K OE, almost fully reversed the effect of HSP60-KD on mitoROS and membrane potential (Fig. 6A, B and Supplementary Fig. 9A, B). Since impairment of mitochondrial integrity leads to

mtDNA leakage, the defect in HSP60 of PRMT3-mediated methylation may increase the membrane permeability of mitochondria. To test this, we examined whether *PRMT3*-KO or PRMT3 inhibition induced mtDNA leakage. We found that *Prmt3*-KO or PRMT3 inhibition led to an increase in mtDNA leakage as shown by IF staining with mitotracker and TFAM, a mtDNA-specific protein (Fig. 6C and Supplementary Fig. 9C). Similarly, HSP60-KD also led to an increase in cytosolic mtDNA leakage, which was reversed by the overexpression of HSP60-WT but not HSP60-R446K mutant (Fig. 6D). To establish the significance of PRMT3-dependent methylation of HSP60 on mitochondrial DNA leakage, we examined the effect of co-expression of PRMT3 with HSP60-WT and R446K mutant in *Prmt3*-KO Hepa1-6 cells. We found that the co-expression of PRMT3 with HSP60-WT but not with HSP60-R446K mutant in *PRMT3*-KO cells effectively prevented mtDNA leakage (Fig. 6E). Thus, our data suggest that HSP60-R446 methylation by PRMT3 was essential for inhibiting mtDNA leakage.

Since mtDNA leakage activates cGAS/STING signaling through binding to cGAS and cGAMP was the direct readout of cGAS activation, we detected the cGAMP production and found that PRMT3-KO or PRMT3 inhibition increased cGAMP production in Hepa1-6 and PLC-8024 cells (Supplementary Fig. 9D–F). Similarly, HSP60-KD led to increased cGAMP production, which was reversed by overexpression of HSP60-WT but not HSP60-R446K (Supplementary Fig. 9G). Moreover, HSP60-WT or mutant OE in PRMT3-KO cells led to a further increase in cGAMP level, which was significantly reduced when PRMT3 was overexpressed (Supplementary Fig. 9H). We then examined whether perturbation of PRMT3-mediated HSP60 arginine methylation leads to activation of cGAS/STING signaling. We found that *PRMT3*-KO or PRMT3 inhibition increased phosphorylation of TBK1, IRF3, and STING, key effectors in the cGAS/STING pathway, in Hepa1-6 cells and PLC-8024 cells (Fig. 6F and Supplementary Fig. 9I). Importantly, PRMT3 inhibitor treatment of *Myc/Trp53$^{-/-}$* mice also led to a drastic increase of phosphorylation of TBK1, IRF3, and STING in the spontaneous HCC tumor samples, accompanied by a dramatic decrease in ADMA that confirmed the effectiveness of PRMT3 inhibition in vivo (Fig. 6G). Similarly, HSP60-KD led to activation of cGAS/STING signaling as shown by increased phosphorylation of TBK1, IRF3, and STING (Fig. 6H, I), which was reversed by overexpression of HSP60-WT but not HSP60-R446K mutant. These findings demonstrated that blocking PRMT3-mediated HSP60 arginine methylation activates the cGAS/STING pathway.

Since activation cGAS/STING signaling effectively up-regulated type I interferon and inflammatory cytokine production[41], we analyzed

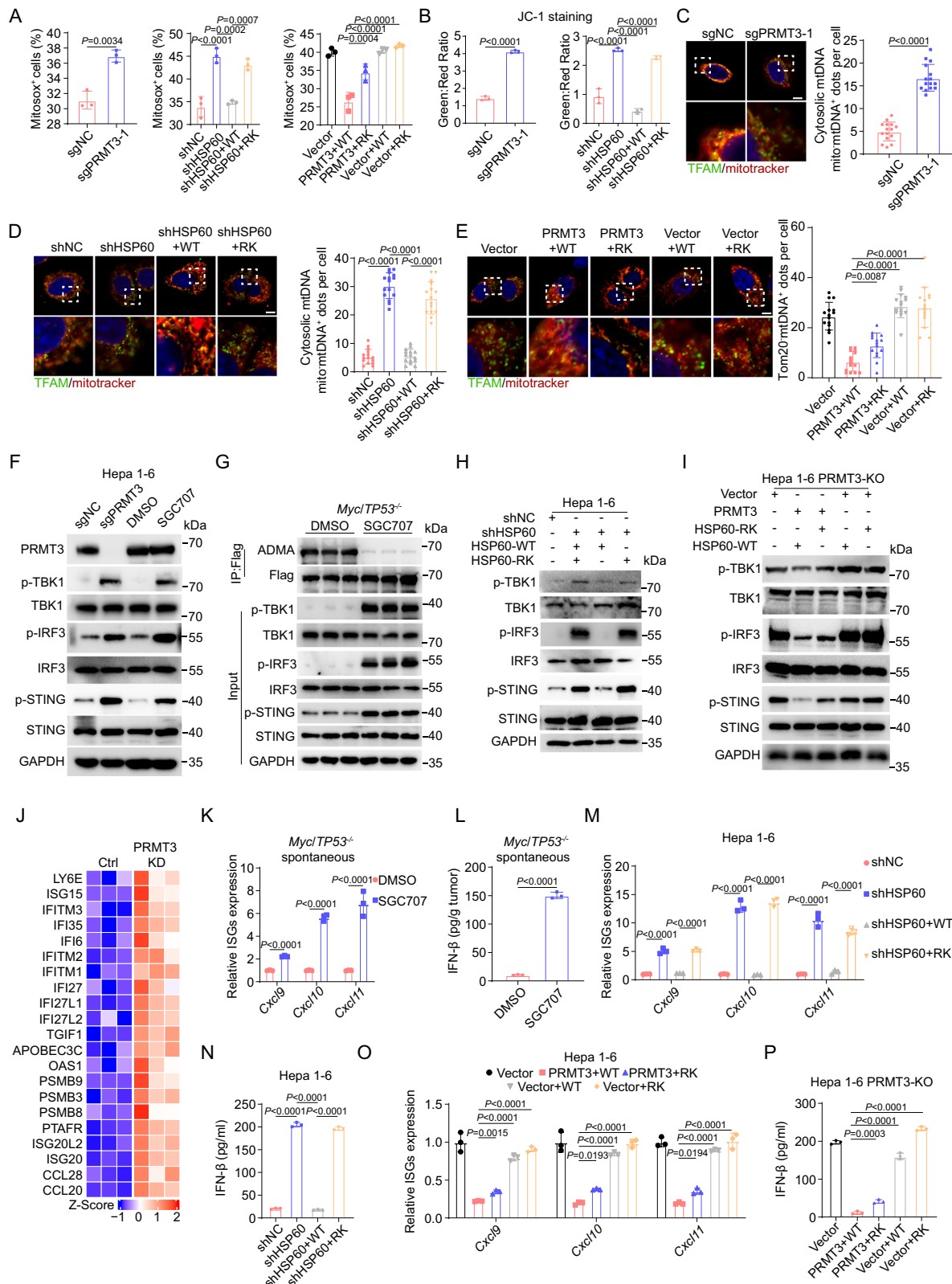

our previously published RNA-seq data[13] and found that *PRMT3*-KD increased the expression of several ISGs in human PLC-8024 cells (Fig. 6J). We also observed an increase in interferon-stimulated genes (ISG) (e.g., *Cxcl9*, *Cxcl10*, *Cxcl11*) and IFNβ production in tumors from *Myc/Trp53⁻/⁻* mice treated with PRMT3 inhibitor, an in *PRMT3*-KO mouse and human HCC cells (Hepa1-6 and PLC-8024) (Fig. 6K, L and Supplementary Fig. 9J–L). Similarly, HSP60-KD led to activation of the

expression of ISGs and increased IFNβ production, which was reversed by HSP60-WT OE but not by HSP60-R446K OE (Fig. 6M, N) Importantly, HSP60-WT or mutant OE in *PRMT3*-KO cells led to further activation of ISGs, which can dramatically suppress when PRMT3 was overexpressed (Fig. 6O, P), suggesting that PRMT3 expression and HSP60-R446 methylation were required for inhibiting cGAS/STING pathway. Thus, our data suggest that PRMT3 overexpression

**Fig. 6 | Inhibition of PRMT3-mediated arginine methylation of HSP60 induces mtDNA leakage and cGAS/STING signaling activation. A** MitoSOX⁺ cells were detected by flow cytometry in indicated groups (*n* = 3 biologically independent samples). **B** JC-1 aggregates: JC-1 monomers ratio in indicated groups (*n* = 3 biologically independent samples). **C** The effect of PRMT3-KO on mtDNA release was assessed by IF staining with a dsDNA-specific antibody. Representative images (scale bar, 5 μm) and quantitative results are shown (*n* = 15). **D** The effect of Flag-tagged HSP60-WT and HSP60-R446K mutant OE on mtDNA release in HSP60-KD cells was assessed by IF staining with mitotracker and TFAM. Representative images (scale bar, 5 μm) and quantitative results are shown (*n* = 15). **E** The effect of Flag-tagged HSP60-WT and HSP60-R446K mutant OE on mtDNA release in PRMT3-KO Hepa1-6 cells and PRMT3-KO Hepa1-6 cells with PRMT3-OE was assessed by IF staining with mitotracker and TFAM. Representative images (scale bar, 5 μm) and quantitative results are shown (*n* = 15). **F** The effects of PRMT3-KO or SGC707 treatment on the expression of TBK1/p-TBK1, IRF3/p-IRF3 and STING/p-STING in Hepa1-6 cells (*n* = 3 biologically independent samples). **G** The effects of SGC707 treatment on the expression of TBK1/p-TBK1, IRF3/p-IRF3 and STING/p-STING in *Myc/TrpS3*⁻/⁻ spontaneous tumors (*n* = 3 biologically independent samples). **H** The effect of Flag-tagged HSP60-WT and HSP60-R446K mutant OE on the expression of TBK1/p-TBK1, IRF3/p-IRF3 and STING/p-STING in HSP60-KD cells (*n* = 3 biologically independent samples). **I** The effect of Flag-tagged HSP60-WT and HSP60-R446K mutant OE on the expression of TBK1/p-TBK1, IRF3/p-IRF3 and STING/p-STING in PRMT3-KO Hepa1-6 cells and PRMT3-KO Hepa1-6 cells with PRMT3 OE (*n* = 3 biologically independent samples). **J** Differentially expressed interferon-stimulated genes (ISGs) between PLC-8024 cells with and without knockdown of PRMT3 were identified by RNA-seq. **K** The effects of SGC707 treatment on the expression of ISGs in *Myc/TrpS3*⁻/⁻ spontaneous tumors (*n* = 3 biologically independent samples). **L** The effects of SGC707 treatment on IFNβ production in *Myc/TrpS3*⁻/⁻ spontaneous tumors as detected by ELISA assay (*n* = 3 biologically independent samples). **M** The effect of Flag-tagged HSP60-WT and HSP60-R446K mutant OE on the expression of ISGs in HSP60-KD cells (*n* = 3 biologically independent samples). **N** The effect of Flag-tagged HSP60-WT and HSP60-R446K mutant OE on IFNβ production in HSP60-KD cells as detected by ELISA assay (*n* = 3 biologically independent samples). **O** The effect of Flag-tagged HSP60-WT and HSP60-R446K mutant OE on the expression of ISGs in PRMT3-KO Hepa1-6 cells and PRMT3-KO Hepa1-6 cells with PRMT3-OE (*n* = 3 biologically independent samples). **P** The effect of Flag-tagged HSP60-WT and HSP60-R446K mutant OE on IFNβ production in PRMT3-KO Hepa1-6 cells and PRMT3-KO Hepa1-6 cells with PRMT3-OE as detected by ELISA assay (*n* = 3 biologically independent samples). Data in (**A–E**, **K–P**) are presented as mean ± SD. Data were analyzed by two-sided Student's *t* test in (**A**) (left), (**B**) (left), (**C**, **K**, **L**), by one-way ANOVA in (**A**) (middle, right), (**B**) (right), (**D**, **E**, **M–P**). Source data are provided as a Source Data file.

suppresses cGAS/STING activation and the activation of type I interferon response in HCC. To explore the clinical relevance of these findings, we analyzed the TCGA-LIHC data for possible association of PRMT3 expression with ISG expression. We found that PRMT3 was negatively correlated with the expression of several ISGs (e.g., *IFITM1, IFITM3, ISG15, IFI27, OAS1, IFI27L1, IFI35, IFITM2, ISG20, LY6E, PSMB3, PSMB8, PSMB9, CCL5, CXCL10,* and *IFI6*) in TCGA-LIHC dataset (Supplementary Fig. 9M). This observation is consistent with our findings that prolonged IFNγ treatment induces PRMT3 expression, which in turn suppresses the expression of ISGs (Supplementary Fig. 4F). Collectively, our data suggest that PRMT3-mediated methylation of HSP60 at R446 prevents the activation of anti-tumor immunity through the maintenance of mitochondrial integrity and inhibition of cGAS/ STING activation HCC in vitro and in vivo.

To examine the role of HSP60 oligomerization in cGAS/STING signaling and anti-tumor immunity, we aimed to determine whether the HSP60 mutant with defective oligomerization exhibits phenotypes similar to those observed in the methylation-defective mutant. We examined whether HSP60-D3G[42], a mutant with defective oligomerization, was involved in immune modulation by comparing the effect of HSP60-D3G overexpression and HSP60-R446K mutant overexpression in HSP60-KD cells on mtROS, mtDNA release, cGAMP level, IFNβ targets, and IFNβ production. We found that both HSP60-D3G and HSP60-R446K overexpression could not reverse the effect of HSP60 KD on mitoROS and membrane potential (Supplementary Fig. 10A, B). Similarly, both HSP60-D3G and HSP60-R446K overexpression could not attenuate cytosolic mtDNA leakage and decrease the cGAMP level induced by HSP60-KD in HCC cells (Supplementary Fig. 10C, D). Also, similarly, HSP60 KD led to activation of the expression of ISGs and increased IFNβ production, which could not be reversed by either HSP60-D3G or HSP60-R446K overexpression (Supplementary Fig. 10E, F). Importantly, we found that HSP60-D3G and HSP60-R446K overexpression had comparable effects on mtROS, membrane potential, mtDNA leakage, cGAMP level, ISGs expression, and IFNβ production (Supplementary Fig. 10). Collectively, our data suggest that PRMT3 inhibition activated anti-tumor immunity by impairing HSP60 oligomerization in HCC.

### The anti-tumor immunity induced by *Prmt3* KO or inhibition was dependent on the activation of cGAS/STING signaling

Since PRMT3 inhibition activated cGAS/STING signaling through promoting mtDNA release, we decided to examine whether cGAS/STING hyperactivation directly contributes to the delayed tumor progression and increased T cell infiltration observed in *Prmt3*-KO or PRMT3 inhibitor-treated tumors. We found that *STING*-KD markedly reversed the growth inhibition induced by *Prmt3* KO in Hepa1-6 cells in immune-competent mice (Fig. 7A, B and Supplementary Fig. 11A, B). Also, *STING* KD significantly decreased the infiltration of CD4⁺ T cells and CD8⁺ T cells, including IFNγ⁺ CD8⁺ T cells, GZMB⁺ CD8⁺ T cells in *Prmt3*-KO tumors (Fig. 7C, D). Furthermore, *STING* KD notably inhibited the expression of several ISGs and IFNβ production induced by *Prmt3* KO (Fig. 7E, F). To determine whether cGAS inhibition could reverse the effect of PRMT3 inhibitor on tumor progression and T cell infiltration, we treated the Myc/*Trp53*⁻/⁻ spontaneous model with a well-characterized cGAS inhibitor, RU.521, and found that RU.521 combined with SGC707 markedly reduced the anti-tumor activities of SGC707, led to bigger tumor volumes and weights (Fig. 7G, H) and a significant reduction in T cell infiltration (CD4⁺ T cells, CD8⁺ T cells, IFNγ⁺ CD8⁺ T cells, and GZMB⁺ CD8⁺ T cells) compared to the SGC707-treated group (Fig. 7I–L). Moreover, RU.521 treatment notably inhibited the expression of ISGs and IFNβ production induced by PRMT3 inhibition (Fig. 7M, N). To further examine whether cGAS hyperactivation directly contributes to the delayed tumor progression and increased T cell infiltration observed in *Prmt3*-KO tumors, we knocked down cGAS in *Prmt3*-KO Hepa1-6 cells. We found that *cGAS* KD markedly reversed the growth inhibition induced by *Prmt3* KO in Hepa1-6 tumors (Fig. 7O, P and Supplementary Fig. 11C, D). Also, *cGAS* KD significantly decreased the infiltration of CD4⁺ T cells and CD8⁺ T cells, including IFNγ⁺ CD8⁺ T cells, GZMB⁺ CD8⁺ T cells in *Prmt3*-KO tumors (Fig. 7Q, R). Furthermore, *cGAS* KD notably inhibited the expression of several ISGs and IFNβ production induced by *Prmt3* KO (Fig. 7S, T). Collectively, these results strongly suggest that the activation of cGAS/STING signaling directly mediated the anti-tumor effects of *PRMT3* KO or inhibition through activation of T cell-mediated anti-tumor immunity in HCC.

### *Prmt3* KO or inhibition enhances immunotherapy response in HCC

Due to the limited efficacy of immunotherapy in HCC, we decided to determine whether *Prmt3* KO or PRMT3 inhibition improves the response of HCC to ICB. We found that *Prmt3* KO dramatically improved the response of tumor-bearing mice to anti-PD1 therapy using the Hepa1-6 model that displays a modest response to anti-PD1 therapy (Fig. 8A, B and Supplementary Fig. 12A, B). Also, anti-PD1 therapy further increased the infiltration of CD4⁺ T cells and CD8⁺ T cells, including IFNγ⁺ CD8⁺ T cells and GZMB⁺ CD8⁺ T cells, in *Prmt3*-

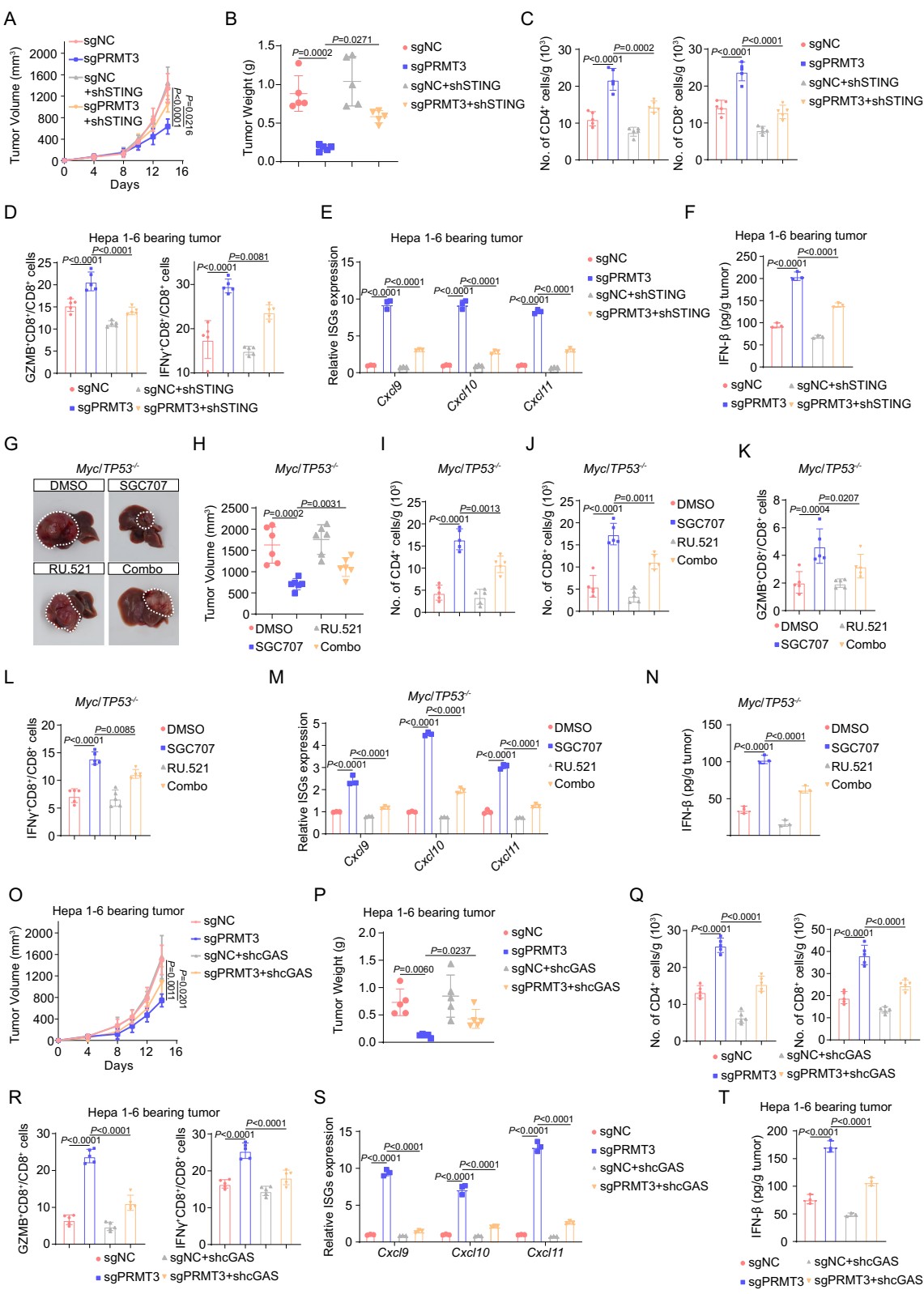

KO tumors compared to IgG control. In contrast, anti-PD1 treatment had little effect on the infiltration of T cells in *Prmt3*-WT tumors (Fig. 8C–F). Moreover, anti-PD1 treatment dramatically increased the expression of *Ifnb1* and *Tnf*, markers for T cell activation compared to IgG treatment in *Prmt3*-KO tumors (Fig. 8G). Furthermore, comparable results were observed for PRMT3 inhibitor treatment in the Hepa1-6 model. SGC707 also dramatically sensitized Hepa1-6 tumors to anti-

PD1 therapy compared to the anti-PD1 therapy group and DMSO-treated group (Fig. 8H, I and Supplementary Fig. 12C, D) and potentiate the effect of anti-PD1 on increasing T cell infiltration and activation (Fig. 8J–M). Importantly, we found that SGC707 treatment dramatically enhanced the efficacy of anti-PD1 therapy in the *Myc/Trp53*[−/−] spontaneous HCC model, which displays de novo resistance to anti-PD1 therapy, as reflected by a significant reduction in tumor volume

**Fig. 7 | The anti-tumor immunity induced by *Prmt3*-KO or inhibition was dependent on the activation of cGAS/STING signaling. A** The effect of STING-KD on the tumor growth of PRMT3-KO Hepa1-6 cells as shown by measurement of tumor volumes ($n = 6$). **B** The effect of STING-KD on the tumor growth of PRMT3-KO Hepa1-6 cells as shown by tumor weights at the endpoint of the experiment on day 14 ($n = 6$). **C** CD4$^+$ and CD8$^+$ T cells from the treated tumors were analyzed by flow cytometry for the indicated groups (sgNC, sgPRMT3, sgNC + shSTING, and sgPRMT3 + shSTING) ($n = 5$). Data were analyzed by FlowJo. **D** Flow cytometry analysis assessing the percentage of T cell functional markers IFNγ and GZMB from tumors in indicated groups. (sgNC, sgPRMT3, sgNC + shSTING and sgPRMT3 + shSTING) ($n = 5$). Data were analyzed by FlowJo. **E** The effects of STING-KD on the expression of ISGs in tumors from PRMT3-KO Hepa1-6 cells ($n = 3$ biologically independent samples). **F** The effects of STING-KD on the IFNβ production in tumors from PRMT3-KO Hepa1-6 cells as detected by ELISA assay ($n = 3$ biologically independent samples). **G**, **H** The effects of SGC707/RU.521 treatment on tumor volumes of *Myc/Trp53$^{-/-}$* spontaneous model ($n = 6$). CD4$^+$ (**I**) and CD8$^+$ (**J**) T cells from tumors were analyzed by flow cytometry in indicated groups (DMSO, SGC707, RU.521, and Combo) ($n = 5$). Data were analyzed by FlowJo. Flow cytometry analysis assessing the percentage of T cell functional markers IFNγ (**K**) and GZMB (**L**) from tumors in indicated groups (DMSO, SGC707, RU.521, and Combo) ($n = 5$). Data were analyzed by FlowJo. **M** The effects of SGC707/RU.521 treatment on the expression of ISGs in *Myc/Trp53$^{-/-}$* spontaneous model as shown by qRT-PCR ($n = 3$ biologically independent samples). **N** The effects of SGC707/RU.521 treatment on IFNβ production in *Myc/Trp53$^{-/-}$* spontaneous model as detected by ELISA assay ($n = 3$ biologically independent samples). **O** The effect of cGAS-KD on the tumor growth of PRMT3-KO Hepa1-6 cells as shown by measurement of tumor volumes ($n = 6$). **P** The effect of cGAS-KD on the tumor growth of PRMT3-KO Hepa1-6 cells as shown by tumor weights at the endpoint of the experiment on day 14 ($n = 6$). **Q** CD4$^+$ and CD8$^+$ T cells from the treated tumors were analyzed by flow cytometry for the indicated groups (sgNC, sgPRMT3, sgNC + shcGAS and sgPRMT3 + shcGAS) ($n = 5$). Data were analyzed by FlowJo. **R** Flow cytometry analysis assessing the percentage of T cell functional markers IFNγ and GZMB from tumors in indicated groups (sgNC, sgPRMT3, sgNC + shcGAS and sgPRMT3 + shcGAS) ($n = 5$). Data were analyzed by FlowJo. **S** The effects of cGAS-KD on the expression of ISGs in tumors from PRMT3-KO Hepa1-6 cells ($n = 3$ biologically independent samples). **T** The effects of cGAS-KD on the IFNβ production in tumors from PRMT3-KO Hepa1-6 cells as detected by ELISA assay ($n = 3$ biologically independent samples). Data in (**A**–**E**, **H**–**T**) are presented as mean ± SD. Data were analyzed by one-way ANOVA in (**A**–**E**, **H**–**T**). Source data are provided as a Source Data file.

(Fig. 8N, O). As expected, SGC707 treatment reprograms the relatively "cold" immune TME into a "hot" TME, as reflected by a significant increase in the infiltration of CD4$^+$ T cells, CD8$^+$ T cells, IFNγ$^+$ CD8$^+$ T cells, GZMB$^+$ CD8$^+$ T cells compared to the vehicle-treated group (Fig. 8P–R). Strikingly, SGC707 combined with anti-PD1 led to further increases in CD4$^+$ and CD8$^+$ T cell infiltration, including the GZMB$^+$ and IFNγ$^+$ CD8$^+$ T cells (Fig. 8P–R). Collectively, our findings suggested that targeting PRMT3 effectively enhances the response of HCC to anti-PD1 immunotherapy.

## Discussion

Immunotherapy including anti-PD1 and anti-PD-L1 has been recently widely used in HCC treatment. However, only a minority of HCC patients are responsive to ICB[4]. PRMT3 is frequently upregulated and has been proven to promote tumor progression in HCC[43], but its role in anti-tumor immunity is largely unknown. Here, we show that PRMT3 expression is negatively correlated with immune infiltration and response to ICB in HCC patients. Importantly, we found that ICB-induced activation of effector T cells triggered PRMT3 expression via an IFNγ-STAT1-dependent pathway. PRMT3 methylates HSP60 and promotes its multimerization to maintain mitochondrial homeostasis, which restricts mtDNA leakage and cGAS/STING pathway activation. Also, our study suggests that targeting PRMT3 may be an effective approach to improve the response of HCC to ICB.

Due to the low responsive rate of ICB in HCC patients, it is urgent to develop an effective biomarker to identify patients who would benefit from anti-PD1 and anti-PDL1 treatment[4]. Here, we used the TCGA dataset and found that PRMT3 was negatively correlated with the CD8$^+$ T cell infiltration and cytotoxic T cells, which was closely related to the treatment response of ICB. This correlation was further validated in HCC patients. In our investigation, PRMT3 was negatively correlated with treatment response, OS, and PFS after anti-PD1 and anti-PDL1 treatment in 2 independent cohorts. PRMT3 expression could be a potentially effective biomarker for immunotherapy in HCC. Clinically, PRMT3 expression in biopsy specimens before ICB treatment may help to guide individualized therapeutic strategies for HCC patients. However, a larger cohort should be enrolled to validate the effectiveness of using PRMT3 expression as a biomarker for predicting treatment response to ICB.

Accumulating studies proved that immune checkpoint therapy might lead to negative feedback and secondary immune resistance[6,7]. ICB-induced effector T cell activation triggered sialylation of IgG to inactivate the cGAS/STING pathway of macrophage and inhibit the anti-tumor immunity of tumor-associated macrophage[7]. In some cases, ICB would accelerate tumor progression by promoting IFNγ-induced oncogenic pathways and cancer stemness[24]. These observations indicated that T cell activation might lead to unpredictively adverse effects on tumor progression. In our study, immunotherapy-induced CD8$^+$ T cell activation up-regulated PRMT3 expression dependent on IFNγ-STAT1 signaling. Because high PRMT3 expression inhibited anti-tumor immunity and efficacy of ICB by methylating HSP60, our findings suggested that PRMT3 might be an important regulator of secondary immunotherapy resistance in HCC. For HCC patients with high PRMT3 expression, the primary immune resistance might occur because of the inactivation of cGAS/STING signaling and little immune infiltration. Patients with lower PRMT3 expression would benefit from ICB due to the abundant effective immune cell infiltration. However, ICB-induced PRMT3 overexpression in these tumors may drive the development of acquired resistance to ICB in HCC patients. Thus, PRMT3 inhibition may be an effective approach to delay the emergence of resistance to ICB in HCC patients who are highly responsive to ICB treatment. The development of a clinical-grade PRMT3 inhibitor is urgently needed for translating the preclinical findings into clinical trials in HCC patients.

PRMT3 has been reported to be involved in several biological processes in cancer progression recently[44,45]. We previously found that PRMT3 acted as a key driver for oxaliplatin resistance in HCC by methylating IGF2BP1 at R452 and activating WNT signaling[13]. IGF2BP1 was also reported to be involved in immunotherapy response by regulating several targets, such as PKM2, PDL1 and AXIN2[46–48]. However, PKM2, PDL1, and AXIN2 were not identified as targets of PRMT3 in HCC in our previous study[13]. In contrast, HSP60, which regulates the unfolding protein process and has been implicated in immune modulation, ranked as the top 1 candidate in the IP-MS in the Hepa1-6 cell line. Thus, we speculated that PRMT3 regulated the immune microenvironment of HCC by methylating HSP60, but not IGF2BP1. Although WNT signaling was also closely related to tumor immune microenvironment[49] and could be potentially regulated by PRMT3, the components of WNT signaling were not identified as direct substrates of PRMT3 and WNT signaling is downstream of the PRMT3-IGF2BP1-HEG1-WNT axis[13]. Moreover, a recent publication suggested that PRMT3 represses the innate immune response in viral infection by methylating cytosolic RNA and DNA sensors, such as RIG-I, MDA5, and cGAS[50]. However, RIG-I, MDA5, and cGAS were not identified in our IP-LC/MS data. We also performed Co-IP and found that endogenous PRMT3 did not interact with RIG-I, MDA5, and cGAS (Supplementary Fig. 13). Thus, RIG-I, MDA5, and cGAS might not be substrates of PRMT3 and are unlikely to mediate the function of PRMT3 in the inhibition of type I interferon signaling in HCC. Future studies are required to determine whether other PRMT3 substrates and WNT

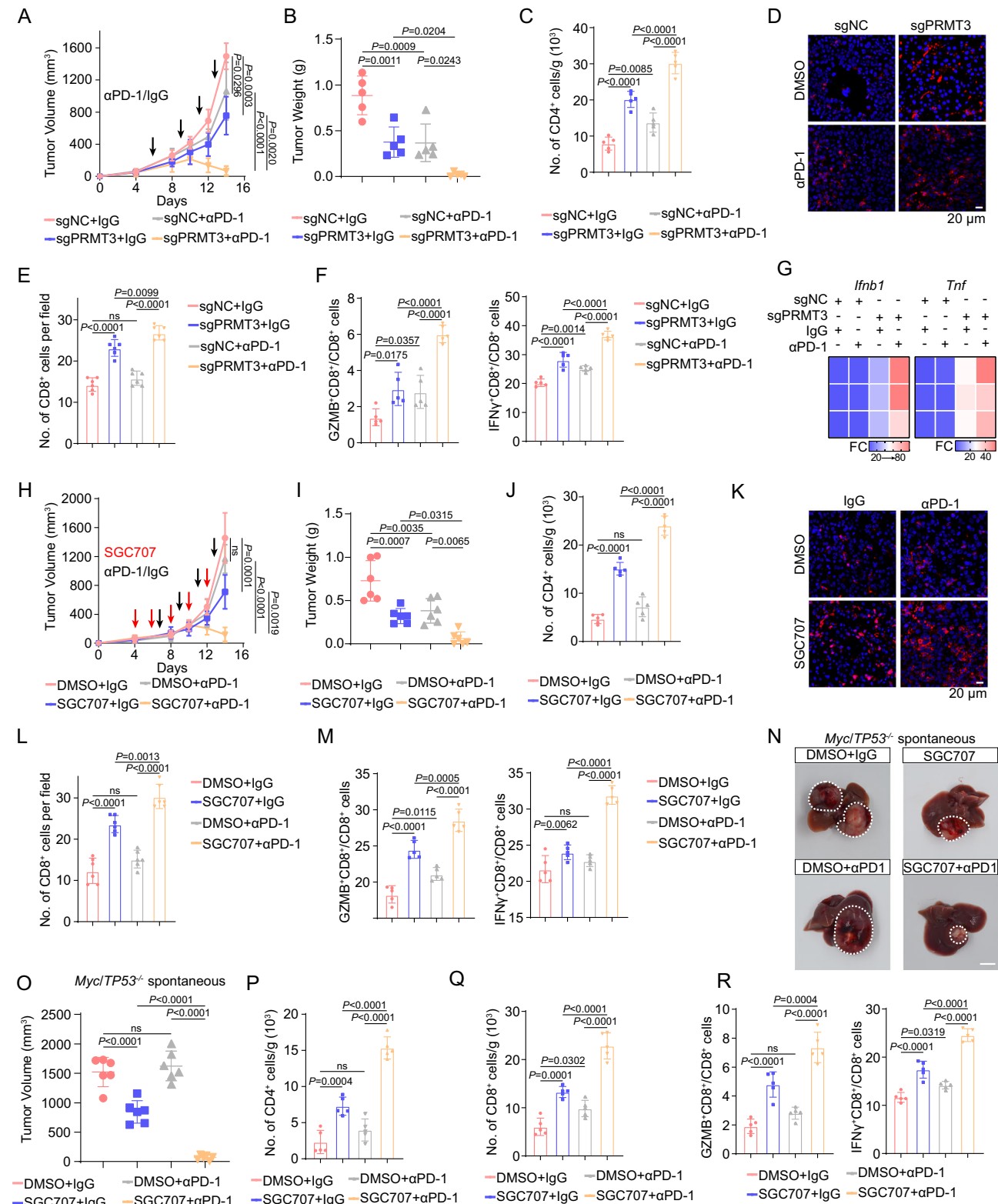

activation contribute to the functions of PRMT3 in anti-tumor immunity.

Furthermore, our data showed the comparable effects of global (pharmacologic) and genetic inhibition of PRMT3 on tumor growth and therapeutic outcomes, suggesting that PRMT3 inhibitors may mediate its anti-tumor effects through inhibition of PRMT3 expression in cancer cells. Interestingly, through analyzing publicly available HCC scRNA-seq datasets (GSE149614, GSE156625, and GSE202642)[8,51,52], we found that PRMT3 was mainly expressed in HCC cells in tumor niche and its expression levels in immune cells and stroma cells were much lower than in tumor cells (Supplementary Fig. 14). Also, our IF staining of PRMT3 in human and mouse HCC tissues showed that PRMT3 protein expression is much higher in cancer cells than in other cell types (Fig. 1E). Thus, the effects of pharmacologic inhibition of PRMT3 on tumor growth is likely to mediate through inhibition of PRMT3 in cancer cells. However, despite the low expression level of PRMT3 in

**Fig. 8 | *Prmt3*-KO or inhibition enhances immunotherapy response in HCC.**
**A**, **B** Hepa1-6 cells with or without PRMT3 knockout were implanted into C57BL/6 mice and treated with IgG or anti-PD1 (αPD1; *n* = 6 per group). Tumor volumes (**A**) and tumor weights (**B**) were presented. **C** CD4[+] T cells from tumors were analyzed by flow cytometry in indicated groups (sgNC + IgG, sgPRMT3 + IgG, sgNC + αPD1, and sgPRMT3 + αPD1) (*n* = 5). Data were analyzed by FlowJo. **D** Immunofluorescence identifying CD8[+] T cells in subcutaneous tumors treated with IgG or αPD1 from BALB/c mice injected with the indicated Hepa1-6 cells (*Prmt3*-KO or *Prmt3*-WT). **E** Quantitative analysis of CD8[+] T cells per field identified by immunofluorescence in indicated groups (sgNC + IgG, sgPRMT3 + IgG, sgNC + αPD1, and sgPRMT3 + αPD1) (*n* = 6). **F** Flow cytometry analysis assessing the percentage of T cells that were positive for functional markers IFNγ and GZMB from tumors in indicated groups (sgNC + IgG, sgPRMT3 + IgG, sgNC + αPD1, and sgPRMT3 + αPD1) (*n* = 5). Data were analyzed by FlowJo. **G** Ifnb1 and Tnf were assessed by qRT-PCR in indicated groups (sgNC + IgG, sgPRMT3 + IgG, sgNC + αPD1, and sgPRMT3 + αPD1) (*n* = 3 biologically independent samples). **H, I** Hepa1-6 cells were implanted into C57BL/6 mice and treated with DMSO + IgG, DMSO + anti-PD1, SGC707 + IgG, and SGC707 + anti-PD1 (αPD1; *n* = 6 per group). Tumor volumes (**H**) and tumor weights (**I**) were presented. Black arrow: anti-PD1/IgG treatment; Red arrow: SGC707 treatment. **J** CD4[+] T cells from tumors were analyzed by flow cytometry in indicated groups (DMSO + IgG, DMSO + anti-PD1, SGC707 + IgG, and SGC707 + anti-PD1) (*n* = 5). Data were analyzed by FlowJo. **K** Immunofluorescence identifying CD8[+] T cells in subcutaneous tumors from BALB/c mice treated with DMSO + IgG, DMSO + anti-PD1, SGC707 + IgG, and SGC707 + anti-PD1. **L** Quantitative analysis of CD8[+] T cells per field identified by immunofluorescence in indicated groups (DMSO + IgG, DMSO + anti-PD1, SGC707 + IgG, and SGC707 + anti-PD1) (*n* = 6). **M** Flow cytometry analysis assessing the percentage of T cells that were positive for functional markers IFNγ and GZMB from tumors in indicated groups (DMSO + IgG, DMSO + anti-PD1, SGC707 + IgG, and SGC707 + anti-PD1) (*n* = 5). Data were analyzed by FlowJo. **N, O** *Myc/TrpS3*[−/−] spontaneous HCC tumors were induced in mice, followed by treatment with DMSO + IgG, DMSO + anti-PD1, SGC707 + IgG, and SGC707 + anti-PD1 (αPD1; *n* = 6 per group). Tumor volumes were measured to compare the treatment effects on tumor progression. CD4[+] (**P**) and CD8[+] (**Q**) T cells from tumors were analyzed by flow cytometry in indicated groups (DMSO + IgG, DMSO + anti-PD1, SGC707 + IgG, and SGC707 + anti-PD1) (*n* = 5). Data were analyzed by FlowJo. **R** Flow cytometry analysis assessing the percentage of T cells that were positive for functional markers IFNγ and GZMB from *Myc/TrpS3*[−/−] spontaneous tumors in indicated groups (DMSO + IgG, DMSO + anti-PD1, SGC707 + IgG, and SGC707 + anti-PD1) (*n* = 5). Data were analyzed by FlowJo. Data in (**A–C, E, F, H–J, L, M, O–R**) are presented as mean ± SD. Data were analyzed by one-way ANOVA in (**A–C, E, F, H–J, L, M, O–R**). Source data are provided as a Source Data file.

stroma cells and immune cells, PRMT3 may play a role in stromal cells and immune cells in development or other diseases, which is worth being explored in future studies.

HSP60, a chaperone belonging to the heat shock proteins family that is located in mitochondria, has been reported to contain multiple post-translational modifications that regulate its functions[38]. For example, high glucose condition induces O-GlcNAcylation of HSP60 and influences its biological activities[53]. Other modifications, including S-guanylation, S-nitrosylation, citrullination, deamination, oxidation, and biotinylation, were also reported to regulate its biological functions[54–58]. It was reported that arginine asymmetric di-methylation of HSP60 was associated with the cellular senescence and proliferation potential of fibroblasts[59]. However, it is unclear which arginine residue was methylated and by which PRMTs. Here, we identified that HSP60 was methylated by PRMT3 at R446, which was located near the ATP-binding site and was reported to regulate its chaperoning activity and the ability to oligomerize. HSP60 in oligomeric conformation is known to work as a protein folding machine[40]. The function of protein folding in mitochondria was closely related to mtROS production, which was reported to regulate membrane permeability of mitochondria and mtDNA release[32,60–62]. Thus, inhibition of R446 methylation may inhibit the mitochondrial unfolded protein response, which could increase the mtROS production and membrane permeability of mitochondria in HCC. Indeed, our results indicate that inhibition of HSP60-R446 methylation restricted its oligomerize, increased the mtROS production and membrane permeability of mitochondria, promoted mtDNA leakage, and thus activated the cGAS/STING signaling. Moreover, we demonstrated that HSP60-D3G, a mutant with defective oligomerization[42], activated mtDNA-cGAS-STING-IFNβ pathway in HCC, and HSP60-D3G and HSP60-R446K overexpression had comparable effects on mtROS, membrane potential, mtDNA leakage, cGAMP level, ISGs expression, and IFNβ production. Thus, these data suggested that HSP60 oligomerization was directly involved in anti-tumor immunity and PRMT3 inhibition activated anti-tumor immunity by impairing HSP60 oligomerization.

To determine whether PRMT3 also plays a role in suppressing anti-tumor immunity in other cancer types, we analyzed the correlations between T cell infiltration and PRMT3 mRNA expression levels in various cancer types using Timer 2.0[37]. We also found a negative correlation between PRMT3 expression and CD8[+] T cell infiltration in immunotherapy-sensitive cancer types such as skin cutaneous melanoma (SKCM) and clear cell renal carcinoma (KIRC) (Supplementary Fig. 15). Previous studies also demonstrated that PRMT3 was involved in tumor progression and therapy resistance in several cancer types, including colorectal cancer, pancreatic cancer, endometrial carcinoma, and glioblastoma[44,45,63,64]. Thus, our studies suggest that PRMT3-mediated methylation of HSP60 may similarly drive therapeutic resistance to ICB in other cancer types, which will be explored in future studies.

It has been well established that several forms of cell death, including apoptosis, can elicit inflammatory response/immune reaction through releasing damage-associated molecular patterns (DAMP)[65]. In our previous study, *PRMT3*-KO or PRMT3 inhibitor induced apoptosis as indicated by elevated expression of cleaved-caspase 3[13]. We did not observe any significant up-regulation of HMGB1, a classical DAMP, in *PRMT3*-KD cells without stress conditions using our previous published RNA-seq dataset (GSE206502), but observed an upregulation of S100A9, another reported DAMP (increased by ~2.4 folds). Also, GSEA analysis of this RNA-seq dataset indicated that *PRMT3*-KD significantly up-regulated several inflammation-related pathways (Supplementary Fig. 16). Thus, we speculate that the upregulation of S100A9 and the inflammation-related pathways are caused by the impaired mitochondrial homeostasis and the activation of cGAS/STING pathway due to *PRMT3* knockout or inhibition. Similarly, HSP60 knockdown could also activate inflammation-related pathways due to impaired mitochondrial homeostasis and the activation of cGAS/STING pathway. However, we cannot rule out the possibility that PRMT3 knockout, HSP60 knockdown, or the presence of PRMT3 inhibitor led to an increase in the secretion and release HMGB1 and other DAMPs that directly cause inflammation/immune reaction, which will warrant a future study.

Although we demonstrated that the activation of T cell-dependent anti-tumor immunity mediates in large part the effects of *PRMT3* KO or PRMT3 inhibition in HCC progression in multiple model systems, we also observed a significant increase in the total number of B cells and a significant decrease in monocytes and macrophages in *Prmt3*-KO Hepa1-6 tumors (Fig. 3V). B cells have been linked to tumor progression, but their role in HCC remains unclear[66]. Also, monocytes and macrophages, could also promote tumor progression and resistance to immunotherapy[67,68]. Whether B cells, monocytes, or macrophages partially mediate the effects of PRMT3 on promoting immunotherapy resistance needs to be explored in future studies.

## Methods

### Patients and tissues

The usage of HCC samples was approved by the Ethics Committee of Sun Yat-Sen University Cancer Center (Ethics Approval ID: B2023-138-01). Written informed consent was obtained from each patient. The study design conformed to the ethical guidelines of the 1975 Declaration of Helsinki. Human HCC tissues and matched adjacent non-tumor liver tissues were obtained from patients who received curative surgery at the Sun Yat-sen Cancer Center (SYSUCC; Guangzhou, China). The cohort of HCC specimens, which included 228 HCC patients, and did not receive ICB treatment and were used for survival analysis, were collected from January 2010 to May 2015. Based on the median IHC scores, we then divided these patients into PRMT3-High group (112 patients) and PRMT3-Low group (116 patients). The HCC patient samples for efficacy evaluation were prospectively obtained from HCC patients who received anti-PD1/PDL1 treatment at the Sun Yat-sen University Cancer Center from 2018 to 2021. Patients were divided into 2 cohorts randomly to validate the association between PRMT3 and the efficacy of anti-PD1/PDL1 treatment.

### Evaluation of treatment efficacy

We evaluated the MRI images to determine the efficacy of immunotherapy in HCC using the Response Evaluation Criteria in Solid Tumors (RECIST) Guidelines[69,70]. Briefly, the image analysis was performed to measure the longest diameter for all target lesions and the responses are categorized into 4 groups: (1) complete response (CR)—the disappearance of all target lesions; (2) partial response (PR)—at least a 30% decrease in the sum of the longest diameter of target lesions, taking as reference the baseline sum longest diameter; (3) progressive disease (PD)—at least a 20% increase in the sum of the longest diameter of target lesions, taking as reference the smallest sum longest diameter recorded since the treatment started or the appearance of one or more new lesions; (4) stable disease (SD)—neither sufficient shrinkage to qualify for a partial response nor sufficient increase to qualify for progressive disease, taking as reference the smallest sum longest diameter since the treatment started. Complete response (CR) and partial response (PR) were defined as response (R). Progressive disease and stable disease were defined as non-response (NR). Each MRI image was evaluated by two independent radiologists in a single-blind method.

### Cell lines and cell culture

PLC-8024 (JNO-206), HepG2 (JNO-10-14-3), HEK293T (JNO-H0488) and Hepa1-6 cells (JNO-M0144) were purchased from the Guangzhou Jenniobio Biotechnology with STR (short tandem repeat) appraisal certificates. Cells were cultured in Dulbecco's Modified Eagle medium (DMEM; Thermo Fisher, C11995500BT) supplemented with 10% fetal bovine serum (FBS; Gibco, A5669701) at 37 °C in 5% $CO_2$.

### Single-cell RNA sequencing (scRNA-seq)

Tumor tissue was incubated in dissociation buffer containing 2 mg/ml Collagenase IV(Sigma, C5138), 2 mg/ml Collagenase II (Sigma C6885), 10 mg/ml Dispase II (Corning,354235) and 50,000 U/ml DNase I (Sigma, DN25) for 30 min at 37 °C. Enzymatic digestion was quenched with Dulbecco's Modified Eagle Medium (DMEM; Thermo Fisher, C11995500BT) supplemented with 10% fetal calf serum (FCS, Gibco, 10270106). Cells were resuspended at a concentration of 1 million cells per ml and stained with PBST containing 1% FCS. Then 15 µl of single-cell suspension at a concentration of ~900,000 cells/ml was loaded into one channel of the Chromium™ Single Cell B Chip (10X Genomics, 1000073), aiming for a recovery of 8000–9000 cells. The Chromium Single Cell 3' Library & Gel Bead Kit v3 (10X Genomics, 1000075) was used for single-cell barcoding, cDNA synthesis and library preparation, following the manufacturer's instructions according to the Single Cell 3' Reagent Kits User Guide Version 3. Libraries were sequenced on

Illumina novaseq6000 using a paired-end 150 bp. The CellRanger (6.1.2) pipeline was used for demultiplexing, barcode processing, alignment, and initial clustering of the raw scRNA-seq profiles. Raw sequencing reads were then aligned to the mouse genome (GRCm39, ENSEMBL). We then used Seurat (V4.3.0) to process the unique molecular identifier (UMI) count matrix and integrate all cells according to sample ID by using Harmony[71]. We performed the clustering analysis of all cells and only reserved the CD45+ clusters. To define the main immune cell types, we then split the cells according to commonly used cell markers (Cd3e for T cells, Cd79a for B cells, Lyz2 for Macrophages, S100a9 for Monocytes, Mki67 for Proliferating cells, Klra8 for NK cells, Sparc for Endothelial cells, and Fscn1 for Dendritic cells).

### Flow cytometry

The tumors were digested according to the manufacturer's instructions. T cells were stained with fluorochrome-conjugated antibodies according to the manufacturer's instructions and then analyzed by fluorescence-activated cell sorting (FACS). T cells under analysis were stained with surface markers, fixed, and permeabilized with IntraPrep reagent (Beckman Coulter, A07802), and finally stained with intracellular markers, GZMB (Invitrogen, 17-8898-82, 2.5 µl), IFNγ (BD Horizon, 563376, 2.5 µl). Data were acquired with a Gallios flow cytometer (Beckman Coulter) and analyzed with FlowJo (V10.5.3).

### Isolation of CD8+ T cells from tumor tissues

Tumor dissociation was performed using a Tumor Dissociation Kit (Miltenyi Biotec, 130-096-730). In brief, fat, fibrous and necrotic areas were removed from the tumor sample. Then tumors were cut into small pieces of 2–4 mm. The tumor tissue was enzymatically digested using the kit components and the gentleMACS™ Dissociators. CD8+T cells in single-cell suspensions were purified from tumors using a magnetic cell sorting column purification system (CD8+ T Cell Isolation Kit, Miltenyi Biotec, 130-117-044). The conditioned medium (CM) of sorted cells was collected after culturing the cells for 3 days. Then 2 ml CM was added into six-well plate to stimulate HCC cells for 48 h.

### Immunoblotting (IB)

Cells and tissues were lysed with RIPA lysis buffer (MedChemExpress, HY-K1001). Proteins were extracted and loaded in SDS-PAGE, and transferred onto PVDF membrane (Millipore, Billerica, 03010040001). After blocking with 5% skim milk (Beyotime, P0216) and sequential incubation with the primary and secondary antibodies: anti-PRMT3 (Abcam, Ab191562, 1:2000), anti-PRMT3 (Proteintech, 17628-1-AP, 1:1000), anti-HSP60 (Proteintech, 15282-1-AP, 1:1000), anti-ADMA (Cell Signaling Technology, 13522S, 1:1000), anti-FLAG (Cell Signaling Technology, 14793, 1:1000), anti-STAT1 (Proteintech, 10144-2-AP, 1:1000), anti-STING (Proteintech, 19851-1-AP, 1:1000), anti-p-STING (Cell Signaling Technology, 19781S, 1:1000), anti-TBK1 (Proteintech, 28397-1-AP, 1:1000), anti-p-TBK1 (Cell Signaling Technology, 5483S, 1:1000), anti-IRF3 (Proteintech, 11312-1-AP, 1:1000), anti-p-IRF3 (Cell Signaling Technology, 29047S#, 1:1000), anti-GAPDH (Proteintech, 60004-1-Ig, 1:2000), anti-mouse IgG (Cell Signaling Technology, 7076S, 1:3000), anti-rabbit IgG (Cell Signaling Technology, 7074S, 1:3000) (Supplementary Table 1), the blots were detected using the ECL detection kit (Yesen, 36208ES76).

### Quantitative RT real-time PCR (qPCR)

Total RNA extraction and cDNA synthesis were performed using RNA-Quick Purification Kit (EZBioscience, B0004D) and Super-Script™ III CellsDirect™ cDNA synthesis kits (Invitrogen, 18080300) according to the manufacturer's instructions. The cDNA products were used for qPCR analysis using SYBR

Green PCR kit (EZBioscience, CQ20). Supplementary Table 2 includes detailed information about the sequence of the used primers.

### Immunohistochemical (IHC) of clinical HCC specimens

Paraffin-embedded HCC tissues were cut into 5-mm sections. The sections were subsequently incubated with antibodies against human PRMT3 (Abcam, Ab191562, 1:200). The IHC score was determined by immunohistochemical staining intensity and the percentage of positively stained cells. Firstly, staining intensity, reflecting the color depth of the immunoreaction, was assessed subjectively under a microscope and graded on a scale of 0 to 3: 0 for no staining, 1 for weak staining, 2 for moderate staining, and 3 for strong staining. Subsequently, the percentage of positively stained cells, indicative of the extent of protein expression within the tissue sample, was estimated and recorded as a value ranging from 0 to 100% (0–25%, 1; 26–50%, 2; 51–75, 3; 76–100%, 4). The IHC score was then calculated by multiplying the staining intensity score by the percentage of positive cells. Immunohistochemical results were evaluated by 2 independent observers who were blinded to the clinical outcomes.

### Construction of PRMT3-KO cells using CRISPR/Cas9 technology

The gRNAs for PRMT3 knockout were designed using the MIT online tool CRISPRPICK (Supplementary Table 3). HEK293T cells were seeded in 6 cm plate and transfected with 2.5 mg lentiCRISPRv2-PRMT3-KO or lentiCRISPRv2 control plasmids, 5 mg psPAX2 and 2.5 mg pVSV-G plasmids using Lipofectamine 2000 (Thermo Fisher Scientific, 11668019) to produce lentivirus. The supernatants containing lentivirus were harvested and filtered 72 h post-transfection. To generate PRMT3-knockout cells, HCC cells were transduced with lentiviral vectors containing sgPRMT3 with 6 mg/mL of polybrene (Beyotime Biotechnology, C0351). After 72 h of transduction, HCC cells were subjected to 2 mg/mL puromycin (Gibco; A1113803) selection for several days. The efficiency was confirmed by western blotting.

### Small interfering RNAs (siRNA) for STAT1

Small interfering RNAs (siRNA) for STAT1 were purchased from Genepharma (A10001). Transfection of small interfering RNA was performed with Lipofectamine-RNAiMAX (Invitrogen, 13778100). After 6–8 h, the supernatant was replaced with fresh medium and the efficiency was identified by qRT-PCR and western blotting 72 h after transfections. Targeting sequences were listed in Supplementary Table 3.

### Chromatin-immunoprecipitation-quantitative PCR (ChIP-qPCR)

ChIP assays were performed with a ChIP kit (Cell Signaling Technology, 2952825) according to the manufacturer's instruments. Briefly, 1% formaldehyde (Sigma-Aldrich, 50-00-0) solution was added to induce PLC-8024 and Hepa1-6 cells crosslinking followed by glycine solution to quench the reaction. Afterward, the cells were lysed, and the nucleoprotein complexes were sonicated for 10 cycles of 10 s power-on and 20 s intervals with an intensity of 200 W with the sonicate conductor (Qsonica, Q700). Then, anti-STAT1 antibody or IgG, was added and incubated with the complexes overnight at 4 °C. The next day, Protein A/G magnetic beads were added to precipitate the indicated fragments for an additional 4 h at 4 °C. After extraction and purification of the indicated DNA, semiquantitative PCR was performed to identify the region interacting with the PRMT3-specific primers. The indicated primers were present in Supplementary Table 4. The experiments were performed in triplicate and the amount of immunoprecipitated DNA was normalized to the input.

### Immunoprecipitation (IP), mass spectrometry analysis of PRMT3 interacting proteins, and HSP60 arginine methylation

Cells and tissues were lysed using IP lysis buffer (Thermo Fisher Scientific, 87788) supplemented with proteinase inhibitor (Sigma-Aldrich, 539470) for 10 min, and cleared by centrifugation at 12,000 × g at 4 °C for 20 min. Cell lysate (2 mg) was subjected to IP with the indicated antibodies overnight at 4 °C. Then the lysate was incubated with protein A/G agarose beads (Thermo Fisher Scientific, 78610) for 1 h at room temperature. The beads were washed 3 times with IP lyse buffer. Then the proteins were eluted with loading buffer and detected by western blotting. For Mass spectrum analysis of PRMT3 interaction and HSP60 arginine methylation, the peptides were extracted and evaporated for liquid chromatography-mass spectrometry (LC-MS) analysis at the FITGENE company (Guangzhou, China).

### Immunofluorescent (IF)

For IF analysis of cultured cells, HCC cells were grown on chamber slides precoated with poly (L-lysine). Cells were fixed with cold paraformaldehyde. For the analysis of HCC tissues, we used formalin-fixed paraffin-embedded tissues (FFPE). Cultured cells or paraffin sections were permeabilized with PBS containing 0.1% Triton X-100, and blocked with AquaBlock (East Coast Bio, PP82). Cells were probed with the following primary antibodies as following: anti-PRMT3 (Proteintech, 17628-1-AP, 1:200), anti-HSP60 (Proteintech, 15282-1-AP, 1:200), anti-TFAM (Proteintech, CL488-22586, 1:200), mitotracker (Beyotime, C1035, 1:1000) (Supplementary Table 3). After washing the cells with PBS-T three times, the cells were incubated with 594 (or 488) labeled secondary antibodies (1:200) and DAPI-containing mounting solution VECTASHIELD (Vector Laboratories, Vector H-1000). The slices were visualized by using a Nikon inverted microscope Eclipse Ti-U equipped with a digital camera, or a Nikon A1 laser scanning confocal microscope at the Center for Advanced Microscopy/Nikon Imaging Center (CAM).

### Multiple immunofluorescent staining, tissue imaging, and analysis

Liver cancer tissue samples were used as experimental samples and tonsil tissues were used as controls (both as FFPE samples). All the tissues were cut and made as section slides with 2-μm thicknesses. The slides were deparaffinized in xylene for 10 min and repeated three times, and rehydrated in absolute ethyl alcohol for 5 min and repeated twice, 95% ethyl alcohol for 5 min, 75% ethyl alcohol for 2 min, sequentially. Then the slides were washed with distilled water 3 times. A microwave oven is used for heat-induced epitope retrieval; during epitope retrieval, the slides were immersed in boiling EDTA buffer (Molecular Depot, MDT-B2015130) for 15 min. Antibody Diluent/Block (MyBioSource, MBS539614) was used for blocking. The mIF staining part was performed and analyzed according to a 4-plex-5-color panel, and specifications (with primary antibodies used) are as the following: PRMT3 (Abcam, ab191562), CD4 (ZSGB-BIO, ZM0418), CD8 (ZSGB-BIO, ZA0508) and CD45 (Abcam, ab40763). All the primary antibodies were incubated for 1 h at 37 °C. Then slides were incubated with Polymer HRP Ms+Rb (PANO, 10013001050) for 10 min at 37 °C. Alpha X 7-Color IHC Kit (Alpha X Bio, AXT37100031) was used for visualization. The correspondences between primary antibodies and fluorophores are listed below: AlphaTSA 520 (CD4), AlphaTSA 570 (PRMT3), AlphaTSA 620 (CD8), AlphaTSA 690 (CD45). After each cycle of staining, heat-induced epitope retrieval was performed to remove all the antibodies including primary antibodies and secondary antibodies. The slides were counter-stained with DAPI (Cell Signaling Technology, 4083) for 5 min and enclosed in Antifade Mounting Medium (NobleRyder, I0052). Axioscan7 (ZEISS, Germany) was used for imaging the visual capturing. Data analysis was performed with Halo (3.4, Indica Labs, United States).

## HSP60 oligomerization detection

We examined HSP60 oligomerization using an established method as described[39]. Before protein extraction, cells were incubated with the crosslinking reagent BS3 reagent (Thermo Fisher, 21580) at a concentration of 1 mM for 15 min at room temperature. Then, 2 ml 1 M Tris-HCl pH 7.5 (Solarbio, T1140) was added for an additional 15 min to terminate the reaction. Then, the cells were lysed and subjected to western blotting as described above to assess HSP60 multimerization. The multimers were represented by bands over 180 KD, while HSP60 monomers were represented by bands at 60 KD.

## Enzyme-linked immunosorbent assay (ELISA)

Concentrations of the IFNβ from culture supernatants and tumor tissues were detected using enzyme-linked immunosorbent assay kits according to the manufacturer's instructions (Beyotime Biotechnology, PI572). To measure cGAMP levels, we used the 2',3'-Cycli-cGAMP ELISA Kit (Arbor Assays, K067-H1) according to the manufacturer's instructions.

## MitoROS detection and mitochondrial membrane potential measurement

MitoROS was detected with a MitoSOX™ Red Mitochondrial Superoxide Indicators (Invitrogen, M36009). Cells were incubated with 2 mL of the MSR reagent working solution to cover cells adhering to coverslips in a well of 35 mm dish for 30 min at 37 °C and 5% $CO_2$. Then cells were washed 3 times with PBS. MitoSOX was measured by flow cytometry.

## Animal experiments

All animal experiments were approved by the Institutional Animal Care and Use Committee of Sun Yat-Sen University Cancer Center (Ethics Approval ID: L102012021003X). BALB/C nude (GDMLAC-02) (male, 4 weeks) and C57BL/6 mice (GDMLAC-04) (male, 4 weeks) were purchased from the GUANGDONG MEDICAL LABORATORY ANIMAL CENTER and kept in an animal room with a 12-h light-dark cycle at a temperature of 20–22 °C with 40-70% humidity in specific-pathogen free (SPF) environment. The subcutaneous tumor model was established by inoculation of $2 \times 10^6$ Hepa1-6 cells into BALB/C nude and C57BL/6 mice. The experimental and control animals were cohoused before and were separated and divided into groups after the inoculation of cell lines. For the *Myc/Trp53*[−/−] spontaneous HCC model, C57BL/6 mice were hydrodynamically (within 3–5 s) injected via the lateral tail with a 0.9% NaCl solution/plasmid mix containing 10 μg sg-p53, 11.4 μg MYC-luc and 3,8 μg SB13 at a final volume of 10% of the body weight of mice. The mice were cohoused before and were separated and divided into groups after tumors reached about 50 mm³. Drug administration was adopted when the tumors reached about 50 mm³ in size, at which point mice were randomized for treatment with DMSO (intraperitoneally), SGC707 (20 mg/kg/every 3 days, intraperitoneally), RU.521 (5 mg/kg/every 3 days, intraperitoneally), anti-CD3 (100 μg/every 3 days, intraperitoneally), anti-CD4 (100 μg/every 3 days, intraperitoneally), anti-CD8 (100/μg every 3 days, intraperitoneally), or an-PD1 antibody (200 μg/every 3 days, intraperitoneally). Mice were euthanized by $CO_2$ asphyxiation for tumor harvesting after the appearance of tumors with a diameter greater than 1.5 cm in any group. The weights of the excised tumors were recorded. Tumors were removed for further analysis.

## Statistics and reproducibility

All the data was analyzed with GraphPad Prism version 8.0 and IBM SPSS Statistics 25 for Windows. Chi-Square Test was used to analyze the differences in the response rates. The two-tailed Student's *t* test was used unless otherwise stated for comparing two groups. The ANOVA was used for comparing more than two groups. All boxplots indicate median (center), 25th and 75th percentiles (bounds of box), and minimum and maximum (whiskers). Experiments were performed a minimum of three times. Cumulative survival time was estimated by the Kaplan–Meier method, and the log-rank test was applied to compare the groups. $p < 0.05$ was considered statistically significant. All grouped data are presented as mean ± SD unless otherwise stated.

## Reporting summary

Further information on research design is available in the Nature Portfolio Reporting Summary linked to this article.

## Data availability

The sequence data generated in this study have been deposited in the GEO database under the accession numbers GSE206502 and GSE249999. Proteomics raw data have been deposited in the Proteome Xchange under the accession number PXD047693. All data are included in the Supplementary Information or available from the authors, as are unique reagents used in this Article. The raw numbers for charts and graphs are available in the Source Data file whenever possible. Source data are provided with this paper.

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

## Acknowledgements

This work was supported by grants from the National Natural Science Foundation of China (Nos. 82172815 and 82373405 to BK.L. and 82272887 to YF.Y.).

## Author contributions

Yunxing Shi, Zongfeng Wu, Shaoru Liu and Dinglan Zuo designed experimental approaches, performed experiments, analyzed data, and co-wrote the manuscript; Yuxiong Qiu, Liang Qiao, Yi Niu, Wei He and Jiliang Qiu performed experiments, analyzed data, and provided critical input; Yunfei Yuan, Guocan Wang and Binkui Li designed experimental approach, analyzed data, provided oversight and critical expertise, and co-wrote the manuscript.

## Competing interests

The authors declare no competing interests.
