## [Peer Review File · Nature Communications]

REVIEWER COMMENTS

Reviewer #1 (Epigenetic regulation of cancer immunity)(Remarks to the Author):

The manuscript provides a comprehensive analysis of the role of PRMT3 in regulating anti-tumor immunity in hepatocellular carcinoma (HCC), through a series of in silico analyses and in vivo studies. The authors identified PRMT3 as a key player associated with CD8 T cell abundance, cytotoxic signature, patients' survival, and response to immune checkpoint blockade (ICB) therapy in TCGA-LIHC and in-house datasets. The study further identified HSP60 as a PRMT3 substrate and explored its function in mediating anti-tumor immunity, shedding light on mitochondrial homeostasis and concomitant activation of type I interferon pathways via mtDNA leakage/cGAS/STING activation. The study then provides data to suggest that targeting PRMT3 in HCC effectively enhanced anti-PD-1 immunotherapy in both subcutaneous and spontaneous mouse HCC models.

A recent publication demonstrates the role of PRMT3 in regulating the anti-viral innate immune response (PMID: 37639603), where PRMT3 inhibition activated type I interferon pathway through methylation of RNA- and DNA-sensors. The current study is somewhat similar, given the parallels between the control of innate immune response in viral infection and the anti-tumor immune response. The hypothesis underlying this study was rigorously explored through well-organized experiments. However, to substantiate authors main conclusions, the authors need to consider addressing the following points:

1. To ensure consistency across multiple datasets, evaluating patients' overall and progression-free survival in the TCGA-LIHC dataset is essential. Figure 1D should provide comprehensive information, including the number of patients in each subset, the expression level of PRMT3 in each patient's group, and a clear description of the method used to analyze patients' data.
2. The data presented in Figure 1G requires a detailed explanation of how to interpret the MRI images. Additionally, please detail the statistical methods employed to determine significance in Figures 1H and 1J. It is also recommended to present quantitative data on low/high PRMT3 expression in Figures 1H-1K.
3. The authors suggest that T-cell activation induces PRMT3 expression through IFN γ -STAT1 pathways (Figure 2). Furthermore, the authors claimed a positive correlation between the IFN γ -dependent transcriptional program (STAT1) and PRMT3 expression (line 167 ~ line 171). To strengthen this claim, authors should investigate whether IFN γ pathway genes (Ccl5, Cxcl10, Isg15) positively correlate with PRMT3, like STAT1. Moreover, test IFN γ -STAT1-dependent induction of PRMT3 in multiple HCC cell lines.
4. Please demonstrate whether the STAT1 binding region is conserved within the PRMT3 promoter in the mouse genome and show STAT1 binding to the mouse PRMT3 promoter.
5. Figure 2 requires additional information: the amount of CM, treatment time of the conditioned medium (CM), and IFN γ (Figures 2E, 2I, and 2K). Please provide blots for STAT1 expression after STAT1 knockdown (Figure 2I) and data without IFN γ treatment (Figure 2K)
6. Does PRMT3 knockout, HSP60 knockdown, or the presence of PRMT3 inhibitor induce cell death that elicits inflammation/immune reaction?
7. Given the significant changes in T cell abundance and activation by PRMT3 KO or its inhibition, authors should determine whether the neutralization of CD8, CD4 T cells, or NK cells attenuates tumor growth inhibition.

8. Please provide direct evidence to support the causative role of HSP60 oligomerization in anti-tumor immunity. Likewise, please demonstrate whether the HSP60 mutant with defective oligomerization exhibits the same phenotypes observed in the methylation-defective mutant.
9. To support the authors' conclusion of immune dependency, authors should evaluate the tumor growth upon HSP60 knockdown (Figure 5B) in immune-compromised mice (e.g., NSG).
10. Please display the activation of STING (phosphorylation, oligomerization) in the immunoblots shown in Figure 6. Reassess whether the "+/-" signs are appropriately aligned in Figure 6H.
11. Provide the PRMT3 expression in different cell types in the tumor niche of HCC from publicly available scRNAseq data. Given the comparable effects of global (pharmacologic) and genetic (tumor cell-specific) inhibition of PRMT3 on tumor growth and therapeutic outcomes, discuss the potential influence of global PRMT3 inhibition on other cell types in the tumor niche, including immune cells and stroma cells.
12. Considering a recent publication suggesting that PRMT3 represses the innate immune response in viral infection (PMID: 37639603), authors should determine and discuss the possibility of PRMT3's involvement in the methylation of cytosolic RNA and DNA sensors, such as RIG-I, MDA5, and cGAS.

Reviewer #2 (ICB/cancer immune therapy)(Remarks to the Author):

The paper by Shi et al describes a role for an IFN γ -induced methyltransferase (PRMT3) in acquired resistance to immunotherapy. Expression of this methyltransferase correlated with T cell infiltration and clinical outcome in HCC patients and tissue slide staining confirmed this on protein level in two HCC patient cohorts receiving checkpoint therapy. PRMT3 is upregulated by IFN γ and prevented T cell infiltration in mouse HCC tumor models. Extensive mechanistic experiments using gene knockdowns and chemical inhibitors in tumor cells demonstrated the pathway of PRMT3-> HSP60 oligomerization in mitochondria-> prevention of cGAS/STING activation->prevention of T cell infiltration. In addition, re-expression studies of WT genes or mutant genes confirmed the findings. Finally, PRMT3 inhibition resulted in much stronger growth delay in immunocompetent mice compared to immunodeficient mice and sensitizes tumors for checkpoint therapy. The specific inhibitor SGC707 represents an ideal inhibitor of this immune resistance pathway and ready seems for clinical use. PRMT3 is suggested to block a feedforward accumulation of IFN γ -induced tumor immunity, in which initial responses are hampered but, moreover, inhibition of this molecule even can provoke inflammation in cold tumors. The study is well organized, well presented and extremely well controlled. Conclusions are supported by the data. Discussion includes open questions and unresolved issues.

Comments:

1. HCC can have a viral cause or can be sporadic. Does the effect of PRMT3 mainly play a role in the virus-induced cancers? Patient databases can be analysed for this.
2. Is there a reason why this resistance mechanism is only functional in HCC? Correlations between T cell profiles and PRMT3 levels (fig 1) can be analysed in other cancer types, including checkpoint sensitive cancers.
3. Fig 1: correlation between cytotoxic signature score and PRMT3 expression is rather poor, especially the PRMT3 low tumors display seem to allow T cell infiltrate (fig 1b-c). In the other panels, samples are

divided between PRMT3 low and high tumors. Were all samples included in these analyses of fig 1d-k? Were samples allocated based on median? Or were only extremes taken for PRMT3 expression in tumors?

4. Figure 2 shows IFN γ -induced expression of PRMT3 and that (obviously) STAT1 is indispensable for this. However type I interferons also transduce signals via STAT1, do they also induce expression of PRMT3?

5. Correlations with T cell influx after PRMT3 inhibition are consistent throughout the study, but fig 3 also shows a dramatic shift in B cell influx in PRMT3-knockout tumors. RAG knockout mice were used to show involvement of the immune system, but these mice also lack B cells. A direct role for T cells should be demonstrated with antibody (aCD4 and aCD8) depletion experiments during treatment in a mouse tumor model.

6. Fig 6 and 7 show upregulation of some STING targets as a result of PRMT3 shutdown, but the relevant CXCL9/10/11 chemokines, which are often responsible for T cell attraction to tumors, were not tested. This is important since the final result is more T cell influx.

7. In their previous paper on PRMT3 (ref 13, Nat Comm 2023) authors show IGF2BP1 as a downstream target after oxaliplatin treatment. Why was this molecule not found among the pulldown proteins in this study? One would expect similar targets after high expression of PRMT3, unless this enzyme itself is modified differently depending on the type of treatment?

Reviewer #3 (HCC)(Remarks to the Author):

The authors reported that PRMT3 acts as a novel driver of immunotherapy resistance in HCC. They first found that PRMT3 expression was induced by activated T cells in response to ICB via IFN- γ -STAT1 signaling pathway, and that high PRMT3 expression inversely correlates with tumor-infiltrating CD8+ T cells and predicts poor response to ICB in HCC patients. Mechanically, they revealed that PRMT3 methylated the mitochondrial chaperone protein HSP60 at R446 to induce its oligomerization for maintenance of mitochondrial homeostasis in tumor cells. Loss of PRMT3 caused mitochondria damage to release mtDNA into cytosol to trigger cGAS-STING activation and subsequent antitumor immunity. PRMT3 inhibition further enhanced the efficacy of ICB treatment anti-PD1 efficacy in HCC mouse models. Overall, the findings reported are intriguing, however, addressing the points below is necessary to strengthen and validate the proposed model and increase the accuracy and reliability of the results. My specific comments are listed here below.

Major points

1. More clinical HCC tumor samples should be included for WB detection of PRMT3 in Fig S2A, and Fig 2B and 2C.

2. In Fig S3A, the authors showed that Myc/Trp53^{-/-} tumors had very few CD8+ T cell infiltration levels, which was responsible for the failure in PRMT3 elevation after anti-PD1 treatment. However, they only conducted immunofluorescence to examine CD8 T cell infiltration. Flow cytometry analysis should be included. In addition, according to the paper (PMID: 34290403 DOI: 10.1038/s41586-021-03741-7), both Hepa1-6 and Myc/Trp53^{-/-} mouse liver cancer had basal infiltration of CD8 T cells, and combination treatment of Lenvatinib and gefitinib further increased influx CD8+ T cells in these tumors. Why the authors obtained the inconsistent results and reached the conclusion that Myc/Trp53^{-/-} are immune “cold” tumor here? The author may deplete CD8 T cells to further examine their effects on PRMT3 expression in Hepa1-6 tumor bearing mouse models.

3. IFN γ and TNF α are cytokines that are released by effector CD8 $^+$ T cells. They should also examine the effects of TNF α on PRMT3 expression. The authors used IFN γ to directly treat tumor cells for detecting PRMT3 expression. They may add the indicated inhibitors of IFN γ and TNF α into CD8 $^+$ T-cell supernatant and check tumor intrinsic PRMT3 expression after incubation.

4. In Fig 4K, why Prmt3-KO or PRMT3 inhibition notably decreased the oligomerization of HSP60, but failed to increased HSP60 monomers? In Fig 5I and J, the tumor volume of PRMT3 $^+$ HSP60-WT seems to be comparable to that of PRMT3 $^+$ HSP60-RK, and in line 321-325, the description is logically mistaken. The conclusion "These findings suggest that other PRMT3 substrates may also contribute to the effects of PRMT3-OE on HCC growth and T cell infiltration" is not that appropriate. These results seemed to suggest that PRMT3 arginine methylation of HSP60 is required for the promoting HCC growth, since the tumor growth and T cell infiltration of HSP60-WT and HSP60-RK were both at comparable levels in the absence of PRMT3.

5. In Fig 6C-E, the author attempted to demonstrate that lack of PRMT3-HSP60 caused mtDNA leakage into cytosol, but the confocal images were not clearly clarified. First, it is impossible that mtDNAs were not detected in mitochondria of WT cells. It is well-acknowledged that TOM20 indicates mitochondrion but not mtDNA. TFAM is the specific mtDNA binding protein, which should be stained to precisely indicate the locations of dsDNA (PMID: 26305956, PMID: 38168624).

6. The authors only examined pTBK1 and pIRF3 to indicate cGAS activation. cGAMP production, the direct readout of cGAS activation, should be evaluated after inhibition of PRMT3-HSP60. Alternatively, the authors may compare the differences between cGAS-KD and WT cells to highlight the importance of cGAS in this paper.

7. It has been reported that PRMT3 directly interacts with cGAS and catalyzes its asymmetric demethylation (PMID: 37639603). It will be a good supplementation to this work if the authors evaluate the effects of PRMT3 on cGAS activity. And the authors may further evaluate the effects of PRMT3 inhibitor on antitumor immunity in WT and cGAS-KD tumor models.

Minor:

In Fig 6H, Figure labels should be corrected.

In Fig 6I, does the authors mistake the labels "pTBK" and "TBK1"?

In Line 369, "Fig. 7H, I" should be corrected to Fig. 6H, I.

Based on the comments above, my verdict of this study is "major revision".

Point-by-point response to the reviewers' comments

REVIEWER COMMENTS

Reviewer #1 (Epigenetic regulation of cancer immunity) (Remarks to the Author):

The manuscript provides a comprehensive analysis of the role of PRMT3 in regulating anti-tumor immunity in hepatocellular carcinoma (HCC), through a series of in silico analyses and in vivo studies. The authors identified PRMT3 as a key player associated with CD8 T cell abundance, cytotoxic signature, patients' survival, and response to immune checkpoint blockade (ICB) therapy in TCGA-LIHC and in-house datasets. The study further identified HSP60 as a PRMT3 substrate and explored its function in mediating anti-tumor immunity, shedding light on mitochondrial homeostasis and concomitant activation of type I interferon pathways via mtDNA leakage/cGAS/STING activation. The study then provides data to suggest that targeting PRMT3 in HCC effectively enhanced anti-PD-1 immunotherapy in both subcutaneous and spontaneous mouse HCC models.

A recent publication demonstrates the role of PRMT3 in regulating the anti-viral innate immune response (PMID: 37639603), where PRMT3 inhibition activated type I interferon pathway through methylation of RNA- and DNA-sensors. The current study is somewhat similar, given the parallels between the control of innate immune response in viral infection and the anti-tumor immune response. The hypothesis underlying this study was rigorously explored through well-organized experiments.

We thank this reviewer for the insightful and constructive comments and the recognition of the strength of our study. We have addressed this reviewer's concerns by including additional data and a detailed description of results and methods in the revised manuscript (see below).

However, to substantiate authors main conclusions, the authors need to consider addressing the following points:

1. To ensure consistency across multiple datasets, evaluating patients' overall and progression-free survival in the TCGA-LIHC dataset is essential. Figure 1D should provide comprehensive information, including the number of patients in each subset, the expression level of PRMT3 in each patient's group, and a clear description of the method used to analyze patients' data.

We thank this reviewer for the comments. As requested, we have included the correlation between *PRMT3* expression and the overall survival and progression-free survival in the TCGA-LIHC dataset in the Results section (**Supplementary Fig. 2A-B**) (Page 5 Line 96-line 99, Page 5 Line 102- page 6 line 104). Also, we have included a detailed description of the SYSUCC HCC cohort, and the methods used for analyzing the data used for Fig. 1D in the Material and methods section (Page 29 line 626- page 30 line 629, Page 39 line 841- line 843). Furthermore, the expression levels of PRMT3 in each group were included in our Source Data.

Result: "Among these 16 genes, which show a negative correlation with both cytotoxic score and CD8+ T cell infiltration, we found that higher expression levels of 7 of them are strongly associated with shorter overall survival in HCC patients (**Fig. 1A, Supplementary Fig. 2A**)." (Page 5 Line 96-line 99)

"Further analysis indicated that PRMT3 was also associated with shorter progression-free survival in HCC patients in TCGA-LIHC dataset (**Supplementary Fig. 2B**)." (Page 5 line 102- page 6 line 104)

Material and methods: "The cohort of HCC specimens, which included 228 HCC patients, and did not receive ICB treatment and were used for survival analysis, were collected from January 2010 to May 2015. Based on the median IHC scores, we then divided these

patients into PRMT3-High group (112 patients) and PRMT3-Low group (116 patients).”
(Page 29 line 626- page 30 line 629)

“Cumulative survival time was estimated by the Kaplan-Meier method, and the log-rank test was performed to compare the survival between these two groups. $P < 0.05$ was considered statistically significant.” **(Page 39 line 841- line 843)**

2. The data presented in Figure 1G requires a detailed explanation of how to interpret the MRI images. Additionally, please detail the statistical methods employed to determine significance in Figures 1H and 1J. It is also recommended to present quantitative data on low/high PRMT3 expression in Figures 1H-1K.

As requested, we have included the following information the Material and methods section: (1) a detailed explanation of how to interpret the MRI images for data presented in Fig. 1G **(Page 30 line 634- line 648)**; (2) detailed statistical methods employed to determine the significance in Fig. 1H and 1J **(Page 39 line 838)**; and (3) quantitative data on PRMT3 expression levels included in our Source Data.

Material and methods:

“We evaluated the MRI images to determine the efficacy of immunotherapy in HCC using the Response Evaluation Criteria in Solid Tumors (RECIST) Guidelines.^{1,2} Briefly, the image analysis was performed to measure the longest diameter for all target lesions and the responses are categorized into 4 groups: (1) complete response (CR)—the disappearance of all target lesions; (2) partial response (PR)—at least a 30% decrease in the sum of the longest diameter of target lesions, taking as reference the baseline sum longest diameter; (3) progressive disease (PD)—at least a 20% increase in the sum of the longest diameter of target lesions, taking as reference the smallest sum longest diameter recorded since the treatment started or the appearance of one or more new lesions; (4) stable disease (SD)—neither sufficient shrinkage to qualify for partial response nor sufficient increase to qualify for progressive disease, taking as reference the smallest sum longest diameter since the treatment started. Complete response (CR) and partial

response (PR) were defined as response (R). Progressive disease and stable disease were defined as non-response (NR). Each MRI image was evaluated by two independent radiologists in a single-blind method.” (Page 30 line 634- line 648)

“Chi-Square Test was used to analyze the differences in the response rates.” (Page 39 line 838)

3. The authors suggest that T-cell activation induces PRMT3 expression through IFN γ -STAT1 pathways (Figure 2). Furthermore, the authors claimed a positive correlation between the IFN γ -dependent transcriptional program (STAT1) and PRMT3 expression (line 167 ~ line 171). To strengthen this claim, authors should investigate whether IFN γ pathway genes (*Ccl5*, *Cxcl10*, *Isg15*) positively correlate with PRMT3, like STAT1. Moreover, test IFN γ -STAT1-dependent induction of PRMT3 in multiple HCC cell lines.

We thank this reviewer for the constructive comments. As requested, we have examined the correlation between IFN γ pathway genes (*Ccl5*, *Cxcl10*, and *Isg15*) and *PRMT3* expression in TCGA-LIHC RNA-seq datasets (Supplementary Fig. 9M) in the Result section (Page 20 line 432- page 21 line 435). We found that the expression levels of IFN γ pathway genes (*Ccl5*, *Cxcl10*, and *Isg15*) were negatively correlated with *PRMT3* expression. Importantly, we provided additional data (Supplementary Fig. 4F) and a plausible explanation for these findings in the Results section (Page 9 line 173- line 180).

Results (Page 9 line 173- line 180): “However, PRMT3 expression is only slightly upregulated at 24 hours but dramatically increased at 48 hours after IFN γ treatment (Supplementary Fig. 4F). As expected, we indeed observed an upregulation of these IFN γ regulated genes at 24 hours, but their expression was dampened at 48 hours. These findings indicated that the upregulation of PRMT3 in response to IFN γ may suppress the expression of interferon-stimulated genes (ISGs), which play a critical role in the anti-tumor immunity. Consistent with this finding, we found that the effect of IFN γ on the induction of its downstream target genes was more effective in PRMT3-KO Hepa1-6 cells than in the control cells (Supplementary Fig. 4F).”

Furthermore, we have tested test the IFN γ -STAT1-dependent induction of PRMT3 in additional HCC cell lines and have added these new findings to the Results section (**Page 9 line 170- line 173, Page 9 line 189- page 10 line 192**).

Results (**Page 9 line 170- line 173**): “To test this, we treated several HCC cell lines (PLC-8024, Hepa1-6, and H22) with TNF α and IFN γ for 24 and 48 hours, and found that IFN γ , but not TNF α , increased PRMT3 expression in a time-dependent manner (Fig.2G, H, Supplementary Fig. 4E).”

Results (**Page 9 line 189- page 10 line 192**): “We found that Stat1-KD effectively abrogated the elevation of Prmt3 mRNA and protein expression induced by IFN γ treatment (Fig. 2I, Supplementary Fig. 4I), which indicates that PRMT3 was regulated via the IFN γ -STAT1 axis.”

4. Please demonstrate whether the STAT1 binding region is conserved within the PRMT3 promoter in the mouse genome and show STAT1 binding to the mouse PRMT3 promoter.

We thank this reviewer for the comments. The STAT1 binding region found in the human PRMT3 promoter is not conserved within the mouse *Prmt3* promoter (**Supplementary Fig. 4L**). However, we found that STAT1 similarly binds to a region of mouse *Prmt3* promoter that is not conserved between human and mouse. We have included these findings in the Results section (**Page 10 line 194- line 200**). Also, our results demonstrated that anti-PD1 therapy, the conditioned medium (CM) prepared from cultured mouse CD8+ T cells, or mouse CD8+ T cell co-culture, and IFN γ stimulation up-regulated PRMT3 expression in HCC cells. Importantly, STAT1 knockdown (KD) in Hepa1-6 cells abolished the effect of IFN γ on PRMT3 up-regulation (**Figure 2, Supplementary Fig. 4**). Thus, our data supported the conclusion that STAT1 transcriptionally regulated PRMT3 in HCC.

Results: “Also, we examined the ENCODE chromatin immunoprecipitation sequencing (ChIP-seq) data (HepG2 cell line) and found that STAT1 directly binds to the promoter region of *PRMT3* (**Fig. 2J**),²⁹ which contains STAT1 binding motif obtained from JASPAR and FIMO database^{30,31} (**Supplementary Fig. 4K, L**). We then performed ChIP-qPCR to

examine the binding of STAT1 to PRMT3 promoter. We found that STAT1 indeed bound to the putative promoters of *PRMT3* in PLC-8024 and Hepa1-6 HCC cells treated with IFN γ (Fig. 2K; Supplementary Fig. 4L).” **(Page 10 line 194- line 200)**

5. Figure 2 requires additional information: the amount of CM, treatment time of the conditioned medium (CM), and IFN γ (Figures 2E, 2I, and 2K). Please provide blots for STAT1 expression after STAT1 knockdown (Figure 2I) and data without IFN γ treatment (Figure 2K)

As requested, we have included detailed information for Figure 2 in the Materials and methods section **(Page 32 line 683-line 691)** and the corresponding figure legends **(Page 48 line 1085- line 1086, Page 48 line 1089- line 1090)** in this revised manuscript.

Materials and methods: “Tumor dissociation was performed using a Tumor Dissociation Kit (Miltenyi Biotec). In brief, fat, fibrous and necrotic areas were removed from the tumor sample. Then tumors were cut into small pieces of 2-4 mm. The tumor tissue was enzymatically digested using the kit components and the gentleMACS™ Dissociators. CD8⁺T cells in single-cell suspensions were purified from tumors using a magnetic cell sorting column purification system (CD8⁺ T Cell Isolation Kit, Miltenyi Biotec). The conditioned medium (CM) of sorted cells was collected after culturing the cells for 3 days. Then 2 ml CM was added into six-well plate to stimulate HCC cells for 48 hours.”

Figure legends: “I. PRMT3 expression in Stat1-KD and control cells treated with IFN γ (20 ng/ml, 48h) as shown by qRT-PCR and western blot (n=3 biologically independent samples).” **(Page 48 line 1085- line 1086)**

“K. ChIP-qPCR was used to determine the binding of STAT1 to PRMT3 promoter region in PLC-8024 cells treated with IFN γ (20 ng/ml, 48h) (n=3 biologically independent samples).” **(Page 48 line 1089- line 1090)**

6. Does PRMT3 knockout, HSP60 knockdown, or the presence of PRMT3 inhibitor induce cell death that elicits inflammation/immune reaction?

We thank this reviewer for the comments. We don't have direct evidence that PRMT3 knockout, HSP60 knockdown, or the presence of PRMT3 inhibitor induce cell death that elicits inflammation/immune reaction at this stage (see below).

It has been well established that several forms of cell death, including apoptosis, can elicit inflammatory response/immune reaction through releasing damage-associated molecular patterns (DAMPs).³ Also, although apoptosis is traditionally considered a non-immunogenic form of cell death that prevents the release of intracellular contents because there is no loss of membrane integrity, recent studies suggest that apoptosis can be immunogenic under stress conditions such as chemotherapy or physical modalities. In our previous study,⁴ PRMT3-KO or PRMT3 inhibitor-induced apoptosis as indicated by elevated expression of cleaved-caspase 3.⁵ We did not observe any significant up-regulation of HMGB1, a classical DAMP, in PRMT3-KD cells without stress conditions using our previous published RNA-seq dataset (GSE206502), but observed an upregulation of S100A9, another reported DAMP (increased by ~2.4 folds). Also, GSEA analysis of our previous RNA-seq dataset⁴ indicated that PRMT3-KD significantly up-regulated several inflammation-related pathways (**Additional Fig. 1, see below**). Thus, we speculate that the upregulation of S100A9 and the inflammation-related pathways are caused by the impaired mitochondrial homeostasis and the activation of cGAS/STING pathway due to PRMT3 knockout or inhibition. Similarly, HSP60 knockdown could also activate inflammation-related pathways due to impaired mitochondrial homeostasis and the activation of cGAS/STING pathway. However, we cannot rule out the possibility that PRMT3 knockout, HSP60 knockdown, or the presence of PRMT3 inhibitor led to an increase in the secretion and release HMGB1 and other DAMPs that directly cause inflammation/immune reaction, which will warrant a future study.

Additional Fig. 1. Gene set enrichment analysis identified inflammation-related pathways in PRMT3 -KD cells.

7. Given the significant changes in T cell abundance and activation by PRMT3 KO or its inhibition, authors should determine whether the neutralization of CD8, CD4 T cells, or NK cells attenuates tumor growth inhibition.

We thank this reviewer and other reviewers for the insightful comments. We have shown that neutralization of CD3+, CD4+, and CD8+ T cells attenuated tumor growth inhibition (**Supplementary Fig. 6**). Of note, the effects were more pronounced for anti-CD3 and anti-CD8 compared to anti-CD4, suggesting that CD8 T cells may play a major role in suppressing tumor progression. Since we observed a dramatic decrease of NK cells in PRMT3-KO tumors as shown by the scRNA-seq analysis, we did not neutralize NK cells in the mice. These findings have been included in the Results section (**Page 12 line 253- page 13 line 261**).

Results: “Given the significant changes in T cell abundance and activation by *Prmt3*-KO and PRMT3 inhibition, we then examined whether the effects of *Prmt3*-KO and PRMT3 inhibition on tumor progression were mediated by CD8⁺ or CD4⁺ T cells. We treated tumor-bearing mice with anti-CD3, anti-CD4, or anti-CD8 antibodies and found that neutralization of CD3⁺, CD8⁺ and CD4⁺ T cells indeed attenuated the tumor growth inhibition induced by *Prmt3*-KO (Supplementary Fig. 6A-F). Of note, the effect of anti-CD8 antibodies on tumor growth was more profound than anti-CD4 antibody. Collectively, our data suggest that T cells, especially the CD8⁺ T cells, indeed mediated the effects of PRMT3 KO or PRMT3 inhibition on HCC progression.”

8. Please provide direct evidence to support the causative role of HSP60 oligomerization in anti-tumor immunity. Likewise, please demonstrate whether the HSP60 mutant with defective oligomerization exhibits the same phenotypes observed in the methylation-defective mutant.

We thank this reviewer for the insightful comments. To determine a causative effect of HSP60 oligomerization in anti-tumor immunity, we decided to focus on HSP60 D3G mutation, which was previously reported to inhibit HSP60 oligomerization in several studies.^{6,7} To examine the role of HSP60 oligomerization in cGAS/STING signaling and anti-tumor immunity and whether the HSP60 mutant with defective oligomerization exhibited the same phenotypes observed in the methylation-defective mutant, we compared the effect of HSP60-D3G overexpression and HSP60-R446K overexpression in HSP60-KD cells on mtROS, mtDNA release, cGAMP level, IFN β targets and IFN β production. We found that the HSP60-D3G mutant with defective oligomerization displayed the same phenotypes as the methylation-defective HSP60-R446K mutant. We have added this to the Results (**Page 21 line 441- line 457**) and Discussion section (**Page 28 line 601- line 607**).

Results (**Page 21 line 441- line 457**): “To examine the role of HSP60 oligomerization in cGAS/STING signaling and anti-tumor immunity and whether the HSP60 mutant with defective oligomerization exhibited the same phenotypes observed in the methylation-defective mutant. We examined whether HSP60-D3G,⁴¹ a mutant with defective oligomerization, was involved in immune modulation by comparing the effect of HSP60-D3G overexpression and HSP60-R446K mutant overexpression in HSP60-KD cells on mtROS, mtDNA release, cGAMP level, IFN β targets, and IFN β production. We found that both HSP60-D3G and HSP60-R446K overexpression could not reverse the effect of HSP60-KD on mitoROS and membrane potential (**Supplementary Fig. 10A, B**). Similarly, both HSP60-D3G and HSP60-R446K overexpression could not attenuate cytosolic mtDNA leakage and decrease the cGAMP level induced by HSP60-KD in HCC cells (**Supplementary Fig. 10C, D**). Also, Similarly, HSP60-KD led to activation of the

expression of ISGs and increased IFN β production, which could not be reversed by either HSP60-D3G and HSP60-R446K overexpression (**Supplementary Fig. 10E, F**). Importantly, we found that HSP60-D3G and HSP60-R446K overexpression had comparable effects on mtROS, membrane potential, mtDNA leakage, cGAMP level, ISGs expression, and IFN β production (**Supplementary Fig. 10**). Collectively, our data suggest that PRMT3 inhibition activated anti-tumor immunity by impairing HSP60 oligomerization in HCC.”

Discussion (**Page 28 line 601-line 607**): “Moreover, we demonstrated that HSP60-D3G, a mutant with defective oligomerization,⁴¹ activated mtDNA-cGAS/STING-IFN β pathway in HCC, and HSP60-D3G and HSP60-R446K overexpression had comparable effects on mtROS, membrane potential, mtDNA leakage, cGAMP level, ISGs expression and IFN β production. Thus, these data suggested that HSP60 oligomerization was directly involved in anti-tumor immunity and PRMT3 inhibition activated anti-tumor immunity by impairing HSP60 oligomerization.”

9. To support the authors' conclusion of immune dependency, authors should evaluate the tumor growth upon HSP60 knockdown (Figure 5B) in immune-compromised mice (e.g., NSG).

We thank this reviewer for the constructive comments. We have examined the effect of HSP60 knockdown on tumor growth in immune-compromised NSG mice. We found that HSP60 knockdown also inhibited HCC growth in Hepa1-6 tumor bearing NSG mice (**Supplementary Fig. 8B-D**), consistent with previous findings that HSP60 plays an oncogenic role in various cancer types.^{8,9} However, we found that HSP60-KD more profoundly delayed tumor progression in immune-competent mice than in immune-deficient mice (**Supplementary Fig. 8B-G**). Therefore, in addition to the cancer cell-intrinsic functions of HSP60, anti-tumor immunity also contributed to the observed delay in tumor growth in HSP60-KD cells. We have included this in the Results section (**Page 16 line 334-line 336**).

Results: “HSP60-KD delayed tumor progression in both immune-deficient and immune-competent mice, and its effects were more profound in immune-competent mice than in immune-deficient mice” (**Page 16 line 334-line 336**).

10. Please display the activation of STING (phosphorylation, oligomerization) in the immunoblots shown in Figure 6. Reassess whether the "+/-" signs are appropriately aligned in Figure 6H.

We thank this reviewer for the comments. As requested, we have included data showing the activation of STING using anti-phospho-STING antibody, a well-established method for examining the activation of cGAS/STING signaling as shown in previous publications.¹⁰⁻¹⁵ We examined the phosphorylation of STING (p-STING) and total STING in the indicated groups and found that PRMT3-KO and HSP60-R442K mutant significantly increased the phosphorylation of STING. Since STING oligomerization occurs before STING phosphorylation, we did not examine STING oligomerization in our experiments. Also, we have checked the "+/-" signs and they are appropriately aligned in Figure 6H in the current version. We have added a detailed description to the Results section (**Page 19 line 407- page 20 line 417**).

Results (**Page 19 line 407- page 20 line 417**): “We found that PRMT3-KO or PRMT3 inhibition increased phosphorylation of TBK1, IRF3 and STING, key effectors in the cGAS/STING pathway, in Hepa1-6 cells and PLC-8024 cells (**Fig. 6F, Supplementary Fig. 9I**). Importantly, PRMT3 inhibitor treatment of *Myc/Trp53^{-/-}* mice also led to a drastic increase of phosphorylation of TBK1, IRF3 and STING in the spontaneous HCC tumor samples, accompanied by a dramatic decrease in ADMA that confirmed the effectiveness of PRMT3 inhibition in vivo (**Fig. 6G**). Similarly, HSP60-KD led to activation of cGAS/STING signaling as shown by increased phosphorylation of TBK1, IRF3 and STING (**Fig. 6H, I**), which was reversed by overexpression of HSP60-WT but not HSP60-R446K mutant.”

11. Provide the PRMT3 expression in different cell types in the tumor niche of HCC from

publicly available scRNAseq data. Given the comparable effects of global (pharmacologic) and genetic (tumor cell-specific) inhibition of PRMT3 on tumor growth and therapeutic outcomes, discuss the potential influence of global PRMT3 inhibition on other cell types in the tumor niche, including immune cells and stroma cells.

As requested, we have analyzed several publicly available HCC scRNA-seq datasets (GSE149614, GSE156625 and GSE202642)¹⁶⁻¹⁸ to examine the expression of PRMT3 in different cell types. We also examined PRMT3 expression across different cell types using our immunofluorescence data. We found that PRMT3 was mainly expressed in epithelial cells and malignant cells in tumor niche. PRMT3 expression in immune cells and stroma cells was much less than tumor cells (**Supplementary Fig. 14, Fig.1E**). Since we observed comparable effects of global (pharmacologic) and genetic (tumor cell-specific) inhibition of PRMT3 on tumor growth and response to immune checkpoint therapy, the effects of pharmacologic inhibition of PRMT3 on immune cells and stroma cells may play a minor role in the response of HCC to immune checkpoint therapy. However, it is possible that PRMT3 inhibition in stromal cells and immune cells may impact other cellular processes, such as cell migration, invasion, and metastasis. To address this question, we will compare the single-cell transcriptomes of PRMT3-KO tumors and PRMT3 inhibitor-treated tumors in our future study. We have added this to the Discussion section (**Page 27 line 571- line 583**).

12. Considering a recent publication suggesting that PRMT3 represses the innate immune response in viral infection (PMID: 37639603), authors should determine and discuss the possibility of PRMT3's involvement in the methylation of cytosolic RNA and DNA sensors, such as RIG-I, MDA5, and cGAS.

We thank this reviewer for the comments. As requested, we have explored the possibility that cytosolic RNA and DNA sensors, such as RIG-I, MDA5, and cGAS, can be methylated by PRMT3. We first examined whether PRMT3 binds to RIG-I, MDA5, and cGAS in HCC cells. We examined our IP-LC/MS data generated from HCC cells, and we did not identify

RIG-I, MDA5, and cGAS as potential PRMT3 interacting proteins. Also, using a comparable immunoprecipitation condition as the PNAS paper,¹⁹ we performed Co-IP and found that endogenous PRMT3 could not interact with RIG-I, MDA5, and cGAS (**Additional Fig. 2**). Thus, RIG-I, MDA5, and cGAS may not be PRMT3 substrates and may not play a role in the regulation of type I interferon signaling inhibition by PRMT3 in HCC. We have added this to the Discussion section. (**Page 26 line 563- page 27 line 569**)

Discussion: “Moreover, a recent publication suggested that PRMT3 represses the innate immune response in viral infection by methylating cytosolic RNA and DNA sensors, such as RIG-I, MDA5, and cGAS.⁴⁹ However, RIG-I, MDA5, and cGAS were not identified in the IP-LC/MS. We also performed Co-IP and found that endogenous PRMT3 could not interact with RIG-I, MDA5, and cGAS (Data not shown). Thus, RIG-I, MDA5, and cGAS might not be substrates of PRMT3 and are unlikely to mediate the function of PRMT3 in the inhibition of type I interferon signaling in HCC.” (**Page 26 line 563- page 27 line 569**)

Additional Fig. 2. WB analysis showed that endogenous PRMT3 did not interact with MDA5, cGAS and RIG-1 in Hepa1-6 cells using reciprocal co-immunoprecipitation.

Reviewer #2 (ICB/cancer immune therapy)(Remarks to the Author):

The paper by Shi et al describes a role for an IFN γ -induced methyltransferase (PRMT3) in acquired resistance to immunotherapy. Expression of this methyltransferase correlated with T cell infiltration and clinical outcome in HCC patients and tissue slide staining confirmed this on protein level in two HCC patient cohorts receiving checkpoint therapy. PRMT3 is upregulated by IFN γ and prevented T cell infiltration in mouse HCC tumor models. Extensive mechanistic experiments using gene knockdowns and chemical inhibitors in tumor cells demonstrated the pathway of PRMT3-> HSP60 oligomerization in mitochondria-> prevention of cGAS/STING activation->prevention of T cell infiltration. In addition, re-expression studies of WT genes or mutant genes confirmed the findings. Finally, PRMT3 inhibition resulted in much stronger growth delay in immunocompetent mice compared to immunodeficient mice and sensitizes tumors for checkpoint therapy. The specific inhibitor SGC707 represents an ideal inhibitor of this immune resistance pathway and ready seems for clinical use. PRMT3 is suggested to block a feedforward accumulation of IFN γ -induced tumor immunity, in which initial responses are hampered but, moreover, inhibition of this molecule even can provoke inflammation in cold tumors.

The study is well organized, well presented and extremely well controlled. Conclusions are supported by the data. Discussion includes open questions and unresolved issues.

We thank this reviewer's insightful and constructive comments and the recognition of the strength of our manuscript. We have addressed this reviewer's concerns by including additional data and detailed description of results and methods in the revised manuscript (see below).

Comments:

1. HCC can have a viral cause or can be sporadic. Does the effect of PRMT3 mainly play a role in the virus-induced cancers? Patient databases can be analysed for this.

We thank this reviewer for the comments. Our new analysis (see below) suggests that PRMT3 plays a similar role in virus-induced HCC and alcohol-related HCC.

In SYSUCC cohort, the majority of the patients (90%) were HBV-related HCC. As shown in Fig. 1D and Supplementary Fig. 3, PRMT3 expression was associated with poor prognosis in HCC in SYSUCC cohort. We also analyzed the correlation between PRMT3 expression and the prognosis of HBV-related HCC in SYSUCC cohort. As expected, high PRMT3 expression was also correlated with poor prognosis of HBV-related HCC. However, the majority of the patients (>90%) were alcohol-related HCC in TCGA-LIHC dataset. As shown in Supplementary Fig. 2A, B, PRMT3 expression was associated with poor prognosis in HCC in the TCGA-LIHC cohort.

2. Is there a reason why this resistance mechanism is only functional in HCC? Correlations between T cell profiles and PRMT3 levels (fig 1) can be analysed in other cancer types, including checkpoint sensitive cancers.

We thank this reviewer for the insightful and constructive comments. Based on the correlation between T cell profiles and PRMT3 levels in other cancer types (see below), this resistance mechanism may be functional in other cancer types, which will be explored in our future studies.

We analyzed correlations between T cell profiles and PRMT3 levels in other cancer types using Timer 2.0 database²⁰ and also found a negative correlation between PRMT3 expression and CD8⁺ T cell infiltration, including in immunotherapy-sensitive cancer types (SKCM and KIRC) (**Additional Fig. 3**). Therefore, the effects of PRMT3 on immune infiltration might be a general mechanism across cancer types. Previous studies also proved that PRMT3 was involved in tumor progression and therapy resistance in several cancer types, including colorectal cancer, pancreatic cancer, endometrial carcinoma, and glioblastoma.²¹⁻²⁴

Additional Fig 3. Associations of PRMT3 expression with CD8+ T cell infiltration evaluated by CIBERSORT in TCGA-SKCM and TCGA-KIRC dataset.

3. Fig 1: correlation between cytotoxic signature score and PRMT3 expression is rather poor, especially the PRMT3 low tumors display seem to allow T cell infiltrate (fig 1b-c). In the other panels, samples are divided between PRMT3 low and high tumors. Were all samples included in these analyses of fig 1d-k? Were samples allocated based on median? Or were only extremes taken for PRMT3 expression in tumors?

We thank this reviewer for the comments. Although the correlation between CD8+ T cells and PRMT3 expression using CIBERSORT has an R score of -0.153, which is lower than that from ImmuneCell AI, it has a highly significant P value ($P=4.51e-03$). The difference in the R values could be due to the different methods used in the deconvolution of cell types using bulk RNA-seq data, and each of these algorithms has its own limitations. To further strengthen our conclusion, we further analyzed several public datasets and found a negative correlation between PRMT3 expression and T cell infiltration in melanoma and Glioblastoma using the Tumor Immune Dysfunction and Exclusion (TIDE database)²⁵ (Supplementary Fig. 2C). We have added these findings to the Results section (Page 6 line 104- line 107).

Also, we have provided detailed information regarding the correlation of PRMT3

expression with HCC prognosis (**Fig. 1D**), immune infiltration (**Fig. 1E-F**), and response to immunotherapy (**Fig. 1H-K**) in the SYSUCC cohort. For Fig. 1D, a total of 20 samples were used in the analysis. In Figures 1E and 1F, 20 samples in each group were used for the analysis of immune infiltration of HCC samples in the SYSUCC cohort by multi-immunofluorescence. In Figure H-K, 30 and 33 patients in 2 independent cohorts who received immunotherapy were enrolled for analyzing the correlation between PRMT3 expression (determined by IHC) and treatment response. All samples were allocated into PRMT3-high and PRMT3-low groups based on the median of PRMT3 mRNA expression or IHC combined score (also see above for response to comments # 1 for reviewer 1). We also included the expression levels of PRMT3 for each patient's group in our Source Data.

4. Figure 2 shows IFN γ -induced expression of PRMT3 and that (obviously) STAT1 is indispensable for this. However, type I interferons also transduce signals via STAT1, do they also induce expression of PRMT3?

We thank this reviewer for the comments. Our new data showed that type I interferons have a subtle effect on the upregulation of PRMT3 expression in HCC cells, which has been added to the Results section (**Page 10 line 200- line 204**). We speculate that the difference between IFN γ and type I interferon on PRMT3 induction could be due to the difference in the active STAT1 transcription complex: IFN γ signaling induced nuclear localization of homodimer of STAT1 (STAT1:STAT1) whereas type I interferons induced nuclear localization of STAT1/STAT2 heterodimer (STAT1:STAT2)²⁶, which might be not able to bind to PRMT3 promoter and transcriptionally regulate PRMT3 expression effectively. The mechanisms behind the different effects between IFN γ and type I interferon on PRMT3 induction will be examined in our future studies.

Results: "Since type I interferons (e.g., IFN α) also transduce signals via STAT1, we examined the effect of IFN α on PRMT3 expression in PLC-8024 and Hepa1-6 cells and found that IFN α slightly up-regulated PRMT3 (Supplementary Fig. 4M). These results suggested that PRMT3 were mainly induced by IFN γ secreted by effector CD8⁺ T cell in

the tumor immune microenvironment.” (Page 10 line 200- line 204)

5. Correlations with T cell influx after PRMT3 inhibition are consistent throughout the study, but fig 3 also shows a dramatic shift in B cell influx in PRMT3-knockout tumors. RAG knockout mice were used to show involvement of the immune system, but these mice also lack B cells. A direct role for T cells should be demonstrated with antibody (aCD4 and aCD8) depletion experiments during treatment in a mouse tumor model.

We thank this reviewer for the insightful comments. As requested, we have provided new data showing a direct role for T cells in the response of PRMT3-KO tumors or tumors treated with PRMT3 inhibitor SGC707. Since PRMT3-KO or inhibition significantly increased T cells infiltration, especially CD8⁺ and CD4⁺ T cells, we treated tumor-bearing mice with anti-CD4 or anti-CD8 neutralizing antibodies and found that neutralization of CD8⁺ or CD4⁺ T cells indeed attenuated tumor growth inhibition of PRMT3-KO tumors or tumors treated with PRMT3 inhibitor. Of note, neutralizing CD8⁺ T cells have a more profound effect than neutralizing CD4⁺ T cells, suggesting that T cells, particularly CD8⁺ T cells, largely mediated the effects of PRMT3 on anti-tumor immunity in HCC. We have added the detail description in the Result section of this revised manuscript (Page 12 line 253- page 13 line 261).

Results: “Given the significant changes in T cell abundance and activation by *Prmt3*-KO and PRMT3 inhibition, we then examined whether the effects of *Prmt3*-KO and PRMT3 inhibition on tumor progression were mediated by CD8⁺ or CD4⁺ T cells. We treated tumor-bearing mice with anti-CD3, anti-CD4, or anti-CD8 antibody and found that neutralization of CD3⁺, CD8⁺ and CD4⁺ T cells indeed attenuated the tumor growth inhibition induced by *Prmt3*-KO (Supplementary Fig. 6A-F). Of note, the effect of anti-CD8 antibodies on tumor growth was more profound than anti-CD4 antibody. Collectively, our data suggest that T cells, especially the CD8⁺ T cells, indeed mediated the effects of PRMT3 KO or PRMT3 inhibition on HCC progression.”

Although we demonstrated that the activation of T cell-dependent anti-tumor immunity mediates in large part the effects of PRMT3-KO or PRMT3 inhibition in HCC progression in multiple model systems, we also observed a significant increase in the total number of B cells and a significant decrease in monocytes and macrophages in Prmt3-KO Hepa1-6 tumors (Fig. 3V). We will examine whether B cells, monocytes, and macrophages play a role in the response of HCC to immunotherapy in our future studies as described in the Discussion section (**Page 29 line 611- line 618**).

6. Fig 6 and 7 show upregulation of some STING targets as a result of PRMT3 shutdown, but the relevant CXCL9/10/11 chemokines, which are often responsible for T cell attraction to tumors, were not tested. This is important since the final result is more T cell influx.

We thank this reviewer for the insightful comments. We have tested the expression of CXCL9/10/11 in PRMT3-KO cells, PRMT3 inhibitor-treated cells, and the corresponding control cells. We found that PRMT3-KO and SGC707 treatment increased the expression of CXCL9/10/11 in PLC-8024 cells and Hepa1-6 cells, respectively (Supplementary Fig. 9J, K). We also observed an increase in CXCL9/10/11 in tumors from *Myc/Trp53^{-/-}* mice treated with PRMT3 inhibitor (Fig. 6K). Similarly, HSP60-KD also led to an increase in the expression of CXCL9/10/11, which was reversed by HSP60-WT OE but not by HSP60-R446 OE (Fig. 6M). Importantly, HSP60 WT or HSP60-R446 mutant OE in PRMT3-KO cells led to further activation of CXCL9/10/11, which were dramatically suppressed when PRMT3 was overexpressed (Fig. 6O). These findings suggest that PRMT3 expression and HSP60-R446 methylation were required for inhibition of cGAS/STING pathway. These new findings have been included in the Results section. (**Page 20 line 421-line 424**)

Results: "We also observed an increase in interferon-stimulated genes (ISGs) (e.g., *Cxcl9*, *Cxcl10*, *Cxcl11* and IFN β production in tumors from *Myc/Trp53^{-/-}* mice treated with PRMT3 inhibitor, an in PRMT3-KO mouse and human HCC cells (Hepa1-6 and PLC-8024) (Fig. 6K-L, Supplementary Fig. 9J-L)." (**Page 20 line 421-line 424**)

7. In their previous paper on PRMT3 (ref 13, Nat Comm 2023) authors show IGF2BP1 as a downstream target after oxaliplatin treatment. Why was this molecule not found among the pull-down proteins in this study? One would expect similar targets after high expression of PRMT3, unless this enzyme itself is modified differently depending on the type of treatment?

We thank this reviewer for the comments. We would like to clarify that IGF2BP1 was also identified as one of the proteins that were pulled down by PRMT3 in mouse HCC cell line Hepa1-6, but it was not among the top candidates. In contrast, IGF2BP1 was among the top 6 candidates for PRMT3-interacting proteins in PLC-8024 and HepG2 cells in our previous study. Although IGF2BP1 was also reported to regulate response to immunotherapy,²⁷⁻²⁹ we analyzed the correlation between CD8⁺ T cells infiltration and IGF2BP1 and HSPD1 and found that IGF2BP1 expression was not negatively correlated with CD8⁺ T cells infiltration (**Additional Fig. 4**). This result suggested that IGF2BP1 might not be involved in the suppression of T cell infiltration and T cell-mediated anti-tumor immunity by PRMT3 in HCC, but instead plays a role in the resistance to oxaliplatin as we reported previously.⁴ However, we cannot completely rule out the possibility that IGF2BP1 also plays a role in anti-tumor immunity through mechanisms distinct from the PRMT3/HSP60-mediated STING/cGAS suppression. We have included this information in the Discussion section (**Page 26 line 552-line 559**).

Additional Fig 4. Associations of IGF2BP1 and HSPD1 expression with CD8⁺ T cell infiltration evaluated by CIBERSORT in TCGA-LIHC dataset.

Reviewer #3 (HCC) (Remarks to the Author):

The authors reported that PRMT3 acts as a novel driver of immunotherapy resistance in HCC. They first found that PRMT3 expression was induced by activated T cells in response to ICB via IFN- γ -STAT1 signaling pathway, and that high PRMT3 expression inversely correlates with tumor-infiltrating CD8⁺ T cells and predicts poor response to ICB in HCC patients. Mechanically, they revealed that PRMT3 methylated the mitochondrial chaperone protein HSP60 at R446 to induce its oligomerization for maintenance of mitochondrial homeostasis in tumor cells. Loss of PRMT3 caused mitochondria damage to release mtDNA into cytosol to trigger cGAS-STING activation and subsequent antitumor immunity. PRMT3 inhibition further enhanced the efficacy of ICB treatment anti-PD1 efficacy in HCC mouse models. Overall, the findings reported are intriguing, however, addressing the points below is necessary to strengthen and validate the proposed model and increase the accuracy and reliability of the results.

We thank this reviewer for recognizing the novelty of our study and the insightful and constructive comments. We have completely addressed all the concerns as detailed below.

My specific comments are listed here below.

Major points

1. More clinical HCC tumor samples should be included for WB detection of PRMT3 in Fig S2A, and Fig 2B and 2C.

As requested, we have included more clinical HCC tumor samples in the WB analysis of PRMT3 expression. Consistent with our previous results, PRMT3 was significantly up-regulated in tumor tissues and anti-PD1 treated HCC samples compared to normal and tumor tissues from patients without anti-PD1 treatment, respectively (Supplementary Fig. 2A and supplementary Fig. 3A, B).

2. In Fig S3A, the authors showed that *Myc/Trp53*^{-/-} tumors had very few CD8⁺ T cell infiltration levels, which was responsible for the failure in PRMT3 elevation after anti-PD1 treatment. However, they only conducted immunofluorescence to examine CD8 T cell infiltration. Flow cytometry analysis should be included. In addition, according to the paper (PMID: 34290403 DOI: 10.1038/s41586-021-03741-7), both Hepa1-6 and *Myc/Trp53*^{-/-} mouse liver cancer had basal infiltration of CD8 T cells, and combination treatment of Lenvatinib and gefitinib further increased influx CD8⁺ T cells in these tumors. Why the authors obtained the inconsistent results and reached the conclusion that *Myc/Trp53*^{-/-} are immune “cold” tumor here? The author may deplete CD8 T cells to further examine their effects on PRMT3 expression in Hepa1-6 tumor bearing mouse models.

We thank this reviewer for the comments. As requested, we have included flow cytometry analysis of CD8⁺ T cell infiltration and found that CD3⁺CD8⁺ T cell infiltration in *Myc/Trp53*^{-/-} tumors was significantly less than Hepa1-6 subcutaneous tumor (Supplementary Fig. 4D). Also, we would like to clarify why we stated that HCCs from *Myc/Trp53*^{-/-} are immune “cold” tumors. We agree with this reviewer that both Hepa1-6 and *Myc/Trp53*^{-/-} mouse liver cancer had basal infiltration of CD8 T cells. Indeed, we also detected CD8⁺ T cell infiltration in *Myc/Trp53*^{-/-} tumors, which accounts for lower than 10% of the total infiltrated immune cells (Fig. 3T, Fig. 6J, and Fig. 7Q), which was consistent with previous results (~5%) (PMID: 34290403 DOI: 10.1038/s41586-021-03741-7). However, CD8⁺ T cell infiltration in Hepa1-6 tumor was more than 20% as we showed in Figures 3, 5, 7, 8, which was much more than the *Myc/Trp53*^{-/-} tumors. That is why we consider *Myc/Trp53*^{-/-} tumors as relatively “cold” tumors compared to Hepa1-6 tumors. Regardless, we have made changes to our description of the results to reflect such a comparison. **(Page 11 line 234, Page 12 line 236, Page 24 line 504).**

Also, we treated mice with the neutralizing anti-CD4 or anti-CD8 antibodies and found that neutralization of CD8⁺ and CD4⁺ T cells indeed attenuates tumor growth inhibition. Of note, neutralizing CD8⁺ T cells has a more profound effect than neutralizing CD4⁺ T cells, suggesting CD8⁺ T cells play a major role in the effect of PRMT3 KO or PRMT3 inhibition. These results strongly support the notion that T cells, particularly CD8⁺ T cells largely

mediated the effects of PRMT3 on anti-tumor immunity in HCC. We have added these findings to the Result section (**Page 12 line 253- page 13 line 261**).

Results: “Given the significant changes in T cell abundance and activation by *Prmt3*-KO and PRMT3 inhibition, we then examined whether the effects of *Prmt3*-KO and PRMT3 inhibition on tumor progression were mediated by CD8⁺ or CD4⁺ T cells. We treated tumor-bearing mice with anti-CD3, anti-CD4, or anti-CD8 antibodies and found that neutralization of CD3⁺, CD8⁺, and CD4⁺ T cells indeed attenuated the tumor growth inhibition induced by *Prmt3*-KO (Supplementary Fig. 6A-F). Of note, the effect of anti-CD8 antibodies on tumor growth was more profound than anti-CD4 antibody. Collectively, our data suggest that T cells, especially the CD8⁺ T cells, indeed mediated the effects of PRMT3 KO or PRMT3 inhibition on HCC progression.”

3. IFN γ and TNF α are cytokines that are released by effector CD8⁺ T cells. They should also examine the effects of TNF α on PRMT3 expression. The authors used IFN γ to directly treat tumor cells for detecting PRMT3 expression. They may add the indicated inhibitors of IFN γ and TNF α into CD8⁺ T-cell supernatant and check tumor intrinsic PRMT3 expression after incubation.

We thank this reviewer for the comments. As requested, we have examined the effects of TNF α on PRMT3 expression in PLC-8024 and Hepa1-6 cells. We found that TNF α failed to induce PRMT3 expression at both 24 and 48 hours (Supplementary Fig. 4E). Also, we further investigated the effect of IFN γ and TNF α on PRMT3 expression by adding inhibitors of IFN γ (IFN- γ Antagonist 1 acetate)³⁰ and TNF α (R-7050)³¹ into CD8⁺ T-cell supernatant and check tumor intrinsic PRMT3 expression in PLC-8024 and Huh7 cells after incubation. We found that PRMT3 induction was significantly attenuated by the addition of inhibitor of IFN γ , but not inhibitor of TNF α , to the CD8⁺ T-cell supernatant (Supplementary Fig. 4H). These results suggested that IFN γ , but not TNF α , was secreted by CD8⁺ T-cell to induce tumor intrinsic PRMT3 overexpression in HCC. We have added these new findings into the Results section (**Page 9 line 180- line 186**).

Results: “To further investigate the effect of IFN γ and TNF α on PRMT3 expression, we added inhibitors of IFN γ (IFN γ antagonist 1 acetate)²⁷ and TNF α (R-7050)²⁸ into CD8⁺ T-cell supernatant and checked PRMT3 expression in PLC-8024 cells after incubation. PRMT3 induction was significantly attenuated with the inhibitor of IFN γ but not TNF α inhibitor (Supplementary Fig. 4H). These results suggested that IFN γ , but not TNF α , secreted by CD8⁺ T-cell induced PRMT3 expression in HCC.”

4. In Fig 4K, why Prmt3-KO or PRMT3 inhibition notably decreased the oligomerization of HSP60, but failed to increased HSP60 monomers? In Fig 5I and J, the tumor volume of PRMT3+ HSP60-WT seems to be comparable to that of PRMT3+ HSP60-RK, and in line 321-325, the description is logically mistaken. The conclusion “These findings suggest that other PRMT3 substrates may also contribute to the effects of PRMT3-OE on HCC growth and T cell infiltration” is not that appropriate. These results seemed to suggest that PRMT3 arginine methylation of HSP60 is required for the promoting HCC growth, since the tumor growth and T cell infiltration of HSP60-WT and HSP60-RK were both at comparable levels in the absence of PRMT3.

We thank this reviewer for the comments. In Fig. 4K, the lack of notable decrease in HSP60 monomer when HSP60 oligomerization was reduced by Prmt3-KO cells or PRMT3 inhibition can be explained by the very high level of HSP60 expression in PLC-8024 cells. We found that the HSP60 oligomerization form accounted for only about 8% of the total HSP60 in PLC-8024 cells. Therefore, the increased HSP60 monomers in *Prmt3*-KO or PRMT3 inhibition groups were not obvious in our Western blot analysis in Fig 4K. We further evaluated the effects of PRMT3 on oligomers and monomers of HSP60 in Hepa1-6 cells, a cell line with a lower expression level of HSP60. Consistent with the result in HEK293T, the *Prmt3*-KO or PRMT3 inhibition notably decreased the oligomerization of HSP60 and increased HSP60 monomers in Hepa1-6 cells (Supplementary Fig. 7L).

Also, we agree with the reviewer on the possible problem for our description of the results in Fig 5I, J, and K. The tumor volumes and weights of PRMT3+ HSP60-WT group were

higher than PRMT3+ HSP60-RK group, which suggested that the effect of HSP60-WT on tumor growth was more profound than HSP60-RK in the presence of PRMT3. We have deleted the conclusion “These findings suggest that other PRMT3 substrates may also contribute to the effects of PRMT3-OE on HCC growth and T cell infiltration”. Also, the tumor growth and T cell infiltration of HSP60-WT and HSP60-RK were comparable in the absence of PRMT3. These results indeed suggested that PRMT3 was required for HSP60’s function to promote HCC growth, which is consistent with our conclusion that arginine methylation of HSP60 at R442 is crucial for its tumor-promoting and immunoinhibitory functions in HCC (Figure 5-6).

5. In Fig 6C-E, the author attempted to demonstrate that lack of PRMT3-HSP60 caused mtDNA leakage into cytosol, but the confocal images were not clearly clarified. First, it is impossible that mtDNAs were not detected in mitochondria of WT cells. It is well-acknowledged that TOM20 indicates mitochondrion but not mtDNA. TFAM is the specific mtDNA binding protein, which should be stained to precisely indicate the locations of dsDNA (PMID: 26305956, PMID: 38168624).

We thank this reviewer for the constructive comment. First, we would like to clarify our IF staining results. In Figure 6C-E of our previous manuscript, we used anti-dsDNA (green) and anti-Tom20 (red) to demonstrate mtDNA leakage in HCC cells. The mtDNA was detected in the mitochondria of all groups. However, the majority of mtDNA (green) in the cytosol was co-localized with the mitochondria (red), especially, in the WT cells. Therefore, mtDNA in mitochondria could not be easily recognized in the confocal images due to the more pronounced red color. The dsDNA which leaked into the cytosol was not co-localized with mitochondria. Thus, the green foci represented the mtDNA leaked from mitochondria (red) in the confocal images.

Also, we have followed the reviewer’s suggestion and used antibodies for mtDNA binding protein TFAM to precisely indicate the locations of dsDNA in our IF staining. We used anti-TFAM antibody and mitotracker to examine mtDNA leakage using IF staining. We

found that Prmt3 KO or PRMT3 inhibition led to an increase in mtDNA leakage (Fig. 6C, supplementary Fig. 9C). Similarly, HSP60 KD also led to an increase in cytosolic mtDNA leakage, which was reversed by the overexpression of HSP60 WT but not HSP60 R446K mutant (Fig. 6D). Furthermore, to establish the significance of PRMT3-dependent methylation of HSP60 on mitochondrial DNA leakage, we examined the effect of co-expression of PRMT3 with HSP60-WT and R446K mutant in *Prmt3*-KO Hepa1-6 cells. We found that the co-expression of PRMT3 with HSP60-WT but not with HSP60-R446K mutant in *Prmt3*-KO cells effectively prevented mtDNA leakage (Fig. 6E). Thus, our data suggest that HSP60-R446 methylation by PRMT3 was essential for inhibiting mtDNA leakage. We have added these findings to the Results section (**Page 18 line 386- page 19 line 399**).

6. The authors only examined pTBK1 and pIRF3 to indicate cGAS activation. cGAMP production, the direct readout of cGAS activation, should be evaluated after inhibition of PRMT3-HSP60. Alternatively, the authors may compare the differences between cGAS-KD and WT cells to highlight the importance of cGAS in this paper.

We thank this reviewer for the comments. As suggested, we used cGAMP production as the direct readout of cGAS activation. We found that *PRMT3*-KO or PRMT3 inhibition increased cGAMP production in Hepa1-6 and PLC-8024 cells (Supplementary Fig. 9D-F). Similarly, *HSP60*-KD led to increased cGAMP production, which was reversed by overexpression of HSP60-WT but not HSP60-R446K (Supplementary Fig. 9G). Importantly, HSP60 WT or mutant OE in *PRMT3*-KO cells led to a further increase of cGAMP level, which can be dramatically suppressed when PRMT3 was overexpressed (Supplementary Fig. 9H). We have added these findings to the Results section (**Page 19 line 400-line 407**).

Also, as suggested, we have examined the effect of cGAS-KD on the growth of PRMT3-KO cells in vivo to highlight the important role of cGAS in PRMT3-driven tumor progression. Since PRMT3 inhibition activated cGAS/STING signaling through mtDNA release, we decided to examine whether cGAS/STING hyperactivation directly contributes to the delayed tumor progression and increased T cell infiltration observed in PRMT3-KO tumors.

We found that cGAS-KD combined with PRMT3-KO markedly reduced the effects of PRMT3, led to bigger tumor volumes and weights (Fig. 7O, P) and a significant reduction in T cell infiltration (CD4⁺ T cells, CD8⁺ T cells, IFN γ ⁺ CD8⁺ T cells, and GZMB⁺ CD8⁺ T cells) compared to the SGC707-treated group (Fig. 7Q-R). Moreover, cGAS-KD notably inhibited the expression of ISGs and IFN β production induced by PRMT3 inhibition (Fig. 7S, T). Thus, we demonstrated that cGAS/STING hyperactivation directly contributes to the delayed tumor progression and increased T cell infiltration observed in PRMT3-KO tumors. We have added these findings to the Results section (**Page 22 line 475- page 23 line 482**).

7. It has been reported that PRMT3 directly interacts with cGAS and catalyzes its asymmetric demethylation (PMID: 37639603). It will be a good supplementation to this work if the authors evaluate the effects of PRMT3 on cGAS activity. And the authors may further evaluate the effects of PRMT3 inhibitor on antitumor immunity in WT and cGAS-KD tumor models.

We thank this reviewer for the comments. We agree with the reviewer that the mechanism underlying PRMT3's function in antiviral immunity found in the recent publication (PMID: 37639603) could be relevant to our studies. To examine whether cGAS, RIG-I, and MDA5, three PRMT3-interacting proteins identified in the aforementioned study, are also PRMT3-interacting proteins in HCC cells, we examined our IP-LC/MS data and RIG-I, MDA5, and cGAS were not identified in our data. To further confirm our IP-LC/MS results, we performed Co-IP and found that endogenous PRMT3 could not interact with RIG-I, MDA5, and cGAS (Additional Figure 1). Thus, RIG-I, MDA5, and cGAS may not be substrates of PRMT3 in HCC cells. We have added these findings to the Discussion section (**Page 26 line 563- line 569**).

As suggested, we further determined the effects of PRMT3 inhibition using PRMT3 KO on the anti-tumor immunity in cGAS-WT and cGAS-KD tumors. We found that cGAS-KD combined with PRMT3-KO markedly reduced the effects of PRMT3 KO on anti-tumor

immunity, led to bigger tumor volumes and weights (Fig. 7O, P) and a significant reduction in T cell infiltration (CD4⁺ T cells, CD8⁺ T cells, IFN γ ⁺ CD8⁺ T cells, and GZMB⁺ CD8⁺ T cells) compared to the cGAS-WT/PRMT3-KO tumors (Fig. 7Q-R). Moreover, cGAS-KD notably inhibited the expression of ISGs and IFN β production induced by PRMT3 inhibition (Fig. 7S, T). Although we didn't examine the effect of PRMT3 inhibitor in this setting, we expect that KD cGAS in PRMT3 inhibitor-treated cells would have a similar effect as PRMT3 KO due to the highly similar phenotypes between PRMT3-KO tumors and PRMT3 inhibitor-treated tumors in our study. Thus, we demonstrated that cGAS/STING hyperactivation directly contributes to the delayed tumor progression and increased T cell infiltration observed in PRMT3-KO tumors. We have added these findings to the Results section (**Page 22 line 475- page 23 line 482**).

Additional Figure 2. WB analysis showed that endogenous PRMT3 did not interact with MDA5, cGAS and RIG-1 in Hepa1-6 cells using reciprocal co-immunoprecipitation.

Minor:

In Fig 6H, Figure labels should be corrected.

As requested, Figure labels in Fig 6H have been corrected in the current version.

In Fig 6I, does the authors mistake the labels "pTBK" and "TBK1"?

We thank this reviewer for the comments. We mislabeled the legends for "pTBK" and "TBK1" in Fig 6I and have corrected it in the current version.

In Line 369, "Fig. 7H, I" should be corrected to Fig. 6H, I.

As requested, "Fig. 7H, I" has been corrected to Fig. 6H, I in the current version.

Based on the comments above, my verdict of this study is "major revision".

Reference

- 1 Schwartz, L. H. *et al.* RECIST 1.1-Update and clarification: From the RECIST committee. *European journal of cancer (Oxford, England : 1990)* **62**, 132-137, doi:10.1016/j.ejca.2016.03.081 (2016).
- 2 Llovet, J. M. & Lencioni, R. mRECIST for HCC: Performance and novel refinements. *Journal of hepatology* **72**, 288-306, doi:10.1016/j.jhep.2019.09.026 (2020).
- 3 Murao, A., Aziz, M., Wang, H., Brenner, M. & Wang, P. Release mechanisms of major DAMPs. *Apoptosis : an international journal on programmed cell death* **26**, 152-162, doi:10.1007/s10495-021-01663-3 (2021).
- 4 Shi, Y. *et al.* PRMT3-mediated arginine methylation of IGF2BP1 promotes oxaliplatin resistance in liver cancer. *Nature communications* **14**, 1932, doi:10.1038/s41467-023-37542-5 (2023).
- 5 Montico, B., Nigro, A., Casolaro, V. & Dal Col, J. Immunogenic Apoptosis as a Novel Tool for Anticancer Vaccine Development. *International journal of molecular sciences* **19**, doi:10.3390/ijms19020594 (2018).
- 6 Parnas, A. *et al.* The MitCHAP-60 disease is due to entropic destabilization of the human mitochondrial Hsp60 oligomer. *The Journal of biological chemistry* **284**, 28198-28203, doi:10.1074/jbc.M109.031997 (2009).
- 7 Wang, J. *et al.* MitCHAP-60 and Hereditary Spastic Paraplegia SPG-13 Arise from an Inactive hsp60 Chaperonin that Fails to Fold the ATP Synthase β -Subunit. *Scientific reports* **9**, 12300, doi:10.1038/s41598-019-48762-5 (2019).
- 8 Kumar, R. *et al.* A mitochondrial unfolded protein response inhibitor suppresses prostate cancer growth in mice via HSP60. *The Journal of clinical investigation* **132**, doi:10.1172/jci149906 (2022).
- 9 Kumar, S. *et al.* Hsp60 and IL-8 axis promotes apoptosis resistance in cancer. *British journal of cancer* **121**, 934-943, doi:10.1038/s41416-019-0617-0 (2019).
- 10 Ka, N. L. *et al.* NR1D1 Stimulates Antitumor Immune Responses in Breast Cancer by Activating cGAS-STING Signaling. *Cancer research* **83**, 3045-3058, doi:10.1158/0008-5472.Can-23-0329 (2023).
- 11 Xiang, W. *et al.* Inhibition of ACLY overcomes cancer immunotherapy resistance via polyunsaturated fatty acids peroxidation and cGAS-STING activation. *Science advances* **9**, eadi2465, doi:10.1126/sciadv.adi2465 (2023).
- 12 Cho, M. G. *et al.* MRE11 liberates cGAS from nucleosome sequestration during tumorigenesis. *Nature* **625**, 585-592, doi:10.1038/s41586-023-06889-6 (2024).
- 13 Ghosh, M. *et al.* Mutant p53 suppresses innate immune signaling to promote tumorigenesis. *Cancer cell* **39**, 494-508. e495, doi:10.1016/j.ccell.2021.01.003 (2021).
- 14 Wang, K. *et al.* Gas therapy potentiates aggregation-induced emission luminogen-based photoimmunotherapy of poorly immunogenic tumors through cGAS-STING pathway activation. *Nature communications* **14**, 2950, doi:10.1038/s41467-023-38601-7 (2023).
- 15 Li, J. Y. *et al.* TRIM21 inhibits irradiation-induced mitochondrial DNA release

- and impairs antitumour immunity in nasopharyngeal carcinoma tumour models. *Nature communications* **14**, 865, doi:10.1038/s41467-023-36523-y (2023).
- 16 Zhu, G. Q. *et al.* CD36(+) cancer-associated fibroblasts provide immunosuppressive microenvironment for hepatocellular carcinoma via secretion of macrophage migration inhibitory factor. *Cell discovery* **9**, 25, doi:10.1038/s41421-023-00529-z (2023).
- 17 Sharma, A. *et al.* Onco-fetal Reprogramming of Endothelial Cells Drives Immunosuppressive Macrophages in Hepatocellular Carcinoma. *Cell* **183**, 377-394. e321, doi:10.1016/j.cell.2020.08.040 (2020).
- 18 Lu, Y. *et al.* A single-cell atlas of the multicellular ecosystem of primary and metastatic hepatocellular carcinoma. *Nature communications* **13**, 4594, doi:10.1038/s41467-022-32283-3 (2022).
- 19 Zhu, J. *et al.* Asymmetric arginine dimethylation of cytosolic RNA and DNA sensors by PRMT3 attenuates antiviral innate immunity. *Proceedings of the National Academy of Sciences of the United States of America* **120**, e2214956120, doi:10.1073/pnas.2214956120 (2023).
- 20 Li, T. *et al.* TIMER: A Web Server for Comprehensive Analysis of Tumor-Infiltrating Immune Cells. *Cancer research* **77**, e108-e110, doi:10.1158/0008-5472.Can-17-0307 (2017).
- 21 Zhang, X. *et al.* PRMT3 promotes tumorigenesis by methylating and stabilizing HIF1 α in colorectal cancer. *Cell death & disease* **12**, 1066, doi:10.1038/s41419-021-04352-w (2021).
- 22 Liao, Y. *et al.* PRMT3 drives glioblastoma progression by enhancing HIF1A and glycolytic metabolism. *Cell death & disease* **13**, 943, doi:10.1038/s41419-022-05389-1 (2022).
- 23 Hsu, M. C. *et al.* Protein Arginine Methyltransferase 3 Enhances Chemoresistance in Pancreatic Cancer by Methylating hnRNPA1 to Increase ABCG2 Expression. *Cancers* **11**, doi:10.3390/cancers11010008 (2018).
- 24 Wang, Y. *et al.* PRMT3-Mediated Arginine Methylation of METTL14 Promotes Malignant Progression and Treatment Resistance in Endometrial Carcinoma. *Advanced science (Weinheim, Baden-Wurtemberg, Germany)* **10**, e2303812, doi:10.1002/advs.202303812 (2023).
- 25 Jiang, P. *et al.* Signatures of T cell dysfunction and exclusion predict cancer immunotherapy response. *Nature medicine* **24**, 1550-1558, doi:10.1038/s41591-018-0136-1 (2018).
- 26 Philips, R. L. *et al.* The JAK-STAT pathway at 30: Much learned, much more to do. *Cell* **185**, 3857-3876, doi:10.1016/j.cell.2022.09.023 (2022).
- 27 Lv, J. *et al.* HNRNPL induced circFAM13B increased bladder cancer immunotherapy sensitivity via inhibiting glycolysis through IGF2BP1/PKM2 pathway. *Journal of experimental & clinical cancer research : CR* **42**, 41, doi:10.1186/s13046-023-02614-3 (2023).
- 28 Ni, Z. *et al.* JNK Signaling Promotes Bladder Cancer Immune Escape by Regulating METTL3-Mediated m6A Modification of PD-L1 mRNA. *Cancer research* **82**, 1789-1802, doi:10.1158/0008-5472.Can-21-1323 (2022).

- 29 Zhai, J. *et al.* ALKBH5 Drives Immune Suppression Via Targeting AXIN2 to Promote Colorectal Cancer and Is a Target for Boosting Immunotherapy. *Gastroenterology* **165**, 445-462, doi:10.1053/j.gastro.2023.04.032 (2023).
- 30 Seelig, G. F., Prosser, W. W., Hawkins, J. C. & Senior, M. M. Development of a receptor peptide antagonist to human gamma-interferon and characterization of its ligand-bound conformation using transferred nuclear Overhauser effect spectroscopy. *The Journal of biological chemistry* **270**, 9241-9249, doi:10.1074/jbc.270.16.9241 (1995).
- 31 King, M. D., Alleyne, C. H., Jr. & Dhandapani, K. M. TNF-alpha receptor antagonist, R-7050, improves neurological outcomes following intracerebral hemorrhage in mice. *Neuroscience Letters* **542**, 92-96, doi:10.1016/j.neulet.2013.02.051 (2013).

REVIEWERS' COMMENTS

Reviewer #1 (Remarks to the Author):

the authors have addressed most of the points raised by the reviewers and as a result the revised manuscript is notably improved.

Two points I would suggest to address in the discussion of the revised manuscript is (i) the lack of STAT1 binding domain on PRMT3 promoter of human - raising the concern that model proposed may not be applicable to human (ii) the lack of PRMT3 and HSP60 KO studies as per possible inflammation or immune reaction

Reviewer #2 (Remarks to the Author):

the authors performed an excellent job in this revision. multiple additional experiments were performed, supporting their conclusions.

Reviewer #3 (Remarks to the Author):

The authors have adequately addressed all my concerns. I have no more issues to report.

Reviewer #1 (Remarks to the Author):

the authors have addressed most of the points raised by the reviewers and as a result the revised manuscript is notably improved.

Two points I would suggest to address in the discussion of the revised manuscript is (i) the lack of STAT1 binding domain on PRMT3 promoter of human - raising the concern that model proposed may not be applicable to human (ii) the lack of PRMT3 and HSP60 KO studies as per possible inflammation or immune reaction

We really appreciate this reviewer for the critical and constructive comments on our manuscript, which has significantly improved the clarity of our paper. We have included the limitation of study in the discussion.

Discussion: "It has been well established that several forms of cell death, including apoptosis, can elicit inflammatory response/immune reaction through releasing damage-associated molecular patterns (DAMP). In our previous study, PRMT3-KO or PRMT3 inhibitor induced apoptosis as indicated by elevated expression of cleaved-caspase 3. We did not observe any significant up-regulation of HMGB1, a classical DAMP, in PRMT3-KD cells without stress conditions using our previous published RNA-seq dataset (GSE206502), but observed an upregulation of S100A9, another reported DAMP (increased by ~2.4 folds). Also, GSEA analysis of this RNA-seq dataset indicated that PRMT3-KD significantly up-regulated several inflammation-related pathways (Supplementary Fig. 17). Thus, we speculate that the upregulation of S100A9 and the inflammation-related pathways are caused by the impaired mitochondrial homeostasis and the activation of cGAS/STING pathway due to PRMT3 knockout or inhibition. Similarly, HSP60 knockdown could also activate inflammation-related pathways due to impaired mitochondrial homeostasis and the activation of cGAS/STING pathway. However, we cannot rule out the possibility that PRMT3 knockout, HSP60 knockdown, or the presence of PRMT3 inhibitor led to an increase in the secretion and release HMGB1 and other DAMPs that directly cause inflammation/immune reaction, which will warrant a future study."

However, we have discovered and verified the STAT1 binding domain on PRMT3 promoter of human in our manuscript (Figure 2J, K; Supplementary Figure 5K). Thus, the model proposed was also applicable to human.

Reviewer #2 (Remarks to the Author):

the authors performed an excellent job in this revision. multiple additional experiments were performed, supporting their conclusions.

We thank the reviewer for their support and helpful comments throughout the review process.

Reviewer #3 (Remarks to the Author):

The authors have adequately addressed all my concerns. I have no more issues to report.

We thank the reviewer for their support and helpful comments throughout the review process.